# Risk and cross validation in ridge regression with correlated samples

**Alexander Atanasov** [* 1 2]   **Jacob A. Zavatone-Veth** [* 2 3]   **Cengiz Pehlevan** [2 4 5]

## Abstract

Recent years have seen substantial advances in our understanding of high-dimensional ridge regression, but existing theories assume that training examples are independent. By leveraging techniques from random matrix theory and free probability, we provide sharp asymptotics for the in- and out-of-sample risks of ridge regression when the data points have arbitrary correlations. We demonstrate that in this setting, the generalized cross validation estimator (GCV) fails to correctly predict the out-of-sample risk. However, in the case where the noise residuals have the same correlations as the data points, one can modify the GCV to yield an efficiently-computable unbiased estimator that concentrates in the high-dimensional limit, which we dub CorrGCV. We further extend our asymptotic analysis to the case where the test point has nontrivial correlations with the training set, a setting often encountered in time series forecasting. Assuming knowledge of the correlation structure of the time series, this again yields an extension of the GCV estimator, and sharply characterizes the degree to which such test points yield an overly optimistic prediction of long-time risk. We validate the predictions of our theory across a variety of high dimensional data.

Statistics classically assumes that one has access to independent and identically distributed (i.i.d.) samples. However, this fundamental assumption is often violated when one considers data sampled from a time series—*e.g.*, in the case of financial, climate, or neuroscience data (Bouchaud & Potters, 2003; Mudelsee, 2014; Williams & Linderman, 2021)—rendering results obtained under the i.i.d. assumption inapplicable. In particular, estimators of the out-of-sample prediction risk based on cross-validation assume independence (Golub et al., 1979; Craven & Wahba, 1978; Bates et al., 2024; Xu & Huang, 2012; Gu & Ma, 2005; Xu et al., 2018; 2019). To enable accurate prediction of risk for regression from timeseries data, it is imperative that correlations be taken into account.

In the paradigmatic setting of ridge regression, a significant body of recent research has aimed to provide sharp asymptotic characterizations of the out-of-sample risk when the in-sample risk is estimated using i.i.d. samples (Hastie et al., 2022; Dobriban & Wager, 2018; Advani & Ganguli, 2016; Canatar et al., 2021b; Loureiro et al., 2021; Atanasov et al., 2024; Mel & Pennington, 2021; Mel & Ganguli, 2021; Bordelon et al., 2020; Jacot et al., 2020; Gerace et al., 2020). The crucial feature of these high-dimensional asymptotics is that they allow both the dimensionality of the covariates and the number of examples to be large.[1] This body of work reveals two broad principles: First, ridge regression has a spectral bias towards learning functions aligned with eigenfunctions of the feature covariance (Canatar et al., 2021b; Bordelon et al., 2020; Mel & Pennington, 2021; Mel & Ganguli, 2021). Second, ridge regression displays *Gaussian universality* in high dimension, *i.e.*, the out-of-sample risk for a given dataset will under general conditions be asymptotically identical to that for a Gaussian dataset with matched first and second moments (Hastie et al., 2022; Montanari & Saeed, 2022; Hu & Lu, 2022). Importantly, nearly all of these works assume the training examples are i.i.d.. A rare exception is a recent paper by Bigot et al. (2024), who allow for *independent but non-identically distributed Gaussian data* with per-example modulation of total variance.

These sharp asymptotics are *omniscient* risk estimates: they assume one has access to the true joint distribution of covariates and labels from which the training set is sampled. If one wants to perform hyperparameter tuning in practice—for ridge regression this of course means tuning the ridge parameter—a non-omniscient estimate is required. A standard approach to this problem is the method of generalized

---

[*]Equal contribution  [1]Department of Physics, Harvard University [2]Center for Brain Science, Harvard University [3]Society of Fellows, Harvard University [4]The John A. Paulson School of Engineering and Applied Sciences, Harvard University [5]The Kempner Institute for the Study of Natural and Artificial Intelligence, Harvard University. Correspondence to: JAZ-V <jzavatoneveth@fas.harvard.edu>, CP <cpehlevan@seas.harvard.edu>.

*Proceedings of the 42$^{nd}$ International Conference on Machine Learning*, Vancouver, Canada. PMLR 267, 2025. Copyright 2025 by the author(s).

---

[1]In contrast, classical asymptotics assume that the dimension is small relative to the number of examples (Hastie et al., 2009).

cross-validation, which dates back at least to the 1970s (Golub et al., 1979; Craven & Wahba, 1978; Bates et al., 2024). The GCV estimates the out-of-sample risk by applying a multiplicative correction to the in-sample risk, which itself can be estimated from the data. Recent works have shown that the GCV is asymptotically exact in high dimensions: the limiting GCV estimate coincides with the omniscient asymptotic (Jacot et al., 2020; Hastie et al., 2022; Atanasov et al., 2024). Though classical cross-validation and the GCV estimator assume i.i.d. training data points, several works have aimed to extend these methods to cases in which there are correlations (Altman, 1990; Opsomer et al., 2001; Carmack et al., 2012; Rabinowicz & Rosset, 2022; Lukas, 2010; Wang, 1998). However, most of these works focus on correlations only in the label noise, and none of them show that their estimators are asymptotically exact in high dimensions. Indeed, we will show that none of the previously-proposed corrections to the GCV are asymptotically exact in high dimensions, and require further modification to accurately predict out-of-sample risk.

Here, we fill this gap in understanding by providing a detailed asymptotic characterization of ridge regression with correlated samples. We first show that even under relatively mild sample-sample correlations previously-obtained omniscient risk asymptotics assuming i.i.d. data are not predictive. Correspondingly, the ordinary GCV estimator fails to accurately estimate the out-of-sample risk. We then compute sharp high-dimensional asymptotics for the out-of-sample risk when the training examples are drawn from a general matrix Gaussian with anisotropic correlations across both features and samples. When the test point is uncorrelated with the training data and the label noise has the same correlation structure as the covariates, we show that there exists a corrected GCV estimator. We term this estimator the **CorrGCV**. Unlike previous attempts to correct the original GCV, CorrGCV is asymptotically exact (Figure 1).[2]

In deriving the CorrGCV, we uncover an interesting duality between train-test covariate shift and a mismatch in the covariate and noise correlations. Finally, we extend all of these results to the case when the test point is correlated with the training points. Focusing on the setting of time series regression, this gives us a sharp characterization of how accuracy depends on prediction horizon, and makes precise the notion that testing on near-horizon data gives an overly optimistic picture of long-term forecast accuracy. In all, our results both advance the theoretical understanding of ridge regression in high dimensions and provide application to time series data.

---

[2]Code to compute the CorrGCV and reproduce all of our experiments is publicly available on GitHub at https://github.com/Pehlevan-Group/S_transform.

## 1. Setup and notation

We begin by briefly introducing the setup of our work, and fixing notation. We consider ridge regression on a dataset $\mathcal{D} = \{\boldsymbol{x}_t, y_t\}_{t=1}^T$ of $T$ data points, with $N$-dimensional covariates $\boldsymbol{x}_t \in \mathbb{R}^N$ and scalar labels $y_t$. We minimize the mean-squared error over this dataset, with a ridge penalty:

$$L(\boldsymbol{w}) = \frac{1}{T} \sum_{t=1}^T (y_t - \boldsymbol{x}_t^\top \boldsymbol{w})^2 + \lambda \|\boldsymbol{w}\|^2.$$

We write $\hat{\boldsymbol{w}} = \operatorname{argmin}_{\boldsymbol{w}} L(\boldsymbol{w})$. As long as $\lambda > 0$ the solution is unique. We will consider both the underparameterized case of classical statistics where $T > N$ as well as the modern overparameterized setting where $N > T$ (Hastie et al., 2022). We define the overparameterization ratio $q \equiv N/T$. All of our results will hold exactly in the limit of $N, T \to \infty$ with $q$ fixed.

We now state our statistical assumptions on the data. Defining the design matrix $\boldsymbol{X} \in \mathbb{R}^{T \times N}$ such that $\boldsymbol{X}_{ti} = [\boldsymbol{x}_t]_i$ and collecting the labels into a vector $\boldsymbol{y} \in \mathbb{R}^T$, we assume the labels are generated from a deterministic linear "teacher" $\bar{\boldsymbol{w}} \in \mathbb{S}^{N-1}$ plus noise $\boldsymbol{\epsilon} \in \mathbb{R}^T$ as:

$$\boldsymbol{y} = \boldsymbol{X}\bar{\boldsymbol{w}} + \boldsymbol{\epsilon}.$$

We assume that $\boldsymbol{X}$ is Gaussian, with zero mean and

$$\mathbb{E}[x_{i,t} x_{j,s}] = \Sigma_{ij} K_{ts} \tag{1}$$

for a feature-feature covariance $\boldsymbol{\Sigma} \in \mathbb{R}^{N \times N}$ and a sample-sample correlation $\boldsymbol{K} \in \mathbb{R}^{T \times T}$. We have the representation $\boldsymbol{X} = \boldsymbol{K}^{1/2} \boldsymbol{Z} \boldsymbol{\Sigma}^{1/2}$ for $\boldsymbol{Z} \in \mathbb{R}^{T \times N}$ a matrix with i.i.d. standard Gaussian elements $Z_{ti} \sim \mathcal{N}(0, 1)$, where $\boldsymbol{\Sigma}^{1/2}$ denotes the principal square root of the positive-definite symmetric matrix $\boldsymbol{\Sigma}$. We assume that the noise $\boldsymbol{\epsilon}$ is independent of $\boldsymbol{X}$, and is Gaussian with mean zero and covariance

$$\mathbb{E}[\epsilon_t \epsilon_s] = \sigma_\epsilon^2 K_{ts}'.$$

Without loss of generality, we will always assume the normalization $\frac{1}{T} \operatorname{Tr}(\boldsymbol{K}) = 1$, as the overall scale can be absorbed into $\boldsymbol{\Sigma}$. We will often assume that the data are statistically stationary, in which case $K_{tt} = 1$ for all $t$; we will explicitly highlight when we impose this condition. Similarly, we normalize $\frac{1}{T} \operatorname{Tr}(\boldsymbol{K}') = 1$, as the scale here can be absorbed into $\sigma_\epsilon$.

We contrast this with prior treatments of linear regression in the proportional limit where $\boldsymbol{K}, \boldsymbol{K}'$ were chosen to be the identity matrix (Hastie et al., 2022; Dobriban & Wager, 2018; Advani & Ganguli, 2016; Canatar et al., 2021b; Loureiro et al., 2021; Atanasov et al., 2024; Mel & Pennington, 2021; Mel & Ganguli, 2021; Bordelon et al., 2020; Jacot et al., 2020; Gerace et al., 2020), and where the original GCV estimator (Golub et al., 1979) applies. In Section 3.2 we consider the case where $\boldsymbol{K} = \boldsymbol{K}'$. There, we show that the GCV estimator (Golub et al., 1979) has a natural analogue. In Section 3.3 we consider the more general case and show that there is an obstruction to a GCV estimator.

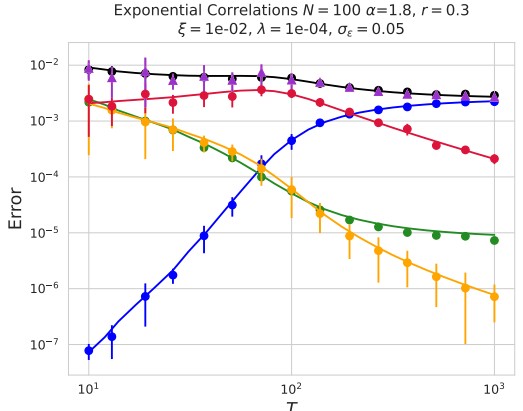
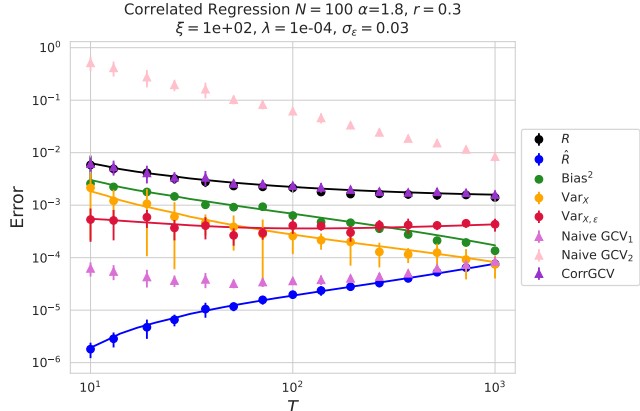

*Figure 1.* Empirical risk $\hat{R}$, out-of-sample risk $R$, and fine-grained bias-variance decompositions for ridge regression with structured features and correlated examples. Theory is plotted in solid lines. Experiments with error bars over 10 dataset repetitions are plotted as markers. The data points are exponentially correlated as $\mathbb{E}[\boldsymbol{x}_t \cdot \boldsymbol{x}_s] \propto e^{-|t-s|/\xi}$. Left: Weak correlations, $\xi = 10^{-2}$. Here, the generalized cross validation method (orchid) as well as its other proposed corrections in the presence of correlations (pink, purple) all agree and are overlaid. Right: Strong correlations, $\xi = 10^2$. Here, we see that the naive estimates of the GCV proposed in prior works fail in this setting. They either underestimate (purple) or overestimate (pink) the out-of-sample risk. We define the naive GCVs in the text, and connect them with prior proposals in Appendix G. By contrast, our proposed estimator, CorrGCV, correctly predicts the out-of-sample risk in all settings.

We define the **empirical covariance** $\hat{\boldsymbol{\Sigma}} \equiv \frac{1}{T}\boldsymbol{X}^\top \boldsymbol{X}$ and **kernel Gram matrix** $\hat{K} \equiv \frac{1}{T}\boldsymbol{X}\boldsymbol{X}^\top$. Writing $\boldsymbol{y} \in \mathbb{R}^T$ as the vector of training labels and $\boldsymbol{\epsilon} \in \mathbb{R}^T$ as the vector of label noises, we have:

$$\hat{\boldsymbol{w}} = (\hat{\boldsymbol{\Sigma}}+\lambda)^{-1}\frac{\boldsymbol{X}^\top \boldsymbol{y}}{T} = \hat{\boldsymbol{\Sigma}}(\hat{\boldsymbol{\Sigma}}+\lambda)^{-1}\bar{\boldsymbol{w}}+(\hat{\boldsymbol{\Sigma}}+\lambda)^{-1}\frac{\boldsymbol{X}^\top \boldsymbol{\epsilon}}{T}.$$

We also let $\hat{\boldsymbol{y}} = \boldsymbol{X}\hat{\boldsymbol{w}}$ be the vector of predictions on the training set. We are interested in analytically characterizing the in-sample and out-of-sample risks. The in-sample risk is defined as:

$$\hat{R}_{in}(\hat{\boldsymbol{w}}) \equiv \frac{1}{T}\|\boldsymbol{y} - \hat{\boldsymbol{y}}\|^2.$$

For the out of sample risk, we we consider a held-out test point $\boldsymbol{x}$ and label noise $\epsilon$ drawn from the same marginal distribution as a single example $\boldsymbol{x}_t, \epsilon_t$, discarding the time-dependent scale factor $K_{tt}$ in the marginal:

$$R_{out}(\hat{\boldsymbol{w}}) \equiv \mathbb{E}_{\boldsymbol{x},\epsilon}(\boldsymbol{x}^\top \bar{\boldsymbol{w}} + \epsilon - \boldsymbol{x}^\top \hat{\boldsymbol{w}})^2$$
$$= \underbrace{(\bar{\boldsymbol{w}} - \hat{\boldsymbol{w}})^\top \boldsymbol{\Sigma}(\bar{\boldsymbol{w}} - \hat{\boldsymbol{w}})^\top}_{R_g} + \sigma_\epsilon^2.$$

Here we have identified the generalization error $R_g$ as the excess risk, *i.e.*, $R_{out}$ minus the Bayes error $\sigma_\epsilon^2$. We study the case where the test point and label noise are drawn independently from the test set in Section 3. We study the more general case where $\boldsymbol{x}, \epsilon$ have nontrivial correlations with the training set in Section 5.

One could instead consider an estimator based on a weighted loss, *i.e.*, $\hat{\boldsymbol{w}}_{\boldsymbol{M}} = \arg\min_{\boldsymbol{w}} L_{\boldsymbol{M}}(\boldsymbol{w})$ for

$$L_{\boldsymbol{M}}(\boldsymbol{w}) = \frac{1}{T}(\boldsymbol{X}\boldsymbol{w} - \boldsymbol{y})^\top \boldsymbol{M}(\boldsymbol{X}\boldsymbol{w} - \boldsymbol{y}) + \lambda|\boldsymbol{w}|^2$$

for some positive-definite matrix $\boldsymbol{M}$ (Bouchaud & Potters, 2003; Hastie et al., 2009). Indeed, the Bayesian minimum mean squared error estimator is equivalent to minimizing this loss with $\boldsymbol{M} = (\boldsymbol{K}')^{-1}$ (Appendix H). However, under our assumptions on the data this is equivalent to considering an isotropic loss under the mapping $\boldsymbol{K} \leftarrow \boldsymbol{M}^{1/2}\boldsymbol{K}\boldsymbol{M}^{1/2}$ and $\boldsymbol{K}' \leftarrow \boldsymbol{M}^{1/2}\boldsymbol{K}'\boldsymbol{M}^{1/2}$. As a result, our asymptotics for $R_g$ apply immediately to general choices of $\boldsymbol{M}$.

## 2. Deterministic equivalences

We now introduce the key tools of our analysis: deterministic equivalents for the sample covariance. We first review standard aspects of the free probability approach to random matrices, and then state the required deterministic equivalents. Our presentation of free probability follows Potters & Bouchaud (2020).

### 2.1. Weak deterministic equivalents

We define the first and second **degrees of freedom** of a matrix $\boldsymbol{A} \in \mathbb{R}^{N \times N}$ as

$$\mathrm{df}_{\boldsymbol{A}}^1(\lambda) = \frac{1}{N}\mathrm{Tr}[\boldsymbol{A}(\boldsymbol{A} + \lambda)^{-1}],$$
$$\mathrm{df}_{\boldsymbol{A}}^2(\lambda) = \frac{1}{N}\mathrm{Tr}[\boldsymbol{A}^2(\boldsymbol{A} + \lambda)^{-2}].$$

When it is clear from context, we will write these as $\mathrm{df}_1, \mathrm{df}_2$. The $S$-transform $S(\mathrm{df})$ is a formal function of a variable $\mathrm{df}$ defined as

$$S_{\boldsymbol{A}}(\mathrm{df}) \equiv \frac{1 - \mathrm{df}}{\mathrm{df}\,\mathrm{df}_{\boldsymbol{A}}^{-1}(\mathrm{df})}. \tag{2}$$

Here $\mathrm{df}_{\boldsymbol{A}}^{-1}$ is the functional inverse of the equation $\mathrm{df} = \mathrm{df}_{\boldsymbol{A}}^1(\lambda)$, *i.e.*, $\mathrm{df}_{\boldsymbol{A}}^{-1}(\mathrm{df}_{\boldsymbol{A}}^1(\lambda)) = \lambda$. We will use the symbol $\mathrm{df}$ to highlight its role as a formal variable in the $S$-transform, whereas we will call $\mathrm{df}_1$ the actual value of the degrees of freedom for a given matrix $\boldsymbol{A}$ at a ridge $\lambda$. Our definition of $S_{\boldsymbol{A}}$ differs by a sign from the common definition of the $S$ transform in terms of the $t$-function $t_{\boldsymbol{A}}(\lambda) = -\mathrm{df}_{\boldsymbol{A}}^1(-\lambda)$. (2) implies that:

$$\mathrm{df}_{\boldsymbol{A}}^1(\lambda) = \frac{1}{1 + \lambda S_{\boldsymbol{A}}(\mathrm{df}_{\boldsymbol{A}}^1(\lambda))}.$$

The $S$-transform plays a crucial role in high-dimensional random matrix theory and free probability because for two symmetric matrices $\boldsymbol{A}, \boldsymbol{B}$ that are **free** of one another, taking the symmetrized product $\boldsymbol{A} * \boldsymbol{B} \equiv \boldsymbol{A}^{1/2} \boldsymbol{B} \boldsymbol{A}^{1/2}$ with $\boldsymbol{A}^{1/2}$ the principal matrix square root yields:

$$S_{\boldsymbol{A}*\boldsymbol{B}}(\mathrm{df}) = S_{\boldsymbol{A}}(\mathrm{df})S_{\boldsymbol{B}}(\mathrm{df}).$$

This implies the following **subordination relation**:

$$
\begin{aligned}
\mathrm{df}_{\boldsymbol{A}*\boldsymbol{B}}^1(\lambda) &= \frac{1}{1 + \lambda S_{\boldsymbol{A}*\boldsymbol{B}}(\mathrm{df}_{\boldsymbol{AB}}^1(\lambda))} \\
&= \frac{1}{1 + \lambda S_{\boldsymbol{A}}(\mathrm{df}_{\boldsymbol{AB}}^1(\lambda))S_{\boldsymbol{B}}(\mathrm{df}_{\boldsymbol{AB}}^1(\lambda))} \\
&= \mathrm{df}_{\boldsymbol{A}}^1(\lambda S_{\boldsymbol{B}}(\mathrm{df}_{\boldsymbol{AB}}^1(\lambda))).
\end{aligned}
\tag{3}
$$

We define what it means for two random variables to be free, and give a longer discussion on subordination relations and $R$ and $S$ transforms in Appendix B. Often, one views $\boldsymbol{B}$ as a source of multiplicative noise, and $\boldsymbol{A}$ as deterministic. Then $\boldsymbol{A} * \boldsymbol{B}$ is a random matrix. The above equation thus relates the degrees of freedom of a random matrix to the degrees of freedom of a deterministic one. For this reason it is known as (weak) **deterministic equivalence**.

We now specialize to the Wishart matrices $\hat{\boldsymbol{\Sigma}} = \frac{1}{T}\boldsymbol{X}^\top \boldsymbol{X}$ and $\hat{\boldsymbol{K}} = \frac{1}{T}\boldsymbol{X}\boldsymbol{X}^\top$ we consider here, though a subset of the subsequent results extend to more general ensembles (see Appendix B and (Potters & Bouchaud, 2020; Atanasov et al., 2024)). Define the shorthand $\mathrm{df}_1 \equiv \mathrm{df}_{\boldsymbol{\Sigma}}^1(\kappa)$, $\tilde{\mathrm{df}}_1 \equiv \mathrm{df}_{\boldsymbol{K}}^1(\tilde{\kappa})$, $\mathrm{df}_2 \equiv \mathrm{df}_{\boldsymbol{\Sigma}}^2(\kappa)$, and $\tilde{\mathrm{df}}_2 \equiv \mathrm{df}_{\boldsymbol{K}}^2(\tilde{\kappa})$. Then, we define $\kappa$ and $\tilde{\kappa}$ via:

$$\kappa = \lambda S(\mathrm{df}_1), \quad \tilde{\kappa} = \lambda \tilde{S}(\tilde{\mathrm{df}}_1). \tag{4}$$

As the multiplicative noise is a *structured* Wishart matrix, the $S$-transforms appearing in (4) are

$$
\begin{aligned}
S(\mathrm{df}_1) &= S_{\frac{1}{T}\boldsymbol{Z}^\top \boldsymbol{Z}}(\mathrm{df}_1) S_{\boldsymbol{K}}\left(\frac{N}{T}\mathrm{df}_1\right), \\
\tilde{S}(\tilde{\mathrm{df}}_1) &= S_{\frac{1}{T}\boldsymbol{Z}\boldsymbol{Z}^\top}(\tilde{\mathrm{df}}_1) S_{\boldsymbol{\Sigma}}\left(\frac{T}{N}\tilde{\mathrm{df}}_1\right).
\end{aligned}
\tag{5}
$$

For uncorrelated data ($\boldsymbol{K} = \boldsymbol{I}$), $S_{\boldsymbol{K}} = 1$. Weak deterministic equivalents for Wishart matrices with general correlation structure have long been a subject of study in random matrix theory due to their applications to covariance matrix estimation (Burda et al., 2005b;a; Burda & Jarosz, 2022; Potters & Bouchaud, 2020).

## 2.2. One-point strong deterministic equivalents

The equivalence (3) extends to a class of 'equalities' of matrices known as strong deterministic equivalents:

**Definition 2.1** (Strong deterministic equivalence). For two sequences of (possibly random) matrices $\boldsymbol{A}$ and $\boldsymbol{B}$ indexed by their common size $N$, we say that $\boldsymbol{A}$ and $\boldsymbol{B}$ are **deterministically equivalent** and write $\boldsymbol{A} \simeq \boldsymbol{B}$ if $\mathrm{Tr}(\boldsymbol{A}\boldsymbol{M})/\mathrm{Tr}(\boldsymbol{B}\boldsymbol{M}) \to 1$ in probability as $N \to \infty$ for any sequence of test matrices $\boldsymbol{M}$ of bounded spectral norm.[3]

For the Wishart matrices $\hat{\boldsymbol{\Sigma}}$ and $\hat{\boldsymbol{K}}$, we have:

**Lemma 2.2.** *Let $\kappa$ and $\tilde{\kappa}$ be as in* (3). *Then,*

$$
\begin{aligned}
\hat{\boldsymbol{\Sigma}}(\hat{\boldsymbol{\Sigma}} + \lambda)^{-1} &\simeq \boldsymbol{\Sigma}(\boldsymbol{\Sigma} + \kappa)^{-1}, \\
\hat{\boldsymbol{K}}(\hat{\boldsymbol{K}} + \lambda)^{-1} &\simeq \boldsymbol{K}(\boldsymbol{K} + \tilde{\kappa})^{-1}.
\end{aligned}
\tag{6}
$$

*Proof.* We derive (6) using a diagrammatic argument in Appendix C.1; see Potters & Bouchaud (2020); Bun et al. (2016a); Atanasov et al. (2024) for alternative proofs. □

Using the one-point deterministic equivalents, we see that $\kappa$ and $\tilde{\kappa}$ are related by the identity

$$q\mathrm{df}_1 \equiv q\mathrm{df}_{\boldsymbol{\Sigma}}^1(\kappa) \simeq q\mathrm{df}_{\hat{\boldsymbol{\Sigma}}}^1(\lambda) = \mathrm{df}_{\hat{\boldsymbol{K}}}^1(\lambda) \simeq \mathrm{df}_{\boldsymbol{K}}^1(\tilde{\kappa}) \equiv \tilde{\mathrm{df}}_1$$

In addition, we show in Appendix C that $\kappa$ and $\tilde{\kappa}$ satisfy the *duality relation*:

$$\frac{\kappa\tilde{\kappa}}{\lambda} = \frac{1}{\tilde{\mathrm{df}}_1}.$$

In physical terms, we can interpret (6) as a renormalization effect: the effect of the random fluctuations in $\boldsymbol{B}$ can be absorbed into a **renormalized ridge** $\kappa$ (Atanasov et al., 2024). This renormalization generates implicit regularization. Even in the limit of zero regularization, one can have $\lim_{\lambda \to 0} \kappa > 0$ (Appendix F).

## 2.3. Two-point strong deterministic equivalents

As (6) involves a trace against a single test matrix, we refer to it as a 'one-point' equivalent. We will also require 'two-point' equivalents involving pairs of resolvents:

**Lemma 2.3.** *Let $\boldsymbol{\Sigma}'$ be an $N \times N$ test matrix. Then,*

$$
\begin{aligned}
(\hat{\boldsymbol{\Sigma}} + \lambda)^{-1}\boldsymbol{\Sigma}'(\hat{\boldsymbol{\Sigma}} + \lambda)^{-1} &\simeq S^2(\boldsymbol{\Sigma} + \kappa)^{-1}\boldsymbol{\Sigma}'(\boldsymbol{\Sigma} + \kappa)^{-1} \\
&\quad + S^2(\boldsymbol{\Sigma} + \kappa)^{-2}\boldsymbol{\Sigma}\frac{\gamma_{\boldsymbol{\Sigma},\boldsymbol{\Sigma}'}}{1 - \gamma}.
\end{aligned}
$$

*where we define*

$$\gamma \equiv \frac{\mathrm{df}_2}{\mathrm{df}_1}\frac{\tilde{\mathrm{df}}_2}{\tilde{\mathrm{df}}_1}, \quad \text{and} \quad \gamma_{\boldsymbol{\Sigma},\boldsymbol{\Sigma}'} \equiv \frac{\mathrm{df}_{\boldsymbol{\Sigma},\boldsymbol{\Sigma}'}^2 \tilde{\mathrm{df}}_{\boldsymbol{K}}^2}{\mathrm{df}_1\tilde{\mathrm{df}}_1}$$

*for*

$$\mathrm{df}_{\boldsymbol{\Sigma},\boldsymbol{\Sigma}'}^2 \equiv \frac{1}{N}\mathrm{Tr}\left[\boldsymbol{\Sigma}\boldsymbol{\Sigma}'(\boldsymbol{\Sigma} + \kappa)^{-2}\right].$$

*Proof.* See Appendix C.2. □

---

[3]Our convention here follows Bach (2024).

When $\boldsymbol{K} = \mathbf{I}$, this recovers two-point equivalents proved in previous works (Bach, 2024; Patil & LeJeune, 2024). The first term is a "disconnected" component coming from separately averaging the two resolvents, while the second term is a "connected" component coming from averaging them together. These two contributions correspond to the bias-variance decomposition of the estimator over the training data. An analogous deterministic equivalent holds for $\boldsymbol{K}'$; we state this explicitly in Appendix C.

We also derive a final required two-point equivalent:

**Lemma 2.4.** *Let $\boldsymbol{K}'$ be a $T \times T$ test matrix. Then,*
$$(\hat{\boldsymbol{\Sigma}} + \lambda)^{-1} \boldsymbol{X}^\top \boldsymbol{K}' \boldsymbol{X} (\hat{\boldsymbol{\Sigma}} + \lambda)^{-1}$$
$$\simeq S^2 \tilde{S}^2 (\boldsymbol{\Sigma} + \lambda)^{-2} \boldsymbol{\Sigma} \frac{\mathrm{df}^2_{\boldsymbol{K},\boldsymbol{K}'}}{1 - \gamma},$$
*where we let $\mathrm{df}^2_{\boldsymbol{K},\boldsymbol{K}'} \equiv \frac{1}{T} \mathrm{Tr}\left[ \boldsymbol{K}\boldsymbol{K}'(\boldsymbol{K} + \kappa)^{-2} \right]$.*

*Proof.* See Appendix C.2. □

## 3. Predicting an uncorrelated test set

### 3.1. Warm-up: Linear regression without correlations

We begin by reviewing the known results in the case where the data points are assumed to be drawn i.i.d. from a distribution with covariance $\boldsymbol{\Sigma}$. This is case of $\boldsymbol{K} = \mathbf{I}$ introduced above. In this case, we require only $\kappa$ and not $\tilde{\kappa}$, which satisfies the simplified equation
$$\kappa = \lambda S_{\frac{1}{T} \boldsymbol{Z}^\top \boldsymbol{Z}}(\mathrm{df}_1) = \frac{\lambda}{1 - q\mathrm{df}_1}, \tag{7}$$
where $\mathrm{df}_1 \equiv \mathrm{df}^1_{\boldsymbol{\Sigma}}(\kappa) \simeq \mathrm{df}^1_{\hat{\boldsymbol{\Sigma}}}(\lambda)$. Here we have used that the $S$-transform of a Wishart matrix is $S_{\frac{1}{T} \boldsymbol{Z}^\top \boldsymbol{Z}} = (1 - q\mathrm{df}_1)^{-1}$. At this point, we note that depending on whether one picks $\mathrm{df}_1 = \mathrm{df}^1_{\boldsymbol{\Sigma}}(\kappa)$ or $\mathrm{df}_1 = \mathrm{df}^1_{\hat{\boldsymbol{\Sigma}}}(\lambda)$, this either gives a self-consistent equation given omniscient knowledge of $\boldsymbol{\Sigma}$ or a way to estimate $\kappa$ from the data alone, namely by computing $\mathrm{df}_{\hat{\boldsymbol{\Sigma}}}(\lambda)$. Then, one has the following deterministic equivalents:

**Theorem 3.1.** *For uncorrelated data ($\boldsymbol{K} = \mathbf{I}$), one has*
$$R_g \simeq \frac{\kappa^2}{1 - \gamma} \bar{\boldsymbol{w}}^\top \boldsymbol{\Sigma}(\boldsymbol{\Sigma} + \kappa)^{-2} \bar{\boldsymbol{w}} + \frac{\gamma}{1 - \gamma} \sigma_\epsilon^2,$$
*where $\gamma = q\mathrm{df}_2$, and*
$$\hat{R}_{in} \simeq \frac{\lambda^2}{1 - \gamma} \bar{\boldsymbol{w}}^\top \boldsymbol{\Sigma}(\boldsymbol{\Sigma} + \kappa)^{-2} \bar{\boldsymbol{w}} + \frac{\lambda^2}{\kappa^2} \frac{1}{1 - \gamma} \sigma_\epsilon^2.$$

*Proof.* This result is well-known (Hastie et al., 2022; Atanasov et al., 2024; Bordelon et al., 2020; Canatar et al., 2021b; Dobriban & Wager, 2018; Loureiro et al., 2021). It follows as a special case of the proof in Appendix C.4. □

This yields the **generalized-cross-validation estimator** or

**GCV** (Golub et al., 1979; Craven & Wahba, 1978):
$$R_{out} \simeq \frac{\kappa^2}{\lambda^2} \hat{R}_{in} = S^2 \hat{R}_{in}.$$
Since $S$ depends only on $\mathrm{df}_1$, which can be estimated solely from the data, the right hand side of this equation yields a way to estimate the out of sample error from the training error alone. The GCV estimator has also appeared in recent literature as the **kernel alignment risk estimator** or KARE (Jacot et al., 2020). The fact that the GCV estimator is directly related to the $S$-transform in the free probability regime $N, T \to \infty$ was first pointed out in our previous work (Atanasov et al., 2024).

### 3.2. Correlated data with identically correlated noise

We now state the results for general correlated data with matched correlations between the covariates and noise ($\boldsymbol{K} = \boldsymbol{K}'$). Using the deterministic equivalents introduced in Section 2, we obtain:

**Theorem 3.2.** *Assume $\boldsymbol{K} = \boldsymbol{K}'$. Then, we have*
$$R_g \simeq \frac{\kappa^2}{1 - \gamma} \bar{\boldsymbol{w}}^\top \boldsymbol{\Sigma}(\boldsymbol{\Sigma} + \kappa)^{-2} \bar{\boldsymbol{w}} + \frac{\gamma}{1 - \gamma} \sigma_\epsilon^2,$$
*where*
$$\gamma \equiv \frac{\mathrm{df}_2}{\mathrm{df}_1} \frac{\tilde{\mathrm{df}}_2}{\tilde{\mathrm{df}}_1}.$$

*Similarly, we obtain:*
$$\hat{R}_{in} \simeq \frac{\tilde{\mathrm{df}}_1 - \tilde{\mathrm{df}}_2}{S\tilde{\mathrm{df}}_1} \frac{\kappa^2}{1 - \gamma} \bar{\boldsymbol{w}}^\top \boldsymbol{\Sigma}(\boldsymbol{\Sigma} + \kappa)^{-1} \bar{\boldsymbol{w}}$$
$$+ \frac{\tilde{\mathrm{df}}_1 - \tilde{\mathrm{df}}_2}{S\tilde{\mathrm{df}}_1} \frac{1}{1 - \gamma} \sigma_\epsilon^2.$$

*Proof.* See Appendix C.4. □

This yields an extension of the GCV estimator to correlated data. We call this the **CorrGCV** for **correlated generalized cross-validation**:
$$R_{out} = S(\mathrm{df}_1) \frac{\tilde{\mathrm{df}}_1}{\tilde{\mathrm{df}}_1 - \tilde{\mathrm{df}}_2} \hat{R}_{in}.$$
This estimator is unbiased and asymptotically exact in the proportional limit of $T, N \to \infty$. Moreover, it concentrates over draws of the dataset for $T$ sufficiently large.

Like the ordinary GCV, the CorrGCV can be estimated from the training data alone. We give the explicit algorithm in Appendix A. In Figures 1, 2, 3, 4, and 6 and Appendices G and J we compare this estimator with other previously proposed estimators of out-of-sample risk in correlated data. In particular, we compare against the ordinary GCV
$$\text{Naïve GCV}_1 = \frac{1}{(1 - q\mathrm{df}_1)^2},$$

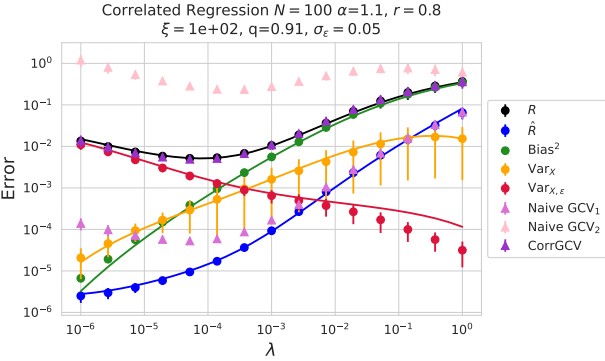

Figure 2. Estimating the optimal ridge parameter for exponential correlations using the CorrGCV. The setup here is as in Figure 1. We see that only the CorrGCV accurately predicts the out-of-sample risk, and thus is the only estimator that allows one to correctly pinpoint the optimal ridge parameter $\lambda$.

which does not take into account the sample-sample correlations, and the estimator proposed by Altman (1990),

$$\text{Naïve GCV}_2 = S^2 = (\kappa/\lambda)^2,$$

which was designed to account for sample-sample correlations in the label noise but not in the covariates. Further comparisons to the estimator of Carmack et al. (2012) are given in Appendix G. When the samples are correlated, only the CorrGCV accurately predicts the out-of-sample risk. As a result, it is the only correction to the GCV that allows accurate tuning of the ridge parameter $\lambda$ (Figure 2).

An important contribution from the theory of high-dimensional regression with uncorrelated samples is a characterization of how power-law decays in covariance eigenspectra—as occur for a variety of real data—give rise to power-law decays in out-of-sample risk $R_g$ as a function of sample size $T$ (Caponnetto & De Vito, 2007; Spigler et al., 2020; Bordelon et al., 2020; Cui et al., 2021; Atanasov et al., 2024). Thus, it is natural to ask whether correlations between samples can alter these scaling laws. We show in Appendix E that the correlation structure of the stationary processes we consider cannot change the decay rates. Concretely, assume that $\Sigma$ has eigenvalues $\lambda_k \sim k^{-\alpha}$ for an exponent $\alpha$ known as the **capacity**. Further, the signal $\bar{w}$ has that $\lambda_k \bar{w}_k^2 \sim k^{-(2\alpha r+1)}$ for an exponent $r$ known as the **source**. For correlated data, the scaling is unchanged from prior predictions of optimal rates in (Caponnetto & De Vito, 2007; Spigler et al., 2020; Bordelon et al., 2020; Cui et al., 2021), namely that $R_g \sim T^{-2\alpha \min(r,1)}$. We illustrate this phenomenon in Figure 3.

### 3.3. Mismatched correlations and OOD generalization

We generalize the above result to a setting where $\epsilon$ does not have the same correlation structure as $K$. That is, $\mathbb{E}[\epsilon_t \epsilon_s] = \sigma_\epsilon^2 K'_{ts}$. In fact, we can consider an even more general case: namely when the covariance of the test point $\Sigma'$ is also different from the covariance of the training set $\Sigma$. This is the case of out-of-distribution generalization under covariate shift. These two different mismatches exhibit a surprising duality. In general, we have

**Theorem 3.3.** *Consider a covariate-shifted setting with test covariance $\Sigma'$ not necessarily equal to $\Sigma$, and noise covariance $K'$ that may not match $K$. Then, we have*

$$R_g \simeq \underbrace{\kappa^2 \bar{w}(\Sigma + \kappa)^{-1}\Sigma'(\Sigma + \kappa)^{-1}\bar{w}}_{\text{Bias}^2}$$
$$+ \underbrace{\kappa^2 \frac{\gamma_{\Sigma,\Sigma'}}{1-\gamma}\bar{w}\Sigma(\Sigma + \kappa)^{-2}\bar{w}}_{\text{Var}_X} + \underbrace{\frac{\gamma_{\Sigma,\Sigma',K,K'}}{1-\gamma}\sigma_\epsilon^2}_{\text{Var}_{X\epsilon}}.$$
(8)

*Here, we have highlighted how the risk naturally splits into three terms given by respective bias and variance components resulting from the covariates $X$ and the noise $\epsilon$, as outlined in Appendix D (Atanasov et al., 2024). We have also defined:*

$$\gamma_{\Sigma,\Sigma'} \equiv \frac{\text{df}^2_{\Sigma,\Sigma'}\text{df}^2_K}{\text{df}_1\tilde{\text{df}}_1}, \quad \gamma_{\Sigma,\Sigma',K,K'} \equiv \frac{\text{df}^2_{\Sigma,\Sigma'}\text{df}^2_{K,K'}}{\text{df}_1\tilde{\text{df}}_1},$$

$$\text{df}^2_{\Sigma,\Sigma'} \equiv \frac{1}{N}\text{Tr}\left[\Sigma\Sigma'(\Sigma + \kappa)^{-2}\right],$$

$$\text{df}^2_{K,K'} \equiv \frac{1}{T}\text{Tr}\left[KK'(K + \tilde{\kappa})^{-2}\right].$$

*Similarly, for the training error, we have*

$$\hat{R}_{in} \simeq \frac{\tilde{\text{df}}_1 - \tilde{\text{df}}_2}{S\tilde{\text{df}}_1}\frac{\kappa^2}{1-\gamma}\bar{w}^\top \Sigma(\Sigma + \kappa)^{-2}\bar{w}$$

$$+ \sigma_\epsilon^2 \tilde{\kappa}\left[\frac{1}{T}\text{Tr}\,K'(K + \tilde{\kappa})^{-1} - \frac{\text{df}_1 - \text{df}_2}{1-\gamma}\frac{\text{df}^2_{K,K'}}{\text{df}_1}\right].$$

*Proof.* See Appendix C.4. □

We validate these formulae across several experiments (Figure 4, Appendix J). As a special case, setting $K = \mathbf{I}_T$ recovers previously-obtained asymptotics for the risk of ridge regression under covariate shift (Patil et al., 2024; Canatar et al., 2021a; Tripuraneni et al., 2021). In general, the training error is proportional to the generalization error only in the setting where $\Sigma = \Sigma'$ and $K = K'$. Thus, it is only in this setting that a GCV exists.

### 3.4. Effect on double descent

It is interesting to consider how correlation structure affects double descent. From Equations (4), (5) one has in the

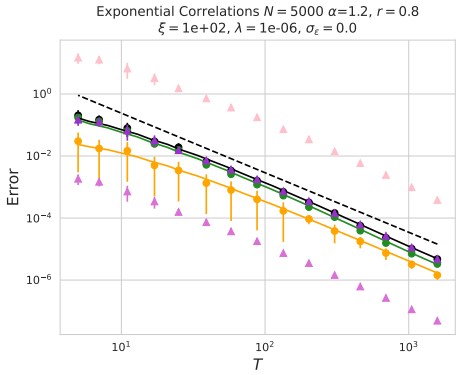

(a) Exponential Correlations

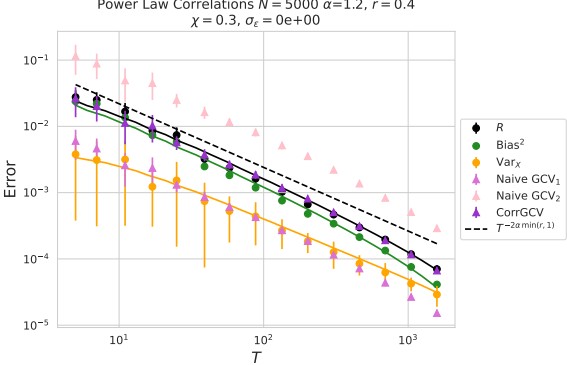

(b) Power Law Correlations

*Figure 3.* Power law scalings for data with a) exponential correlations with $\xi = 10^2$ and b) power law correlations $\mathbb{E}\boldsymbol{x}_t^\top \boldsymbol{x}_{t+\tau} \propto \tau^{-\chi}$ with $\chi = 0.3$. In both cases, the correlations of the data do not affect the scaling of the generalization error as a function of $T$, which generally goes as $T^{-2\alpha \min(r,1)}$, as derived in prior works. Although other estimators correctly predict the rate of decay, only the CorrGCV correctly recovers the exact risk.

structured case that:
$$\kappa = \frac{\lambda}{1 - q\mathrm{df}_1} S_{\boldsymbol{K}}(q\mathrm{df}_1);$$
compare with (7) in the uncorrelated setting. In the ridgeless limit in the overparameterized regime $q > 1$, $\kappa$ will be such that $q\mathrm{df}_{\boldsymbol{\Sigma}}^1(\kappa) = 1$, which is unchanged from the uncorrelated setting (Appendix F). The double descent behavior in that case is unchanged by correlations. At finite ridge, we see that training on correlated data is similar to replacing $\lambda \to \lambda S_{\boldsymbol{K}}(q\mathrm{df}_1)$ with uncorrelated data. Near the double descent peak, this expands to leading order in $q - 1$ as $\lambda S_{\boldsymbol{K}}(1)$. In Appendix F, we show that $S_{\boldsymbol{K}} \geq 1$ pointwise. Thus, correlations will enhance the ridge near the interpolation threshold, meaning that the double-descent peak should be mollified. For strong exponential correlations $K_{ts} = e^{-|t-s|/\xi}$, $\lambda S_{\boldsymbol{K}}(1)$ is approximately $\lambda\xi$. This leads to the less sharp double descent peak in Figure 4(b).

Thus, correlations mollify the double descent phenomenon. We do not say that they regularize the effect, because the risk still explodes in the ridgeless $q \to 1$ limit.

Noise correlations affect the generalization error only if the covariates are also correlated. If the covariates are uncorrelated, namely $\boldsymbol{K} = \mathbf{I}_T$ then $\boldsymbol{K}'$ enters the risk (8) only through $\frac{1}{T}\operatorname{Tr}\boldsymbol{K}'$, which is 1 by definition. Conversely, we can consider the case where the covariates are correlated but the noise is uncorrelated, *i.e.*, $\boldsymbol{K}' = \mathbf{I}_T$. In the overparameterized regime, in the ridgeless limit this yields an error equal to that with uncorrelated data multiplied by $\frac{1}{T}\operatorname{Tr}(\boldsymbol{K}^{-1})$, which is greater than one so long as $\boldsymbol{K} \neq \mathbf{I}_T$ (Appendix F). This implies that having uncorrelated noise is generally worse than having noise with matched correlations. We illustrate this behavior in Figure 4(c), where this effect is visible as a strong magnification of double-descent.

## 4. Algorithmic implementation

In this section, we give an algorithm to compute the CorrGCV given knowledge of the function $S_{\boldsymbol{K}}(q\mathrm{df}_1)$. We discuss how to estimate $S_{\boldsymbol{K}}$ from data in Appendix A.2. The diagram for this calculation is given in Figure 5. In what follows, we will use the notation $b \leftarrow a$ to indicate variable assignment of $b$ given that $a$ has been computed. We also use $a \equiv b$ to highlight that the variable $a$ is shorthand for $b$. This allows one to easily track the causal chain of how the relevant variables are estimated.

First, given $\hat{\boldsymbol{\Sigma}}$ alone, we can calculate $\kappa$ as long as the functional form of $S_{\boldsymbol{K}}$ is known. We obtain:
$$\mathrm{df}_1 \leftarrow \mathrm{df}_{\hat{\boldsymbol{\Sigma}}}^1(\lambda), \quad \kappa \leftarrow \lambda \frac{S_{\boldsymbol{K}}(q\mathrm{df}_1)}{1 - q\mathrm{df}_1}.$$
At this point, the ordinary GCV is directly given by $S^2 \hat{R}_{in}$. Given a functional form of $S_{\boldsymbol{K}}$, this gives $\kappa$ as a differentiable program of $\lambda$. Derivatives can be efficiently evaluated with autograd, for example using the JAX library (Bradbury et al., 2018). This will be important for the next steps.

Second, by applying the duality relation (see Appendix CC), we obtain estimates for $\tilde{\mathrm{df}}_1 \equiv \mathrm{df}_{\boldsymbol{K}}^1(\lambda)$ and $\tilde{\kappa}$:
$$\tilde{\mathrm{df}}_1 \leftarrow q\mathrm{df}_1, \quad \tilde{\kappa} \leftarrow \frac{\lambda}{\kappa q\mathrm{df}_1}.$$
Third, we leverage autograd to estimate $\mathrm{df}_2 \equiv \mathrm{df}_{\boldsymbol{\Sigma}}^2$ by applying another duality relationship:
$$\mathrm{df}_2 \leftarrow \mathrm{df}_1 + \frac{\partial_\lambda \mathrm{df}_1}{\partial_\lambda \log \kappa}.$$
Fourth, by using this and applying autograd again, we obtain an estimate for $\tilde{\mathrm{df}}_2 \equiv \mathrm{df}_{\boldsymbol{K}}^2$:
$$\tilde{\mathrm{df}}_2 \leftarrow \tilde{\mathrm{df}}_1 - q\frac{\partial_\lambda \log \kappa}{\partial_\lambda \log \tilde{\kappa}}(\mathrm{df}_1 - \mathrm{df}_2).$$

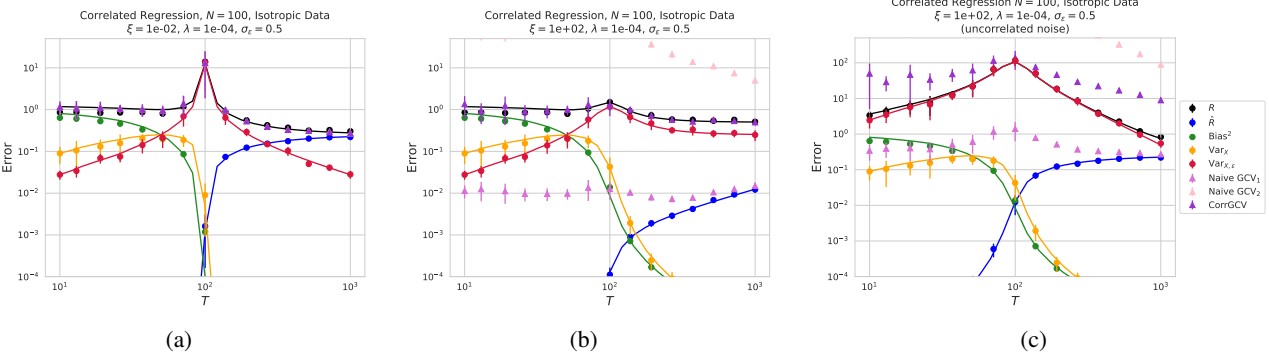

*Figure 4.* Precise asymptotics for double-descent in linear regression with unstructured data across various correlations. We choose an exponential correlation with correlation length $\xi$ and vary $\xi$. a) Weakly correlated data and noise, giving rise to the traditional double descent curve as analyzed in (Advani et al., 2020; Hastie et al., 2022). All GCV-related estimators agree and correctly estimate the out-of-sample risk. b) Strongly correlated data with matched noise correlations. The double descent peak is *mollified*. c) Strongly correlated data but uncorrelated noise. The double descent peak is *exacerbated*. This mismatch in correlations violates the assumptions of the CorrGCV, and thus no GCV can asymptotically match it without knowledge of the noise level $\sigma_\epsilon$. Across all settings the theory curves (solid lines) find excellent agreement with the experiments (solid markers with error bars over 10 different datasets).

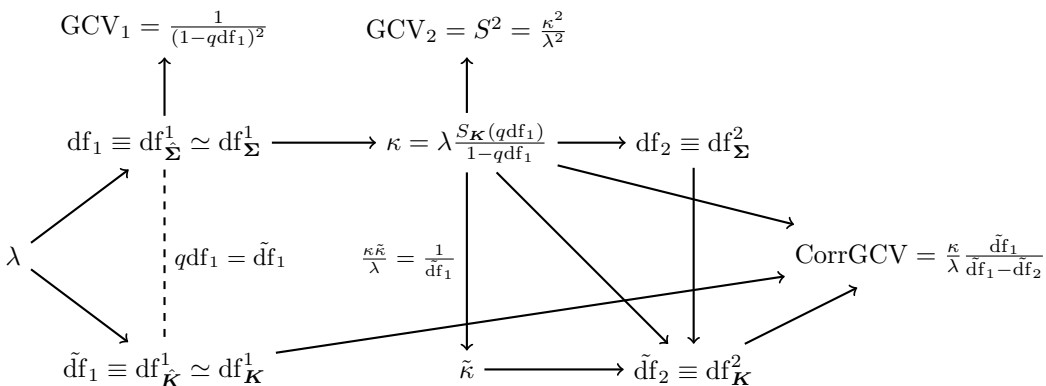

*Figure 5.* A graphical representation of the program needed to obtain the CorrGCV empirically from a given dataset. The asymmetry of the diagram arises from the fact that we estimate $\kappa$ first rather than $\tilde{\kappa}$. This is because it is more reasonable to assume a good estimate of the correlations $\boldsymbol{K}$, which often have properties such as stationarity that improve the estimation process, compared to estimating $\boldsymbol{\Sigma}$. As a result, it is easier to use either an exact form or a differentiable interpolation of $S_{\boldsymbol{K}}(\mathrm{df})$ rather than $S_{\boldsymbol{\Sigma}}$.

Finally, from this we obtain the CorrGCV estimator:

$$E_{CorrGCV} \leftarrow \hat{R}_{in} S \frac{\tilde{\mathrm{df}}_1}{\tilde{\mathrm{df}}_1 - \tilde{\mathrm{df}}_2}.$$

## 5. Testing on correlated data

Finally, we consider the setting where the data point $\boldsymbol{x}$ on which we test on has a nontrivial correlation $\boldsymbol{k}$ with each $\boldsymbol{x}_t$. That is, $\mathbb{E}[\boldsymbol{x}_t \cdot \boldsymbol{x}] \propto k_t$. For simplicity, we assume that the covariates and noise are identically correlated. Such a setting arises naturally in the case of forecasting time series, where one trains a model on a window of data and then aims to predict at a future time within the correlation time of the process. This correlation introduces a multiplicative

correction to the asymptotic risk:

**Theorem 5.1.** *Denote by $R_{out}^{\boldsymbol{k}}$ the out-of-sample risk when the test point has correlation $[\boldsymbol{k}]_t$ with data point $\boldsymbol{x}_t$. In this case, the test point $\boldsymbol{x}$ is conditionally Gaussian with*

$$\mathbb{E}[\boldsymbol{x}|\boldsymbol{X}] = \boldsymbol{X}^\top \boldsymbol{\alpha}, \quad \mathrm{Var}[\boldsymbol{x}|\boldsymbol{X}] = (1-\rho)\boldsymbol{\Sigma},$$

*where $\boldsymbol{\alpha} = \boldsymbol{K}^{-1}\boldsymbol{k}$ and $\rho = \boldsymbol{k}^\top \boldsymbol{K}^{-1}\boldsymbol{k}$. Assume that $\boldsymbol{\epsilon}_t$ has the same correlation as $\boldsymbol{x}_t$, namely $\boldsymbol{K}$. Then,*

$$R_{out}^{\boldsymbol{k}} \simeq R_{out}^{\boldsymbol{k}=0} \left[1 - \rho + \tilde{\kappa}^2 \boldsymbol{\alpha}^\top \boldsymbol{K}(\boldsymbol{K} + \tilde{\kappa})^{-2}\boldsymbol{\alpha}\right].$$

*Proof.* See Appendix C.5. □

We note that because $\tilde{\kappa} > 0$, the spectrum of $\tilde{\kappa}^2(\boldsymbol{K} + \tilde{\kappa})^{-2}$ is strictly bounded between zero and one. Thus, the last term in brackets is strictly bounded from above by $\rho$, meaning that

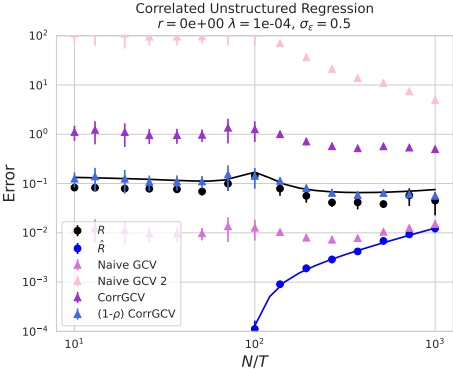

(a) Isotropic, Correlated Test

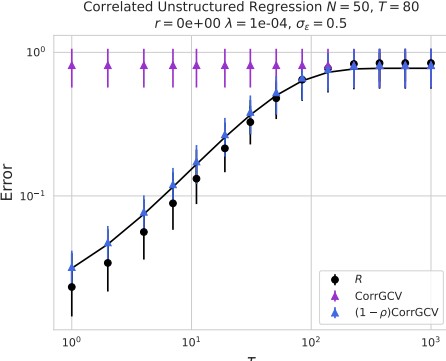

(b) Isotropic, Correlated Test

*Figure 6.* Strongly exponentially correlated data as in Figure 1 b), with $\mathbb{E}\boldsymbol{x}_t\boldsymbol{x}_{t+\tau} \sim e^{-\tau/\xi}$, $\xi = 10^2$. a) Comparison of the out-of-sample risk for testing on a correlated point that is $\tau = 5$ time points in the future from the training data. We emphasize that the CorrGCV here is the CorrGCV for testing on an uncorrelated test point, which is pessimistic relative to the risk for the correlated point at $\tau = 5$. b) Plot of the out-of-sample risk as a function of the horizon time that we test out. We see that testing on closer times is predictably more optimistic. We see that $(1 - \rho)$ times the CorrGCV provides a good approximation of the optimism of testing on a correlated point. Given a knowledge of the correlation structure of $\boldsymbol{K}$, this can be efficiently computed for stationary time series data.

$R_{out}^{\boldsymbol{k}}$ is always less than $R_{out}^{\boldsymbol{k}=0}$. This precisely quantifies the over-optimism of cross-validation when the held-out samples are nearby in time.

The train error is not affected by the correlated test point. Consequently, this yields an extension of the CorrGCV:

$$R_{out} \simeq \frac{S\tilde{\mathrm{df}}_1}{\tilde{\mathrm{df}}_1 - \tilde{\mathrm{df}}_2} \left[1 - \rho + \tilde{\kappa}^2 \boldsymbol{\alpha}^\top \boldsymbol{K}(\boldsymbol{K} + \tilde{\kappa})^{-2} \boldsymbol{\alpha}\right] \hat{R}_{in}.$$

In practice, we find that the simple approximation $R_{out}^{\boldsymbol{k}} \approx (1 - \rho)R_{out}^{\boldsymbol{k}=0}$ works very well (Figure 6).

## 6. Conclusion

In this paper, we have provided a comprehensive characterization of the asymptotic risk of high-dimensional ridge regression with correlated Gaussian datapoints. We expect our asymptotics to extend to non-Gaussian data thanks to the universality of ridge regression (Hu & Lu, 2022; Montanari & Saeed, 2022; Misiakiewicz & Saeed, 2024). Our results show that previously-proposed extensions of the GCV estimator to non-i.i.d. data are asymptotically biased, and immediately give the formula for the corrected GCV. This correction factor requires one to estimate more fine-grained spectral statistics of the sample-sample covariance than the usual GCV. However, as we have shown in the figures and discussed in Appendix A, obtaining an excellent estimate appears relatively straightforward.

Though our results are quite general, they are not without limitations, and there are several opportunities for further theoretical inquiry. First, our matrix-Gaussian model for the training data (1) could be relaxed to allow for sample-dependent covariance between features. On the technical side, to our knowledge Gaussian universality of ridge regression for non-independent data has not been rigorously established, though we expect it to hold (Hu & Lu, 2022; Montanari & Saeed, 2022; Misiakiewicz & Saeed, 2024). One might also want to establish dimension-free deterministic equivalents with relative error bounds, as in Misiakiewicz & Saeed (2024). However, these are largely technical, rather than conceptual, limitations.

After the completion of this work, we became aware of the contemporaneous work of Luo et al. (2024), who derived an asymptotically unbiased non-omniscient risk estimator under the assumption of right rotation-invariance. Concretely, they assume that the design matrix $\boldsymbol{X}$ satisfies $\boldsymbol{X}\boldsymbol{O} \overset{d}{=} \boldsymbol{X}$ for any orthogonal matrix $\boldsymbol{O}$. Their results thus allow for non-Gaussian data, but do not allow for correlations between features, and are therefore complementary to our own.

We conclude by commenting briefly on applications of our results. First and foremost, we anticipate that our asymptotically exact correction to the GCV should be of some utility in timeseries regression settings, whether in finance (Bouchaud & Potters, 2003), neuroscience (Williams & Linderman, 2021), or elsewhere. Moreover, our results might be of use in the study of in-context learning in language models, where most theoretical investigations neglect for the sake of analytical convenience the rich correlations present in language (Lu et al., 2024).

## Author contributions

A.A. and J.A.Z.-V. conceived the project, performed research, and wrote the manuscript. C.P. supervised the project and edited the manuscript.

## Acknowledgements

AA is grateful to Holden Leslie-Bole and Jacob Prince for useful conversations. We also thank Benjamin Ruben for helpful comments on a previous version of this manuscript. JAZV was supported by the Office of the Director of the National Institutes of Health under Award Number DP5OD037354. The content is solely the responsibility of the authors and does not necessarily represent the official views of the National Institutes of Health. JAZV is further supported by a Junior Fellowship from the Harvard Society of Fellows. C.P. is supported by NSF grant DMS-2134157, NSF CAREER Award IIS-2239780, DARPA grant DIAL-FP-038, a Sloan Research Fellowship, and The William F. Milton Fund from Harvard University. This work has been made possible in part by a gift from the Chan Zuckerberg Initiative Foundation to establish the Kempner Institute for the Study of Natural and Artificial Intelligence.

## Impact statement

This paper presents work whose goal is to advance the field of Machine Learning. There are many potential societal consequences of our work, none which we feel must be specifically highlighted here.

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

```python
1  # R_in: In-sample risk, AKA training Error
2  # lamb: Value of the ridge
3  # q: Ratio N/T
4  # eigs_Sh: Eigenvalues of empirical covariance Sigma hat (Sh)
5  # S_K: Function of df given by the S-transform S_K
6  from jax import grad
7  import jax.numpy as jnp
8
9  df1_fn_Sh = lambda l: jnp.mean(eigs_Sh/(l+eigs_Sh))
10 d_df1_fn_Sh = grad(df1_fn_Sh)
11
12 df1_Sh_est = df1_fn_Sh(lamb)
13 df1_Kh_est = q * df1_Sh_est
14
15 R_GCV1 =  R_in/(1-q*df1_est)**2
16
17 S_est = 1/(1-q*df1_est) * S_K(q*df1_est)
18 R_GCV2 =  R_in * S_est**2
19
20 kappa1_fn = lambda l: l/(1-q*df1_fn_Sh(l)) * S_K(q*df1_fn_Sh(l))
21 kappa2_fn = lambda l: l/(q * df1_fn_Sh(l) * k1_fn(l))
22 d_log_kappa1_fn = grad(lambda l: jnp.log(kappa1_fn(l)))
23 d_log_kappa2_fn = grad(lambda l: jnp.log(kappa2_fn(l)))
24 df2_fn_Sh = lambda l: df1_fn_Sh(l) + d_log_kappa1_fn(l) * d_df1_fn_Sh(l)
25
26 df2_Kh_est = q*df1_est - q*(df1_fn_Sh(lamb) - df2_fn_Sh(lamb))*d_log_k1_fn(lamb)/
       d_log_k2_fn(lamb)
27
28 R_CorrGCV = R_in * S_est * df1_Kh_est/(df1_Kh_est - df2_Kh_est)
```

Code Block 1: Python code for computing CorrGCV estimator given relevant parameters.

## A. Algorithmic implementation of the CorrGCV

### A.1. Code for implementation

In Code Block 1 we give the code for implementing the CorrGCV estimator in JAX given a ridge $\lambda$, training error $\hat{R}_{in}$, overaparameterization ratio $q$, empirical covariance matrix $\hat{\boldsymbol{\Sigma}}$, and function $S_{\boldsymbol{K}}(\mathrm{df})$ given by the $S$-transform of the correlation structure.

### A.2. Estimating the S-transform from data

If the functional form of the correlations is not exactly known, there are several options.

1. From empirical data, one can fit $\hat{\boldsymbol{K}}$ to a parametric form for which the $S$-transform $S_{\boldsymbol{K}}$ can be calculated exactly. We give some examples of this in Appendix I; a practically-relevant example is an exponential $e^{-|t-t'|/\xi}$ (Potters & Bouchaud, 2020).

2. For a time series with the assumption of stationarity, one can estimate the autocorrelation $A_\tau = K_{t,t+\tau}$ as a function of lag $\tau$ much more reliably than the full matrix $K_{t,t'}$, as it depends on only $\mathcal{O}(T)$ parameters rather than $\mathcal{O}(T^2)$.

3. From $\hat{\boldsymbol{K}}$, one can calculate $\mathrm{df}_{\hat{\boldsymbol{K}}}(\lambda)$ as a function of $\lambda$. By making use of a solver (such as a bisection method), one can evaluate $\mathrm{df}_{\hat{\boldsymbol{K}}}^{-1}(\mathrm{df})$ and then get

$$S_{\hat{\boldsymbol{K}}}(\tilde{\mathrm{df}}) = \frac{1-\tilde{\mathrm{df}}}{\tilde{\mathrm{df}}\,\mathrm{df}_{\hat{\boldsymbol{K}}}^{-1}(\tilde{\mathrm{df}})} = \frac{S_{\boldsymbol{K}}(\tilde{\mathrm{df}})}{q-\tilde{\mathrm{df}}} \Rightarrow S_{\boldsymbol{K}}(\tilde{\mathrm{df}}) = \frac{(q-\tilde{\mathrm{df}})(1-\tilde{\mathrm{df}})}{\tilde{\mathrm{df}}\,\mathrm{df}_{\hat{\boldsymbol{K}}}^{-1}(\tilde{\mathrm{df}})}. \tag{9}$$

We can then evaluate $\kappa_1$ as follows:

- First, sample a fine grid of $\lambda$.
- From the data, we estimate $\mathrm{df}_1 = \mathrm{df}_{\hat{\boldsymbol{\Sigma}}}^1(\lambda)$ for each lambda, leading to a fine grid of $\mathrm{df}_1$ and $\tilde{\mathrm{df}}_1 = q\mathrm{df}_1$

- Given the bisection-based algorithm to evaluate $\mathrm{df}_{\hat{K}}^{-1}$. We can then evaluate $S_K(\tilde{\mathrm{df}})$ using (9). This leads to an exact formula for $S_K$ on the corresponding fine grid of $\tilde{\mathrm{df}}_1 = q\mathrm{df}_1$.
- Using the definition $\kappa = \frac{S_K(q\mathrm{df}_1)}{1 - q\mathrm{df}_1}$, we obtain a fine grid of $\kappa$ as a function of $\lambda$.
- We now apply an appropriate interpolation method that we can differentiate through and guarantee that as the grid gets finer, both the function and its derivatives converge to the true $\kappa$ and $\partial_\lambda \kappa$.

The last bullet point requires further discussion. In practice, because $\kappa$ can vary over several orders of magnitude, and because estimating the CorrGCV requires logarithmic derivatives, it is often best to interpolate $\log \kappa$ against $\log \lambda$. A good interpolator compatible with autograd can be a Gaussian process with appropriately chosen covariance, a piece-wise smooth spline function, polynomial interpolator, or a wavelet based method that converges in the appropriate Sobolev norm. We used a simple degree 5 polynomial interpolation of $S_K$ to generate the power law correlation learning curves in Figure 2(b).

## B. Review of free probability

### B.1. Definition of Freedom

We now define what it means for a set of $n$ random matrices $\{A_i\}_{i=1}^n$ to be jointly (asymptotically) free. All of these matrices are $N \times N$ and we consider the limit of $N \to \infty$. This is the limit in which free probability theory applies for random matrices. There are several texts on this rich subject. See, for example (Mingo & Speicher, 2017; Voiculescu, 1997; Nica & Speicher, 2006).

We say that a polynomial $p(A_i)$ has the mean zero property if

$$\frac{1}{N} \mathrm{Tr}[p(A_i)] \simeq 0.$$

Above, $\simeq 0$ means that this converges to zero in probability as $N \to \infty$.

Take a set of $m$ polynomials $\{p_k\}_{k=1}^m$, each with the mean zero property. Further, take a labeling $\{i_k\}_{k=1}^n$ with each $i_k \in \{1, \ldots, n\}$ so that $i_k \neq i_{k+1}$ for all $k$. The $\{A_i\}_{i=1}^n$ are jointly asymptotically free if and only if

$$\frac{1}{N} \mathrm{Tr}\left[p_1(A_{i_1}) \cdots p_m(A_{i_m})\right] \simeq 0$$

for any $m$, any labeling $\{i_k\}$, and any set of polynomials $\{p_k\}_{k=1}^m$ satisfying the properties above.

For our purposes we only require that the $\frac{1}{T} Z^\top Z$ and $\frac{1}{T} Z Z^\top$ matrices are free of any fixed deterministic matrix, specifically $\Sigma$ and $K$.

### B.2. The R and S transforms

We review here basic properties of the $R$ and $S$ transforms of free probability (Voiculescu et al., 1992; Voiculescu, 1997). The recent book by Potters and Bouchaud (Potters & Bouchaud, 2020) provides an accessible introduction to these techniques in the context of random matrix theory.

Given a $N \times N$ symmetric random matrix $A$, one considers the **resolvent**, also known as the **Stiltjes Transform**:

$$g_A(\lambda) \equiv \frac{1}{N} \mathrm{Tr}\left[(A + \lambda)^{-1}\right].$$

The eigenvalue density of $A$, $\rho_A$, can be obtained from the **inverse Stiltjes transform**:

$$\rho_A(\lambda) = \lim_{\epsilon \to 0} \frac{1}{\pi} \mathrm{Im}\, g_A(-\lambda + i\epsilon).$$

Similarly, one defines the degrees of freedom $\mathrm{df}_A^1(\lambda)$ as

$$\mathrm{df}_A^1(\lambda) = \frac{1}{N} \mathrm{Tr}[A(A + \lambda)^{-1}], \quad \mathrm{df}_A^2(\lambda) = \frac{1}{N} \mathrm{Tr}[A^2(A + \lambda)^{-2}].$$

One has the relationship:

$$g_A(\lambda) = \frac{1 - \mathrm{df}_A^1(\lambda)}{\lambda}.$$

We let $g_A^{-1}, \mathrm{df}_A^{-1}$ denote the respective functional inverses. The $R$ and $S$ transforms are respectively functions of formal

variables $g, \mathrm{df}$ given by:

$$R_{\boldsymbol{A}}(g) \equiv g_{\boldsymbol{A}}^{-1}(g) - \frac{1}{g}, \quad S_{\boldsymbol{A}}(\mathrm{df}) \equiv \frac{1 - \mathrm{df}}{\mathrm{df}\,\mathrm{df}_{\boldsymbol{A}}^{-1}(\mathrm{df})}. \tag{10}$$

They have the property that for any two matrices $\boldsymbol{A}, \boldsymbol{B}$ that are free of one another:

$$R_{\boldsymbol{A}+\boldsymbol{B}}(g) = R_{\boldsymbol{A}}(g) + R_{\boldsymbol{B}}(g), \quad S_{\boldsymbol{A}*\boldsymbol{B}}(\mathrm{df}) = S_{\boldsymbol{A}}(\mathrm{df})S_{\boldsymbol{B}}(\mathrm{df}).$$

From these two respective properties, together with the definitions (10) we obtain the **subordination relations**, also known as (weak) deterministic equivalence:

$$g_{\boldsymbol{A}+\boldsymbol{B}}(\lambda) = g_{\boldsymbol{A}}(\lambda + R_{\boldsymbol{B}}), \quad \mathrm{df}_{\boldsymbol{A}*\boldsymbol{B}}(\lambda) = \mathrm{df}_{\boldsymbol{A}}(\lambda S_{\boldsymbol{B}}). \tag{11}$$

Here, the additive and multiplicative renormalizations of $\lambda$, given by $R_{\boldsymbol{B}}$ and $S_{\boldsymbol{B}}$ respectively, can each be evaluated in two different ways. The first way is from the original noisy matrices $\boldsymbol{A} + \boldsymbol{B}$ and $\boldsymbol{A} * \boldsymbol{B}$. This is the analogue of the empirical estimate of $S, \kappa$ discussed in the text.

$$R_{\boldsymbol{B}} \equiv R_{\boldsymbol{B}}(g_{\boldsymbol{A}+\boldsymbol{B}}(\lambda)), \quad S_{\boldsymbol{B}} \equiv S_{\boldsymbol{B}}(\mathrm{df}_{\boldsymbol{A}*\boldsymbol{B}}^{1}(\lambda)).$$

The second way is from the clean matrix $\boldsymbol{A}$ itself. This is what gives the omniscient estimate of the renormalized ridges. Writing $\kappa_+, \kappa_*$ for $\lambda + R_{\boldsymbol{B}}, \lambda S_{\boldsymbol{B}}$ respectively, we obtain the self-consistent equations:

$$\kappa_+ = \lambda + R_{\boldsymbol{B}}(g_{\boldsymbol{A}}(\kappa_+)), \quad \kappa_* = \lambda S_{\boldsymbol{B}}(\mathrm{df}_{\boldsymbol{A}}^{1}(\kappa_*)).$$

### B.3. Strong deterministic equivalence

The deterministic equivalences in (11) extend to the matrices themselves. Taking $\boldsymbol{A}$ deterministic and $\boldsymbol{B}$ random and free of $\boldsymbol{A}$, we have:

$$(\boldsymbol{A} + \boldsymbol{B} + \lambda)^{-1} \simeq (\boldsymbol{A} + \kappa_+)^{-1}, \quad \boldsymbol{A} * \boldsymbol{B}(\boldsymbol{A} * \boldsymbol{B} + \lambda)^{-1} \simeq \boldsymbol{A}(\boldsymbol{A} + \kappa_*)^{-1}.$$

Here, for two matrices, we use the relation $\simeq$ to denote that the traces of these quantities against any test matrix of bounded spectral norm will converge in probability to the same quantity as $N \to \infty$.

The above two formulas can be derived using replica theory (Bun et al., 2016b; Potters & Bouchaud, 2020), from diagrammatics (Burda et al., 2011; Atanasov et al., 2024), or from cavity arguments (Bach, 2024).

In this paper, we only require the properties of the $S$-transform. The above deterministic equivalences are called "one point" equivalences, as they only involve a single matrix inverse. We will derive two-point equivalents in the sequel.

## C. Deferred proofs of main results

### C.1. Warm-up: 1-point deterministic equivalents

In this section, we prove Lemma 2.2 using a diagrammatic argument based on Atanasov et al. (2024), specialized to the specific case of Wishart noise matrices $\boldsymbol{B}$. The argument is diagrammatic, appealing to the fact that at large $N$, free random matrices obey non-crossing properties in their diagrammatics ('t Hooft, 1973). The argument holds rigorously only in the $N \to \infty$ limit. However, it can rigorously be shown to be the leading term in an asymptotic series in $1/N$ by leveraging known properties of the higher-order corrections (Weingarten, 1978).

In what follows, for the sake of notational levity we absorb a factor of $1/\sqrt{T}$ into each insertion of $\boldsymbol{Z}$ or $\boldsymbol{Z}^\top$. Because each pair of $\boldsymbol{Z}, \boldsymbol{Z}^\top$ insertions will necessarily be averaged over via Wick contraction, this is the same as associating an factor of $1/T$ with each Wick contraction.

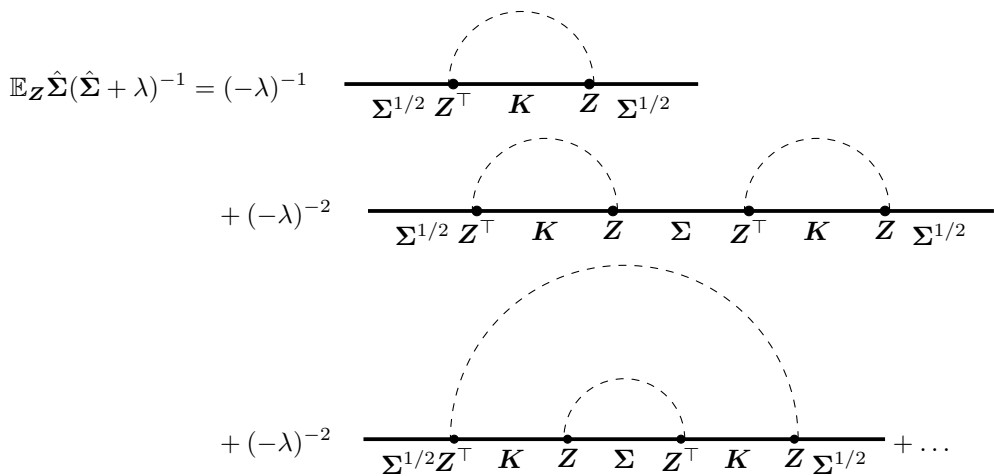

$$\mathbb{E}_{\boldsymbol{Z}} \hat{\boldsymbol{\Sigma}} (\hat{\boldsymbol{\Sigma}} + \lambda)^{-1} = (-\lambda)^{-1}$$

$$+ (-\lambda)^{-2}$$

$$+ (-\lambda)^{-2} \qquad + \dots$$

Such a suppression of crossing diagrams is a defining characteristic of free probability. Because each loop contributes a trace, while each insertion of $\boldsymbol{Z}^\top \boldsymbol{Z}$ contributes a factor of $1/T$, in order for a diagram to give an order 1 contribution we will need to have as many loops as there are pairs of $\boldsymbol{Z}^\top, \boldsymbol{Z}$. Consequently, crossing diagrams such as the following are suppressed as $N, T \to \infty$:

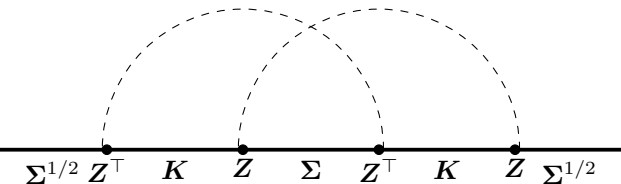

Because crossing diagrams do not contribute, one can observe the following pattern. Any diagram that appears will be a link of averages from one $\boldsymbol{Z}^\top$ to some later $\boldsymbol{Z}$ that creates an arc. Beneath that arc, all averages can only be between the matrices within the arc, by non-crossing. As such, we can expand:

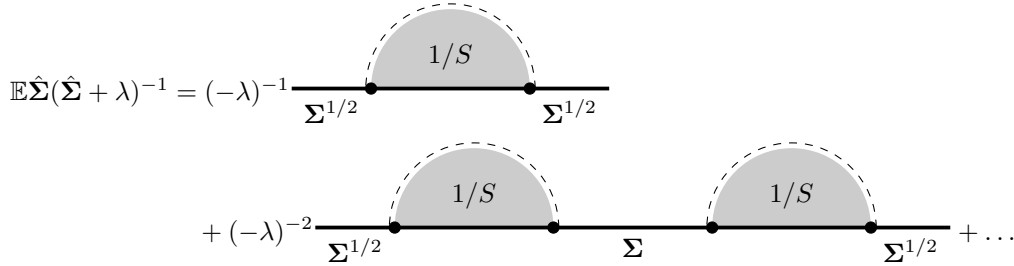

The terms in the dashed lines are pre-emptively denoted by $1/S$. We make two observations.

- Because the isotropic matrix $\boldsymbol{Z}$ is right-invariant to rotations in $\mathbb{R}^N$, $1/S$ must be a deterministic matrix that is invariant under rotation. The only such matrices are constants times the identity.

- Because the isotropic matrix $\boldsymbol{Z}$ is also left-invariant to rotations in $\mathbb{R}^T$, $1/S$ is a rotationally invariant scalar functional of the product of matrices beneath the arc. The only such rotationally invariant scalar functional is any constant multiple of the trace.

The fact that the shaded parts are scalars immediately implies we can resum this as:

$$\mathbb{E}_{\boldsymbol{Z}} \hat{\boldsymbol{\Sigma}} (\hat{\boldsymbol{\Sigma}} + \lambda)^{-1} = \sum_{n=1}^{\infty} \boldsymbol{\Sigma}^n \frac{1}{(-\lambda S)^n} = \boldsymbol{\Sigma} (\boldsymbol{\Sigma} + \lambda S)^{-1}.$$

An immediate consequence is:

$$\mathbb{E}_{\boldsymbol{Z}}(\hat{\boldsymbol{\Sigma}} + \lambda)^{-1} = \frac{1}{\lambda}\left[1 - \mathbb{E}_{\boldsymbol{Z}}\hat{\boldsymbol{\Sigma}}(\hat{\boldsymbol{\Sigma}} + \lambda)^{-1}\right] = S(\boldsymbol{\Sigma} + \lambda S)^{-1}. \tag{12}$$

It now remains to evaluate $S$. We have

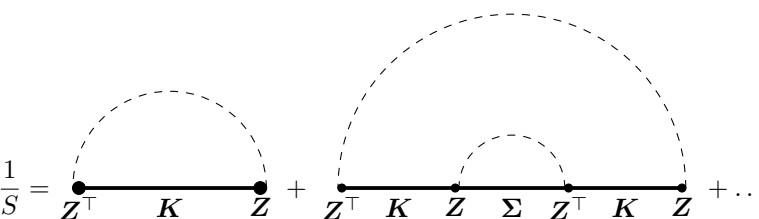

We immediately recognize this as the trace of $\lambda \boldsymbol{K}(\hat{\boldsymbol{K}} + \lambda)^{-1}$. Because $\boldsymbol{Z}$ has entries going as $1/\sqrt{T}$ this implies:

$$\frac{1}{S} = \mathbb{E}_{\boldsymbol{Z}}\,\lambda\frac{1}{T}\mathrm{Tr}\left[\boldsymbol{K}(\hat{\boldsymbol{K}} + \lambda)^{-1}\right]. \tag{13}$$

Here, because the trace is self-averaging, we could drop the $\mathbb{E}_{\boldsymbol{Z}}$ and still keep this as an equality in probability as $T, N \to \infty$.

By a slight adjustment of the above diagrammatic argument, swapping the role of $\boldsymbol{\Sigma}$ and $\boldsymbol{K}$, we have an analogue of (12):

$$\mathbb{E}_{\boldsymbol{Z}}(\hat{\boldsymbol{K}} + \lambda)^{-1} = \tilde{S}(\boldsymbol{K} + \lambda\tilde{S})^{-1}, \quad \frac{1}{\tilde{S}} = \mathbb{E}_{\boldsymbol{Z}}\,\lambda\frac{1}{T}\mathrm{Tr}\left[\boldsymbol{\Sigma}(\hat{\boldsymbol{\Sigma}} + \lambda)^{-1}\right]. \tag{14}$$

We now define

$$\kappa \equiv \lambda S, \quad \tilde{\kappa} \equiv \lambda\tilde{S}, \quad \mathrm{df}_1 \equiv \frac{1}{N}\mathrm{Tr}\left[\boldsymbol{\Sigma}(\boldsymbol{\Sigma} + \kappa)^{-1}\right], \quad \tilde{\mathrm{df}}_1 \equiv \frac{1}{T}\mathrm{Tr}\left[\boldsymbol{K}(\boldsymbol{K} + \tilde{\kappa})^{-1}\right].$$

Note that at this stage we have shown the equivalence:

$$\tilde{\mathrm{df}}_1 \equiv \mathrm{df}_{\boldsymbol{K}}^1(\lambda\tilde{S}) \simeq \mathrm{df}_{\hat{\boldsymbol{K}}}^1(\lambda) = q\,\mathrm{df}_{\hat{\boldsymbol{\Sigma}}}^1(\lambda) \simeq q\,\mathrm{df}_{\boldsymbol{\Sigma}}^1(\lambda S) \equiv q\,\mathrm{df}_1.$$

Putting (14) back into (13) yields:

$$\boxed{\frac{\lambda}{\kappa\tilde{\kappa}} = \tilde{\mathrm{df}}_1.}$$

We recognize as the **duality relation**. We can also get an explicit expression for $S$ in terms of $S_K$ and df by defining the formal function of df

$$S_{\boldsymbol{A}}(\mathrm{df}) \equiv \frac{1 - \mathrm{df}}{\mathrm{df}\,\mathrm{df}_{\boldsymbol{A}}^{-1}(\mathrm{df})}$$

and plugging in $\boldsymbol{A} = \boldsymbol{K}$, $\mathrm{df} = \mathrm{df}_{\boldsymbol{K}}^1(\tilde{\kappa}) = \tilde{\mathrm{df}}_1$ into this:

$$\tilde{\kappa}\tilde{\mathrm{df}}_1 = \frac{1 - \mathrm{df}_1}{S_{\boldsymbol{K}}(\tilde{\mathrm{df}}_1)}.$$

Using $\tilde{\mathrm{df}}_1 = q\,\mathrm{df}_1$ this yields the desired relationship:

$$\kappa = \lambda\frac{S_{\boldsymbol{K}}(q\,\mathrm{df})}{1 - q\,\mathrm{df}}.$$

Thus, given knowledge of $\lambda, \mathrm{df}_1$ and $S_{\boldsymbol{K}}$, one can calculate $\kappa$ exactly. For a discussion of how to obtain $S_{\boldsymbol{K}}$ from a given correlated dataset, see A.2.

## C.2. 2-point deterministic equivalents

In this section, we extend this diagrammatic technique to calculate the necessary 2-point deterministic equivalents stated in Lemmas 2.3 and 2.4. Some of these were derived using leave-one-out (cavity) arguments in recent papers of Bach (Bach, 2024) and (Patil & LeJeune, 2024) in the case of $\boldsymbol{K} = \mathbf{I}$. The remaining results that we derive are to our knowledge novel. As we discuss in Atanasov et al. (2025), they have useful applications to high-dimensional regression problems beyond just those studied in the context of this paper.

As in the prior section, we absorb a factor of $1/\sqrt{T}$ into each $\boldsymbol{Z}$ or $\boldsymbol{Z}^\top$ insertion, or equivalently associate a factor of $1/T$ with each Wick contraction.

We first prove Lemma 2.3, which concerns the evaluation of
$$(\hat{\boldsymbol{\Sigma}} + \lambda)^{-1} \boldsymbol{\Sigma}' (\hat{\boldsymbol{\Sigma}} + \lambda)^{-1}.$$
for arbitrary matrix $\boldsymbol{\Sigma}'$ between the resolvents. In performing this average, we note that the types of diagrams that appear split into two classes. The first are those with no arcs over $\boldsymbol{\Sigma}'$:

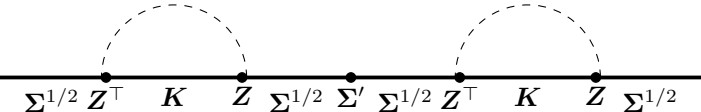

Because no arcs connect the resolvent on the left with the resolvent on the right, we can take the averages of the left and right resolvent separately, and obtain:

$$S^2 (\boldsymbol{\Sigma} + \lambda)^{-1} \boldsymbol{\Sigma}' (\boldsymbol{\Sigma} + \lambda)^{-1}.$$

The second class of term has arcs connecting the two resolvents on both sides. An example of such a diagram is:

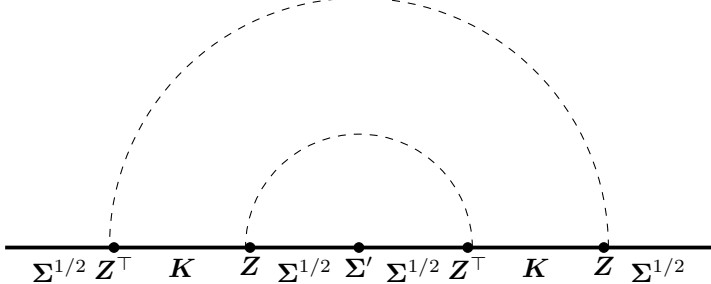

Any term in the second class will have some number of arcs over $\boldsymbol{\Sigma}'$. Note that this will necessarily be an even number $2n$, as they will alternate between averaging a $\boldsymbol{Z}^\top$ on the left with a $\boldsymbol{Z}$ on the right and vice versa. This will give $2n$ loops that are traced over. There will be $n$ loops involving $\boldsymbol{K}$ matrices, which we will call $\boldsymbol{K}$ loops, and $n$ loops involving $\boldsymbol{\Sigma}$ matrices which we will call $\boldsymbol{\Sigma}$ loops. The above term has $n = 1$. An example of a $\boldsymbol{\Sigma}$ loop is:

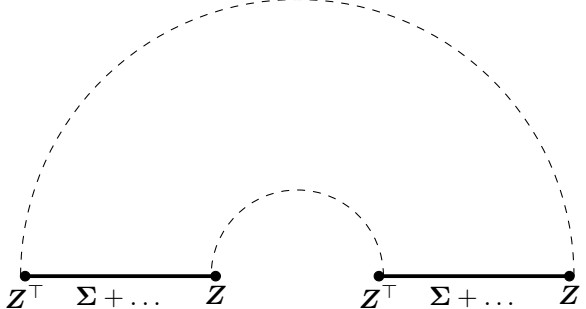

Here, on each side we denote by $\boldsymbol{\Sigma} + \ldots$ the series:
$$\boldsymbol{\Sigma} + (-\lambda)^{-1} \boldsymbol{\Sigma} \boldsymbol{Z}^\top \boldsymbol{Z} \boldsymbol{\Sigma} + (-\lambda)^{-2} \boldsymbol{\Sigma} \boldsymbol{Z}^\top \boldsymbol{Z} \boldsymbol{\Sigma} \boldsymbol{Z}^\top \boldsymbol{Z} \boldsymbol{\Sigma} + \cdots = \lambda \boldsymbol{\Sigma}^{1/2} (\hat{\boldsymbol{\Sigma}} + \lambda)^{-1} \boldsymbol{\Sigma}^{1/2}.$$
We recognize this as a resolvent. Because we have explicitly accounted for arcs connecting terms in the left resolvent with the terms on the right resolvent, within each loop we can treat the left and right resolvent average separately. This average is given in (12), and similarly for $\boldsymbol{K}$. This yields
$$\boldsymbol{\Sigma}\text{-loop} = \lambda^2 \frac{1}{T} \mathrm{Tr}\left[\boldsymbol{\Sigma}^{1/2} \mathbb{E}_{\boldsymbol{Z}}\left[(\hat{\boldsymbol{\Sigma}} + \lambda)^{-1}\right] \boldsymbol{\Sigma} \mathbb{E}_{\boldsymbol{Z}}\left[(\hat{\boldsymbol{\Sigma}} + \lambda)^{-1}\right] \boldsymbol{\Sigma}^{1/2}\right] = q\kappa^2 \mathrm{df}_2,$$
$$\boldsymbol{K}\text{-loop} = \lambda^2 \frac{1}{T} \mathrm{Tr}\left[\boldsymbol{K}^{1/2} \mathbb{E}_{\boldsymbol{Z}}\left[(\hat{\boldsymbol{K}} + \lambda)^{-1}\right] \boldsymbol{K} \mathbb{E}_{\boldsymbol{Z}}\left[(\hat{\boldsymbol{K}} + \lambda)^{-1}\right] \boldsymbol{K}^{1/2}\right] = \tilde{\kappa}^2 \tilde{\mathrm{df}}_2.$$
The innermost loop will have an insertion of $\boldsymbol{\Sigma}'$ in between the two resolvents, each of which is separately averaged, yielding:

$$\frac{S^2}{T} \mathrm{Tr}[\boldsymbol{\Sigma}^{1/2} (\boldsymbol{\Sigma} + \kappa)^{-1} \boldsymbol{\Sigma}' (\boldsymbol{\Sigma} + \kappa)^{-1} \boldsymbol{\Sigma}^{1/2}] \simeq S^2 \frac{1}{T} \mathrm{Tr}[\boldsymbol{\Sigma} \boldsymbol{\Sigma}' (\boldsymbol{\Sigma} + \kappa)^{-2}].$$

We adopt the shorthand

$$\mathrm{df}^2_{\boldsymbol{\Sigma},\boldsymbol{\Sigma}'} \equiv \frac{1}{N}\,\mathrm{Tr}\left[\boldsymbol{\Sigma}\boldsymbol{\Sigma}'(\boldsymbol{\Sigma}+\kappa)^{-2}\right],$$

$$\mathrm{df}^2_{\boldsymbol{K},\boldsymbol{K}'} \equiv \frac{1}{T}\,\mathrm{Tr}\left[\boldsymbol{K}\boldsymbol{K}'(\boldsymbol{K}+\kappa)^{-2}\right].$$

to denote this term by $qS^2\mathrm{df}^2_{\boldsymbol{\Sigma},\boldsymbol{\Sigma}'}$. This must necessarily be followed by a $\boldsymbol{K}$ loop. By making use of the duality relationship, we can write this joint contribution as:

$$\frac{\kappa^2}{\lambda^2}q\mathrm{df}^2_{\boldsymbol{\Sigma},\boldsymbol{\Sigma}'}\tilde{\kappa}^2\tilde{\mathrm{df}}_2 = \frac{\mathrm{df}^2_{\boldsymbol{\Sigma},\boldsymbol{\Sigma}'}\tilde{\mathrm{df}}_2}{\mathrm{df}_1\tilde{\mathrm{df}}_1} \equiv \gamma_{\boldsymbol{\Sigma},\boldsymbol{\Sigma}'}.$$

Between this innermost $\boldsymbol{\Sigma}'$ and $\boldsymbol{K}$ loop and the outside, there can be an arbitrary number of pairs of closed $\boldsymbol{K}$ and $\boldsymbol{\Sigma}$ loops in between. Again applying the duality relation, we see each pair contributes:

$$\frac{1}{\lambda^2}q\kappa^2\mathrm{df}_2\tilde{\kappa}^2\tilde{\mathrm{df}}_2 = \frac{\mathrm{df}_2\tilde{\mathrm{df}}_2}{\mathrm{df}_1\tilde{\mathrm{df}}_1} \equiv \gamma$$

Here we divide by $\lambda^2$ because each $\boldsymbol{Z}\boldsymbol{Z}^\top$ pair introduces a factor of $(-\lambda)^{-1}$. This gives an interpretation of $\gamma$ as the contribution of a pair of $\boldsymbol{\Sigma}$ and $\boldsymbol{K}$ loops. It remains to sum over $n$ to get the final contribution from all of the loops:

$$\gamma_{\boldsymbol{\Sigma},\boldsymbol{\Sigma}'}\sum_{n=0}^{\infty}\gamma^n = \frac{\gamma_{\boldsymbol{\Sigma},\boldsymbol{\Sigma}'}}{1-\gamma}.$$

Finally, outside of the loops, we can perform the average of the left resolvent and the right resolvent separately. This gives the desired relation:

$$(\hat{\boldsymbol{\Sigma}}+\lambda)^{-1}\boldsymbol{\Sigma}'(\hat{\boldsymbol{\Sigma}}+\lambda)^{-1} \simeq S^2(\boldsymbol{\Sigma}+\kappa)^{-1}\boldsymbol{\Sigma}'(\boldsymbol{\Sigma}+\kappa)^{-1} + S^2(\boldsymbol{\Sigma}+\kappa)^{-2}\boldsymbol{\Sigma}\frac{\gamma_{\boldsymbol{\Sigma},\boldsymbol{\Sigma}'}}{1-\gamma}. \tag{15}$$

When $\boldsymbol{K}=\mathbf{I}$, this recovers the earlier result of (Bach, 2024). By applying the same argument with minimal modifications, one obtains the deterministic equivalence for kernel resolvents, namely:

$$(\hat{\boldsymbol{K}}+\lambda)^{-1}\boldsymbol{K}'(\hat{\boldsymbol{\Sigma}}+\lambda)^{-1} \simeq \tilde{S}^2(\boldsymbol{K}+\tilde{\kappa})^{-1}\boldsymbol{K}'(\boldsymbol{K}+\tilde{\kappa})^{-1} + \tilde{S}^2(\boldsymbol{K}+\tilde{\kappa})^{-2}\boldsymbol{K}\frac{\gamma_{\boldsymbol{K},\boldsymbol{K}'}}{1-\gamma}. \tag{16}$$

Here $\gamma_{\boldsymbol{K}\boldsymbol{K}} \equiv \frac{\mathrm{df}^2_{\boldsymbol{\Sigma}}\mathrm{df}^2_{\boldsymbol{K},\boldsymbol{K}'}}{\mathrm{df}_1\tilde{\mathrm{df}}_1}$.

We next prove Lemma 2.4, which gives an equivalent for

$$(\hat{\boldsymbol{\Sigma}}+\lambda)^{-1}\boldsymbol{X}^\top\boldsymbol{K}'\boldsymbol{X}(\hat{\boldsymbol{\Sigma}}+\lambda)^{-1} = (\hat{\boldsymbol{\Sigma}}+\lambda)^{-1}\boldsymbol{\Sigma}^{1/2}\boldsymbol{Z}^\top\boldsymbol{K}^{1/2}\,\boldsymbol{K}'\,\boldsymbol{K}^{1/2}\boldsymbol{Z}\boldsymbol{\Sigma}^{1/2}(\hat{\boldsymbol{\Sigma}}+\lambda)^{-1}.$$

This time, because there are an odd number of $\boldsymbol{Z}$s on both the left and the right side of $\boldsymbol{K}'$ in all diagrams, the disconnected terms which average each side separately will vanish. We are left with just the connected term.

Again, we sum over the $\boldsymbol{K}$ and $\boldsymbol{\Sigma}$ loops. The first term (which involves no $\boldsymbol{\Sigma}$ loops) is:

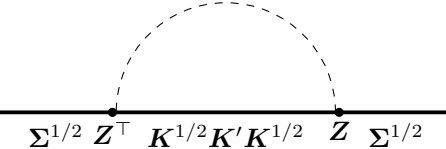

The innermost loop will be a $\boldsymbol{K}$ loop involving an insertion of $\boldsymbol{K}'$ in the middle. As in the calcualtion of the $\boldsymbol{\Sigma}'$ loop in the prior case, this contributes

$$\tilde{S}^2\frac{1}{T}\mathrm{Tr}[\boldsymbol{K}\boldsymbol{K}'(\boldsymbol{K}+\tilde{\kappa})^{-2}] \equiv \tilde{S}^2\mathrm{df}^2_{\boldsymbol{K},\boldsymbol{K}'}.$$

Between this innermost loop and the outer resolvents, there must be an even number $2n \geq 0$ of alternating $\boldsymbol{\Sigma},\boldsymbol{K}$ loops. The contributions of such loops are unchanged. Each pair gives a factor of $\gamma$. We thus get:

$$(\hat{\boldsymbol{\Sigma}}+\lambda)^{-1}\boldsymbol{X}^\top\boldsymbol{K}'\boldsymbol{X}(\hat{\boldsymbol{\Sigma}}+\lambda)^{-1} \simeq S^2\tilde{S}^2(\boldsymbol{\Sigma}+\lambda)^{-2}\boldsymbol{\Sigma}\frac{\mathrm{df}^2_{\boldsymbol{K},\boldsymbol{K}'}}{1-\gamma}. \tag{17}$$

Contracting this with an arbitrary matrix $\boldsymbol{\Sigma}'$ and multiplying by $\frac{\lambda^2}{T}$ gives the more symmetric form:

$$\frac{\lambda^2}{T}\mathrm{Tr}\left[\boldsymbol{\Sigma}'(\hat{\boldsymbol{\Sigma}}+\lambda)^{-1}\boldsymbol{X}^\top\boldsymbol{K}'\boldsymbol{X}(\hat{\boldsymbol{\Sigma}}+\lambda)^{-1}\right] \simeq \frac{\mathrm{df}^2_{\boldsymbol{\Sigma},\boldsymbol{\Sigma}'}\mathrm{df}^2_{\boldsymbol{K},\boldsymbol{K}'}}{\mathrm{df}_1\tilde{\mathrm{df}}_1}\frac{1}{1-\gamma}. \tag{18}$$

We can further simplify this by adopting the shorthand $\gamma_{\boldsymbol{\Sigma},\boldsymbol{\Sigma}',\boldsymbol{K},\boldsymbol{K}'} = \frac{\mathrm{df}^2_{\boldsymbol{\Sigma},\boldsymbol{\Sigma}'}\mathrm{df}^2_{\boldsymbol{K},\boldsymbol{K}'}}{\mathrm{df}_1 \tilde{\mathrm{df}}_1}$.

This result can alternatively be derived using a straightforward but tedious and significantly less conceptually illuminating argument based on a one-point deterministic equivalent for a 'sourced' resolvent. Concretely, this argument starts by writing

$$(\hat{\boldsymbol{\Sigma}} + \lambda)^{-1}\boldsymbol{X}^\top \boldsymbol{K}'\boldsymbol{X}(\hat{\boldsymbol{\Sigma}} + \lambda)^{-1} = \frac{\partial}{\partial J}[\boldsymbol{\Sigma}^{1/2}\boldsymbol{Z}^\top(\boldsymbol{K} + J\boldsymbol{K}^{1/2}\boldsymbol{K}'\boldsymbol{K}^{1/2})\boldsymbol{Z}\boldsymbol{\Sigma}^{1/2} + \lambda]^{-1}\Big|_{J=0},$$

and then proceeds by using a one-point deterministic equivalent before implicitly differentiating the resulting $S$-transforms. Then, various applications of the duality relation allow the result to be simplified into the form obtained using diagrammatics above.

## C.3. Duality relations and derivatives

Here, we explicitly state all duality relationships that one can derive. In the section on 1-point deterministic equivalents, we have proven that:

$$q\mathrm{df}_1 \equiv q\mathrm{df}^1_{\boldsymbol{\Sigma}}(\kappa) \simeq q\mathrm{df}^1_{\hat{\boldsymbol{\Sigma}}}(\lambda) = \mathrm{df}^1_{\hat{\boldsymbol{K}}}(\lambda) \simeq \mathrm{df}^1_{\boldsymbol{K}}(\tilde{\kappa}) \equiv \tilde{\mathrm{df}}_1 \tag{19}$$

We have also seen the first duality relation, namely that:

$$\frac{\kappa\tilde{\kappa}}{\lambda} = \lambda S\tilde{S} = \frac{1}{q\mathrm{df}_1} = \frac{1}{\tilde{\mathrm{df}}_1} \tag{20}$$

Logarithmic differentiation of this yields a second duality relation:

$$\frac{d\log\kappa}{d\log\lambda} + \frac{d\log\tilde{\kappa}}{d\log\lambda} = 1 + \frac{\mathrm{df}_1 - \mathrm{df}^2_{\hat{\boldsymbol{\Sigma}}}(\lambda)}{\mathrm{df}_1}.$$

Here we have used that

$$\frac{d\mathrm{df}_1}{d\lambda} = \frac{\mathrm{df}^2_{\hat{\boldsymbol{\Sigma}}}(\lambda) - \mathrm{df}^1_{\hat{\boldsymbol{\Sigma}}}(\lambda)}{\lambda}. \tag{21}$$

We stress that unlike with $\mathrm{df}_1$, $\mathrm{df}^2_{\hat{\boldsymbol{\Sigma}}}(\lambda) \neq \mathrm{df}^2_{\boldsymbol{\Sigma}}(\kappa) = \mathrm{df}_2$. Since the right hand side can be estimated from the data, this gives us a way to turn an estimate of $\frac{d\log\kappa}{d\log\lambda}$ into an estimate for $\frac{d\log\tilde{\kappa}}{d\log\lambda}$ or vice-versa.

Next, we relate $\mathrm{df}_2$ to $\tilde{\mathrm{df}}_2$. We will be explicit and write all dfs with subscripts to avoid confusion. We have, by differentiating (19):

$$q\kappa\partial_\kappa\mathrm{df}^1_{\boldsymbol{\Sigma}}(\kappa) = q\frac{\kappa}{\lambda}\frac{d\lambda}{d\kappa}\lambda\partial_\lambda\mathrm{df}^1_{\hat{\boldsymbol{\Sigma}}}(\lambda) = \frac{\kappa}{\lambda}\frac{d\lambda}{d\kappa}\lambda\partial_\lambda\mathrm{df}^1_{\hat{\boldsymbol{K}}}(\lambda) = \frac{\kappa}{\tilde{\kappa}}\frac{d\tilde{\kappa}}{d\tilde{\kappa}}\tilde{\kappa}\partial_{\tilde{\kappa}}\mathrm{df}^1_{\boldsymbol{K}}(\tilde{\kappa})$$

Evaluating the left and right sides using (21) we get:

$$q(\mathrm{df}^2_{\boldsymbol{\Sigma}}(\kappa) - \mathrm{df}^1_{\boldsymbol{\Sigma}}(\kappa)) = \frac{d\log\tilde{\kappa}}{d\log\kappa}(\mathrm{df}^2_{\boldsymbol{K}}(\tilde{\kappa}) - \mathrm{df}^1_{\boldsymbol{K}}(\tilde{\kappa}))$$

This yields:

$$\tilde{\mathrm{df}}_2 \equiv \mathrm{df}^2_{\boldsymbol{K}}(\tilde{\kappa}) = q\mathrm{df}^1_{\boldsymbol{\Sigma}}(\kappa) + q\frac{d\log\tilde{\kappa}}{d\log\kappa}(\mathrm{df}^2_{\boldsymbol{\Sigma}}(\kappa) - \mathrm{df}^1_{\boldsymbol{\Sigma}}(\kappa)) \tag{22}$$

Similarly one can write an estimate of $\mathrm{df}_2$ from just the data alone:

$$\mathrm{df}_2 \equiv \mathrm{df}^2_{\boldsymbol{\Sigma}}(\kappa) = \mathrm{df}^1_{\hat{\boldsymbol{\Sigma}}}(\lambda) + \frac{d\log\lambda}{d\log\kappa}(\mathrm{df}^2_{\hat{\boldsymbol{\Sigma}}}(\lambda) - \mathrm{df}^1_{\hat{\boldsymbol{\Sigma}}}(\lambda)) = \mathrm{df}^1_{\hat{\boldsymbol{\Sigma}}}(\lambda) + \frac{d\log\lambda}{d\log\kappa}\partial_\lambda\mathrm{df}^1_{\hat{\boldsymbol{\Sigma}}}(\lambda).$$

Plugging this back into (22) gives an istimate of $\tilde{\mathrm{df}}_2$ from the data alone. This is crucial in allowing the CorrGCV to be efficiently computed.

We now calculate the derivative of $\kappa, \tilde{\kappa}$ on $\lambda$. We have

$$\kappa = S\lambda \Rightarrow \frac{d\log\lambda}{d\log\kappa} = 1 + \frac{d\log 1/S}{d\log\kappa}$$

$$= 1 + \frac{d\log 1/S}{d\log\mathrm{df}_1}\frac{d\log\mathrm{df}_1}{d\log\kappa}$$

$$= 1 - \frac{d\log 1/S}{d\log\mathrm{df}_1}\frac{\mathrm{df}_1 - \mathrm{df}_2}{\mathrm{df}_1}.$$

Using $\tilde{\mathrm{df}}_1 = q\mathrm{df}_1$ we write

$$S = \frac{1}{1 - \tilde{\mathrm{df}}_1}S_{\boldsymbol{K}}(\tilde{\mathrm{df}}_1) = \frac{1}{\tilde{\mathrm{df}}_1\mathrm{df}^{-1}_{\boldsymbol{K}}(\tilde{\mathrm{df}}_1)}.$$

Differentiating this gives:

$$\frac{d\log 1/S}{d\log \tilde{\mathrm{df}}_1} = 1 + \frac{\tilde{\mathrm{df}}_1}{\tilde{\kappa}\mathrm{df}'_1(\tilde{\kappa})} = 1 - \frac{\tilde{\mathrm{df}}_1}{\tilde{\mathrm{df}}_1 - \tilde{\mathrm{df}}_2} = -\frac{\tilde{\mathrm{df}}_2}{\tilde{\mathrm{df}}_1 - \tilde{\mathrm{df}}_2}.$$

All together this is:

$$\frac{d\log \lambda}{d\log \kappa} = 1 + \frac{\tilde{\mathrm{df}}_2}{\tilde{\mathrm{df}}_1 - \tilde{\mathrm{df}}_2}\frac{\mathrm{df}_1 - \mathrm{df}_2}{\mathrm{df}_1}$$

$$= \frac{\mathrm{df}_1\tilde{\mathrm{df}}_1 - \mathrm{df}_2\tilde{\mathrm{df}}_2}{\mathrm{df}_1\tilde{\mathrm{df}}_1 - \mathrm{df}_1\tilde{\mathrm{df}}_2}$$

$$= \frac{1 - \gamma}{1 - \frac{\tilde{\mathrm{df}}_2}{\tilde{\mathrm{df}}_1}}.$$

This finally yields:

$$\frac{\partial \kappa}{\partial \lambda} = S\frac{1 - \frac{\tilde{\mathrm{df}}_2}{\tilde{\mathrm{df}}_1}}{1 - \gamma}. \tag{23}$$

An analogous argument yields:

$$\frac{\partial \tilde{\kappa}}{\partial \lambda} = \tilde{S}\frac{1 - \frac{\mathrm{df}_2}{\mathrm{df}_1}}{1 - \gamma}. \tag{24}$$

Note also that when $\boldsymbol{K} = \mathbf{I}$, $\tilde{\mathrm{df}}_2 = (\tilde{\mathrm{df}}_1)^2 = q^2\mathrm{df}_1^2$, yielding

$$\gamma \equiv \frac{\mathrm{df}_2\tilde{\mathrm{df}}_2}{\mathrm{df}_1\tilde{\mathrm{df}}_1} = q\mathrm{df}_2.$$

This recovers the uncorrelated ridge regression setting.

### C.4. Correlated samples and noise, uncorrelated test point

By applying the deterministic equivalences proved in the prior section, we can directly obtain an exact formula for the asymptotic form of the training and generalization error for a linear model trained on a correlated dataset and evaluated on an uncorrelated test point. In fact, we can do better. The deterministic equivalence (15) allows us to easily consider the case where the test point has a different covariance $\boldsymbol{\Sigma}'$ from the covariance $\boldsymbol{\Sigma}$ of the training set. This allows us to state the formula for generalization under covariate shift as well. In one fell swoop, we thus prove Theorems 3.1, 3.2, and 3.3 by directly proving Theorem 3.3, from which the other two results follow as special cases.

We recall that

$$\hat{\boldsymbol{w}} = \hat{\boldsymbol{\Sigma}}(\hat{\boldsymbol{\Sigma}} + \lambda)^{-1}\bar{\boldsymbol{w}} + (\hat{\boldsymbol{\Sigma}} + \lambda)^{-1}\frac{\boldsymbol{X}^\top\boldsymbol{\epsilon}}{T}, \quad \mathbb{E}_{\boldsymbol{\epsilon}}[\boldsymbol{\epsilon}\boldsymbol{\epsilon}^\top] = \sigma_\epsilon^2\boldsymbol{K}'.$$

We have that the generalization error (for a test point whose distribution has covariance $\boldsymbol{\Sigma}'$) is:

$$R_g = (\bar{\boldsymbol{w}} - \hat{\boldsymbol{w}})^\top\boldsymbol{\Sigma}'(\bar{\boldsymbol{w}} - \hat{\boldsymbol{w}})$$

$$= \underbrace{\lambda^2\bar{\boldsymbol{w}}^\top(\hat{\boldsymbol{\Sigma}} + \lambda)^{-1}\boldsymbol{\Sigma}'(\hat{\boldsymbol{\Sigma}} + \lambda)^{-1}\bar{\boldsymbol{w}}}_{\text{Signal}} + \underbrace{\sigma_\epsilon^2\frac{1}{T^2}\mathrm{Tr}[\boldsymbol{\Sigma}'(\hat{\boldsymbol{\Sigma}} + \lambda)^{-1}\boldsymbol{X}^\top\boldsymbol{K}'\boldsymbol{X}(\hat{\boldsymbol{\Sigma}} + \lambda)^{-1}]}_{\text{Noise}}.$$

The Signal term is immediately obtained by applying the deterministic equivalence in (15).

$$\mathrm{Signal} = \underbrace{\kappa^2\bar{\boldsymbol{w}}^\top\boldsymbol{\Sigma}'(\hat{\boldsymbol{\Sigma}} + \lambda)^{-2}\boldsymbol{\Sigma}\bar{\boldsymbol{w}}}_{\text{Bias}^2} + \underbrace{\kappa^2\boldsymbol{w}^\top\boldsymbol{\Sigma}(\hat{\boldsymbol{\Sigma}} + \lambda)^{-2}\boldsymbol{\Sigma}\bar{\boldsymbol{w}}\frac{\gamma_{\boldsymbol{\Sigma},\boldsymbol{\Sigma}'}}{1 - \gamma}}_{\mathrm{Var}_{\boldsymbol{X}}}.$$

Here we have explicitly delineated which parts of the signal term are due to the bias of the estimator, and which terms are $\mathrm{Var}_{\boldsymbol{X}}$. The latter can be removed by bagging the estimator over different datasets. Note that in diagrammatic language the bias term corresponds exactly the to the disconnected averages of the left and right resolvents separately, which makes sense, as it is the generalization obtained by first averaging the predictor over different training sets before calculating the test risk.

Similarly, by applying the deterministic equivalence (18), we obtain:

$$\mathrm{Noise} = \sigma_\epsilon^2\underbrace{\frac{\gamma_{\boldsymbol{\Sigma},\boldsymbol{\Sigma}',\boldsymbol{K},\boldsymbol{K}'}}{1 - \gamma}}_{\mathrm{Var}_{\boldsymbol{X}\boldsymbol{\epsilon}}}.$$

This yields

$$R_g \simeq \underbrace{\kappa^2 \bar{\boldsymbol{w}}(\boldsymbol{\Sigma}+\kappa)^{-1}\boldsymbol{\Sigma}'(\boldsymbol{\Sigma}+\kappa)^{-1}\bar{\boldsymbol{w}}}_{\text{Bias}^2} + \underbrace{\kappa^2 \frac{\gamma_{\boldsymbol{\Sigma},\boldsymbol{\Sigma}'}}{1-\gamma}\bar{\boldsymbol{w}}\boldsymbol{\Sigma}(\boldsymbol{\Sigma}+\kappa)^{-2}\bar{\boldsymbol{w}}}_{\text{Var}_{\boldsymbol{X}}} + \underbrace{\frac{\gamma_{\boldsymbol{\Sigma},\boldsymbol{\Sigma}',\boldsymbol{K},\boldsymbol{K}'}}{1-\gamma}\sigma_\epsilon^2}_{\text{Var}_{\boldsymbol{X}\epsilon}}. \tag{25}$$

Specializing to the case of $\boldsymbol{\Sigma} = \boldsymbol{\Sigma}'$, $\boldsymbol{K} = \boldsymbol{K}'$ yields the in-distribution matched-noise-correlation setting.

We next treat the training error. We have

$$\hat{R}_{in} = \frac{1}{T}|\boldsymbol{y} - \hat{\boldsymbol{y}}|^2 = \frac{1}{T}|\boldsymbol{X}\boldsymbol{w} + \boldsymbol{\epsilon} - \boldsymbol{X}\hat{\boldsymbol{w}}|^2$$

$$= \frac{1}{T}|\boldsymbol{X}(\hat{\boldsymbol{\Sigma}}+\lambda)^{-1}\bar{\boldsymbol{w}}|^2 + \frac{1}{T}\left|\boldsymbol{\epsilon} - \boldsymbol{X}(\hat{\boldsymbol{\Sigma}}+\lambda)^{-1}\frac{\boldsymbol{X}^\top\boldsymbol{\epsilon}}{T}\right|^2$$

$$= \underbrace{\lambda^2 \bar{\boldsymbol{w}}^\top \hat{\boldsymbol{\Sigma}}(\hat{\boldsymbol{\Sigma}}+\lambda)^{-2}\bar{\boldsymbol{w}}}_{\text{Signal}_{in}} + \underbrace{\frac{\lambda^2}{T}\text{Tr}\left[\boldsymbol{K}'(\hat{\boldsymbol{K}}+\lambda)^{-2}\right]}_{\text{Noise}_{in}}.$$

One can evaluate the signal term by recognizing it as a derivative and applying the chain rule:

$$\text{Signal}_{in} = -\lambda^2 \frac{d}{d\lambda}\bar{\boldsymbol{w}}^\top \hat{\boldsymbol{\Sigma}}(\hat{\boldsymbol{\Sigma}}+\lambda)^{-1}\bar{\boldsymbol{w}}$$

$$= \lambda^2 \frac{d\kappa}{d\lambda}\bar{\boldsymbol{w}}^\top \boldsymbol{\Sigma}(\boldsymbol{\Sigma}+\kappa)^{-2}\bar{\boldsymbol{w}}$$

$$= \lambda^2 S\frac{\tilde{\text{df}}_1 - \tilde{\text{df}}_2}{\tilde{\text{df}}_1}\frac{1}{1-\gamma}\bar{\boldsymbol{w}}^\top \boldsymbol{\Sigma}(\boldsymbol{\Sigma}+\kappa)^{-2}\bar{\boldsymbol{w}}.$$

In the last line, we have inserted the form of $\frac{d\kappa}{d\lambda}$ from (23).

The noise term for generic $\boldsymbol{K}'$ is slightly more involved. We can write it as:

$$\text{Noise}_{in} = -\sigma_\epsilon^2 \frac{\lambda^2}{T}\partial_\lambda \text{Tr}\left[\boldsymbol{K}'(\hat{\boldsymbol{K}}+\lambda)^{-1}\right]$$

$$= -\sigma_\epsilon^2 \frac{\lambda^2}{T}\partial_\lambda \left[\frac{\tilde{\kappa}}{\lambda}\text{Tr}\left[\boldsymbol{K}'(\boldsymbol{K}+\tilde{\kappa})^{-1}\right]\right]$$

$$= \sigma_\epsilon^2 \frac{\tilde{\kappa}}{T}\text{Tr}\left[\boldsymbol{K}'(\boldsymbol{K}+\tilde{\kappa})^{-1}\right] - \sigma_\epsilon^2 \frac{\lambda}{T}\frac{d\tilde{\kappa}}{d\lambda}\text{Tr}\left[\boldsymbol{K}\boldsymbol{K}'(\boldsymbol{K}+\tilde{\kappa})^{-2}\right]$$

Now applying the derivative relationship (24) we get:

$$\text{Noise}_{in} = \sigma_\epsilon^2 \tilde{\kappa}\left[\frac{1}{T}\text{Tr}\,\boldsymbol{K}'(\boldsymbol{K}+\tilde{\kappa})^{-1} - \frac{\text{df}_1 - \text{df}_2}{\text{df}_1}\frac{1}{1-\gamma}\text{df}_{\boldsymbol{K},\boldsymbol{K}'}\right]$$

In the case of $\boldsymbol{K} = \boldsymbol{K}'$ this simplifies to:

$$\text{Noise}_{in} = \sigma_\epsilon^2 \tilde{\kappa}\tilde{\text{df}}_1\left[1 - \frac{\text{df}_1 - \text{df}_2}{\text{df}_1\tilde{\text{df}}_1 - \text{df}_2\tilde{\text{df}}_2}\tilde{\text{df}}_2\right] = \frac{\sigma_\epsilon^2}{S}\frac{\tilde{\text{df}}_1 - \tilde{\text{df}}_2}{\tilde{\text{df}}_1}\frac{1}{1-\gamma}. \tag{26}$$

### C.5. Correlated test point

Here, we prove Theorem 5.1. Specifically, our setting is one where we allow the test point $\boldsymbol{x}$ to have non-vanishing correlation with the training set. Similarly, we assume that the label noise on the test point, $\epsilon$ has nontrivial correlation with the label noise on the test set. We assume a general matrix-Gaussian model for the covariates:

$$\begin{pmatrix}\boldsymbol{x}^\top \\ \boldsymbol{X}\end{pmatrix} \sim \mathcal{N}\left(\boldsymbol{0}, \begin{pmatrix}1 & \boldsymbol{k} \\ \boldsymbol{k}^\top & \boldsymbol{K}\end{pmatrix}\otimes\boldsymbol{\Sigma}\right)$$

where the vector $\boldsymbol{k} \in \mathbb{R}^p$ gives the correlation between the training points and the test point, i.e.,

$$\mathbb{E}[x_i x_{tj}] = \Sigma_{ij}k_t.$$

Similarly for $\epsilon, \boldsymbol{\epsilon}$ we write:

$$\begin{pmatrix}\epsilon \\ \boldsymbol{\epsilon}\end{pmatrix} \sim \mathcal{N}\left(\boldsymbol{0}, \sigma_\epsilon^2\begin{pmatrix}1 & \boldsymbol{k}^\top \\ \boldsymbol{k} & \boldsymbol{K}\end{pmatrix}\right)$$

We further assume that these are independent of the $\boldsymbol{x}$ covariates. That is, $\mathbb{E}[x_{ti}\epsilon] = \mathbb{E}[x_i\epsilon_s] = 0$.

By the usual formulas for Gaussian conditioning, this generative model implies that
$$\boldsymbol{x} \mid \boldsymbol{X} \sim \mathcal{N}(\boldsymbol{X}^\top \boldsymbol{\alpha}, (1-\rho)\boldsymbol{\Sigma})$$
and
$$\epsilon \mid \boldsymbol{\epsilon} \sim \mathcal{N}(\boldsymbol{\alpha}^\top \boldsymbol{\epsilon}, (1-\rho)\sigma_\epsilon^2)$$
where we write
$$\rho = \boldsymbol{k}^\top \boldsymbol{K}^{-1} \boldsymbol{k}, \quad \boldsymbol{\alpha} = \boldsymbol{K}^{-1} \boldsymbol{k}.$$

Then, the out-of-sample risk is
$$R_{out} = \mathbb{E}[\boldsymbol{x}^\top (\bar{\boldsymbol{w}} - \hat{\boldsymbol{w}}) + \epsilon]^2.$$
We have the well-known closed-form expression for $\hat{\boldsymbol{w}}$:
$$\bar{\boldsymbol{w}} - \hat{\boldsymbol{w}} = \lambda(\hat{\boldsymbol{\Sigma}} + \lambda)^{-1}\bar{\boldsymbol{w}} - \frac{1}{T}(\hat{\boldsymbol{\Sigma}} + \lambda)^{-1}\boldsymbol{X}^\top \boldsymbol{\epsilon}.$$

Given that $\mathbb{E}\left[[\boldsymbol{x}_t]_i \epsilon_s\right] = 0$, we have:
$$R_{out} = \underbrace{\lambda^2 \mathbb{E}[\boldsymbol{x}^\top (\hat{\boldsymbol{\Sigma}} + \lambda)^{-1}\bar{\boldsymbol{w}}]^2}_{\text{Signal}} + \underbrace{\mathbb{E}\left[\epsilon - \frac{1}{T}\boldsymbol{x}^\top (\hat{\boldsymbol{\Sigma}} + \lambda)^{-1}\boldsymbol{X}^\top \boldsymbol{\epsilon}\right]^2}_{\text{Noise}}.$$

Here, we have again identified the two terms that appear as signal and noise terms. Taking an expectation over $\boldsymbol{x} \mid \boldsymbol{X}$, we have for the signal term
$$\text{Signal} = (1-\rho)\lambda^2 \mathbb{E}\bar{\boldsymbol{w}}^\top (\hat{\boldsymbol{\Sigma}} + \lambda)^{-1}\boldsymbol{\Sigma}(\hat{\boldsymbol{\Sigma}} + \lambda)^{-1}\bar{\boldsymbol{w}}$$
$$+ \lambda^2 \mathbb{E}\bar{\boldsymbol{w}}^\top (\hat{\boldsymbol{\Sigma}} + \lambda)^{-1}\boldsymbol{X}^\top \boldsymbol{\alpha}\boldsymbol{\alpha}^\top \boldsymbol{X}(\hat{\boldsymbol{\Sigma}} + \lambda)^{-1}\bar{\boldsymbol{w}}.$$
Leveraging the deterministic equivalences (15), (17), we obtain
$$\text{Signal} = \frac{\kappa^2}{1-\gamma}\bar{\boldsymbol{w}}^\top \boldsymbol{\Sigma}(\boldsymbol{\Sigma} + \kappa)^{-2}\bar{\boldsymbol{w}}\left[1 - \rho + \tilde{\kappa}^2 \boldsymbol{\alpha}^\top \boldsymbol{K}(\boldsymbol{K} + \tilde{\kappa})^{-2}\boldsymbol{\alpha}\right] \tag{27}$$

Similarly, taking an expectation first over $\epsilon \mid \boldsymbol{\epsilon}$ and then over $\boldsymbol{x} \mid \boldsymbol{X}$, we have
$$\text{Noise} = (1-\rho)\sigma_\epsilon^2 + \mathbb{E}\left[\boldsymbol{\alpha}^\top \boldsymbol{\epsilon} - \frac{1}{T}\boldsymbol{x}^\top (\hat{\boldsymbol{\Sigma}} + \lambda)^{-1}\boldsymbol{X}^\top \boldsymbol{\epsilon}\right]^2$$
$$= (1-\rho)\left[\frac{1}{T^2}\mathbb{E}\boldsymbol{\epsilon}^\top \boldsymbol{X}(\hat{\boldsymbol{\Sigma}} + \lambda)^{-1}\boldsymbol{\Sigma}(\hat{\boldsymbol{\Sigma}} + \lambda)^{-1}\boldsymbol{X}\boldsymbol{\epsilon} + \sigma_\epsilon^2\right]$$
$$+ \mathbb{E}\left[\boldsymbol{\alpha}^\top \boldsymbol{\epsilon} - \boldsymbol{\alpha}^\top \hat{\boldsymbol{K}}(\hat{\boldsymbol{K}} + \lambda)^{-1}\boldsymbol{\epsilon}\right]^2.$$
Noting that
$$\mathbf{I}_T - \hat{\boldsymbol{K}}(\hat{\boldsymbol{K}} + \lambda)^{-1} = \lambda(\hat{\boldsymbol{K}} + \lambda)^{-1}$$
this simplifies to
$$\text{Noise} = (1-\rho)\left[\frac{1}{T^2}\mathbb{E}\boldsymbol{\epsilon}^\top \boldsymbol{X}(\hat{\boldsymbol{\Sigma}} + \lambda)^{-1}\boldsymbol{\Sigma}(\hat{\boldsymbol{\Sigma}} + \lambda)^{-1}\boldsymbol{X}\boldsymbol{\epsilon} + \sigma_\epsilon^2\right] + \lambda^2 \mathbb{E}\left[\boldsymbol{\alpha}^\top (\hat{\boldsymbol{K}} + \lambda)^{-1}\boldsymbol{\epsilon}\right]^2.$$
Finally, taking the remaining expectation over $\boldsymbol{\epsilon}$, we have
$$\text{Noise} = \sigma_\epsilon^2(1-\rho)\left[\mathbb{E}\frac{1}{T^2}\text{Tr}[\boldsymbol{X}(\hat{\boldsymbol{\Sigma}} + \lambda)^{-1}\boldsymbol{\Sigma}(\hat{\boldsymbol{\Sigma}} + \lambda)^{-1}\boldsymbol{X}\boldsymbol{K}] + 1\right]$$
$$+ \sigma_\epsilon^2 \lambda^2 \mathbb{E}\boldsymbol{\alpha}^\top (\hat{\boldsymbol{K}} + \lambda)^{-1}\boldsymbol{K}(\hat{\boldsymbol{K}} + \lambda)^{-1}\boldsymbol{\alpha}.$$
Now again leveraging deterministic equivalence (17) as well as (16) we obtain:
$$\text{Noise} = \sigma_\epsilon^2(1-\rho)\left[\frac{\gamma}{1-\gamma} + 1\right] + \frac{\sigma_\epsilon^2}{1-\gamma}\tilde{\kappa}^2 \boldsymbol{\alpha}^\top \boldsymbol{K}(\boldsymbol{K} + \tilde{\kappa})^{-2}\boldsymbol{\alpha}$$
$$= \frac{\sigma_\epsilon^2}{1-\gamma}\left[1 - \rho + \tilde{\kappa}^2 \boldsymbol{\alpha}^\top \boldsymbol{K}(\boldsymbol{K} + \tilde{\kappa})^{-2}\boldsymbol{\alpha}\right]. \tag{28}$$
All together Equations (27) and (28) together give the desired result:
$$R_{out}^{\boldsymbol{k}} = R_{out}^{\boldsymbol{k}=0}\left[1 - \rho + \tilde{\kappa}^2 \boldsymbol{\alpha}^\top \boldsymbol{K}(\boldsymbol{K} + \tilde{\kappa})^{-2}\boldsymbol{\alpha}\right].$$

One could straightforwardly extend this to the case where the label noise correlations are different from those of the

covariates with relative ease. As doing so adds complexity to the formulas without much conceptual gain, we will not do so here.

## D. Bias-variance decompositions

Often in prior work, it has been convention to call the proportional to the signal $\bar{\boldsymbol{w}}$ the **Bias**$^2$ and the term proportional to the noise $\sigma_\epsilon^2$ the **variance** component of the total risk. Strictly speaking from the perspective of a fine-grained analysis of variance, this is not true. This was pointed out in detail by (Adlam & Pennington, 2020; Lin & Dobriban, 2021), where fine-grained bias variance decompositions were performed for random feature models.

The estimator $\hat{\boldsymbol{w}}$ is given by:

$$\hat{\boldsymbol{w}} = \hat{\boldsymbol{\Sigma}}(\hat{\boldsymbol{\Sigma}} + \lambda)^{-1}\bar{\boldsymbol{w}} + (\hat{\boldsymbol{\Sigma}} + \lambda)^{-1}\frac{\boldsymbol{X}^\top\boldsymbol{\epsilon}}{T}.$$

We see that the estimator is sensitive to both the draw of $\boldsymbol{\epsilon}$ *and* the choice of $\boldsymbol{X}$. Even in the case of no noise, $\sigma_\epsilon^2 = 0$, the estimator would still have variance over different draws of $\boldsymbol{X}$. This variance can be removed if one had access to multiple datasets by **bagging** over the data. In the limit of infinite bagging, this amounts to a data average. This yields, by deterministic equivalence:

$$\mathbb{E}_{\boldsymbol{X}}\hat{\boldsymbol{w}} = \boldsymbol{\Sigma}(\boldsymbol{\Sigma} + \kappa)^{-1}\bar{\boldsymbol{w}}.$$

Consequently the risk of this estimator would be just the bias:
$$\text{Bias}^2 = R(\mathbb{E}_{\boldsymbol{X},\boldsymbol{\epsilon}}\hat{\boldsymbol{w}}) = \kappa^2\bar{\boldsymbol{w}}^\top\boldsymbol{\Sigma}(\boldsymbol{\Sigma} + \kappa)^{-2}\bar{\boldsymbol{w}}.$$

Similarly, if one fixed the dataset $\boldsymbol{X}$ but averaged over different draws of the noise $\boldsymbol{\epsilon}$, the risk of this averaged predictor would be the same as the risk of a predictor trained on a noiseless dataset $\sigma_\epsilon^2 = 0$. Note that this predictor still has variance over the draw of $\boldsymbol{X}$. We thus get that:

$$\text{Var}_{\boldsymbol{X}} = R(\mathbb{E}_{\boldsymbol{\epsilon}}\hat{\boldsymbol{w}}) - \text{Bias}^2 = \kappa^2\bar{\boldsymbol{w}}^\top\boldsymbol{\Sigma}(\boldsymbol{\Sigma} + \kappa)^{-2}\bar{\boldsymbol{w}}\frac{\gamma}{1 - \gamma}.$$

The remaining term is $\text{Var}_{\boldsymbol{X},\boldsymbol{\epsilon}}$, since it is removable either by bagging over datasets $\boldsymbol{X}$ or by averaging over noise $\boldsymbol{\epsilon}$. It is given by:

$$\text{Var}_{\boldsymbol{X},\boldsymbol{\epsilon}} = R(\hat{\boldsymbol{w}}) - \text{Bias}^2 - \text{Var}_{\boldsymbol{X}} = \frac{\gamma}{1 - \gamma}\sigma_\epsilon^2.$$

We stress this is true under both correlated and uncorrelated data, given that $\kappa, \gamma$ is appropriately defined. This decomposition was computed for linear models in (Canatar et al., 2021b; Atanasov et al., 2024). These equations also hold for out of distribution risk, as given in (25).

## E. Scaling analysis

### E.1. Review of optimal rates

The analysis of the scaling properties of linear regression under power-law decay in the covariance $\boldsymbol{\Sigma}$ and target $\bar{\boldsymbol{w}}$ along the principal components of $\boldsymbol{\Sigma}$ was studied in detail in (Caponnetto & Vito, 2005; Caponnetto & De Vito, 2007). It has received renewed attention given that it can sharply characterize the scaling properties of kernel methods, especially the neural tangent kernel (Jacot et al., 2018) on a variety of realistic datasets (Spigler et al., 2020; Bordelon et al., 2020; Cui et al., 2021). This has implications for neural scaling laws (Kaplan et al., 2020; Bahri et al., 2024) in the lazy regime of neural network training identified in (Chizat et al., 2019).

One defines the source and capacity exponents $\alpha, r$ respectively by looking at the decay of the eigenvalues $\lambda_k$ of $\boldsymbol{\Sigma}$ and components $\bar{w}_k$ of $\bar{\boldsymbol{w}}$ along those eigendirections. Then, for many real datasets one observes the power laws:
$$\lambda_k \sim k^{-\alpha}, \bar{w}_k \sim k^{-\beta}.$$

It is advantageous to write $\beta = 2\alpha r + 1$, as the exponent $r$ appears naturally in the final rates. Then, in the ridgeless limit one has $q\text{df}_1 = 1$. From this one can easily obtain that
$$\kappa \sim T^{-\alpha}.$$

At finite ridge, more generally one has
$$\kappa \sim \max(\lambda, T^{-\alpha}). \tag{29}$$

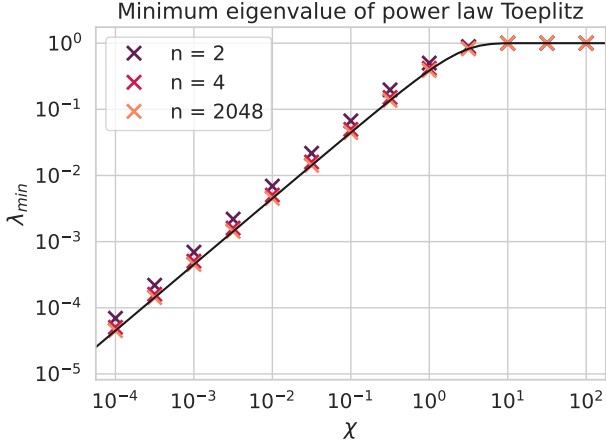

*Figure 7.* Verification of the minimum eigenvalue for a Toeplitz matrix with power law decay $K_{ts} = (1 + |t - s|)^{-\chi}$. We see a great match between (31) (solid black line) and the empirics (x shapes) for matrix sizes ranging between $2 \times 2$ and $1024 \times 1024$. In particular, this guarantees that the matrix will never be singular even at very small positive $\chi$, corresponding to strong power law correlations. It is interesting to note that the minimum eigenvalue of a $2 \times 2$ Toeplitz matrix of this form, which is simply $\lambda_{\min} = 1 - 2^{-\chi}$, is relatively close to the minimum eigenvalue of the infinite operator; their ratio decreases from $-\frac{\log(2)}{\log(2/\pi)} \simeq 1.534\ldots$ at $\chi \downarrow 0$ towards 1 as $\chi \to \infty$.

For this reason, $\kappa$ is often called the resolution or the signal capture threshold. It determines which eigencomponents are learned and which are not. Further, taking $\sigma_\epsilon = 0$, one gets the scaling:

$$R_{out} = R_g \sim \kappa^{-2\min(r,1)}$$

If one takes the ridge to scale with the data as $\lambda \sim T^{-\ell}$ as in (Cui et al., 2021), the full equation in the noiseless setting becomes

$$R_{out} \sim T^{-2\min(\alpha,\ell)\min(r,1)}$$

These are the rates predicted in (Caponnetto & De Vito, 2007).

If one passes data through $N$ linear random features before performing, one can change the resolution scaling by adding a further bottleneck:

$$\kappa \sim \max(\lambda, T^{-\alpha}, N^{-\alpha}). \tag{30}$$

This plays an important role in current models of neural scaling laws (Bahri et al., 2024; Maloney et al., 2022; Bordelon et al., 2024; Atanasov et al., 2024; Defilippis et al., 2024). It is therefore interesting to ask whether the presence of strong correlations could also affect the scaling laws by adding a further bottleneck to (30). We answer this in the negative.

### E.2. Strong correlations do not affect scaling exponents

Here, we ask whether strongly correlated data could change the scaling in (29). First, we note that for correlated data, as long as each new sample is linearly independent of the prior ones in $\mathbb{R}^T$, we have that $\boldsymbol{K}$ does not have any zero eigenvalues. In general, the $T \times T$ correlation matrix $\boldsymbol{K}$ will have Toeplitz structure when the correlations are stationary and will be invertible as long as each new datapoint is not a linear combination of the prior ones. It could however be that the smallest eigenvalue goes to zero as $T \to \infty$. This would lead $S_{\boldsymbol{K}}$ to blow up and could lead to a worse scaling law. We show that this will not happen for exponential, nearest neighbor, and power law correlations. In the latter case, we require some assumptions on the power law exponent.

**Proposition E.1.** *Under exponential, nearest neighbor, and power law correlations, the $T \times T$ correlation matrix $\boldsymbol{K}$ remains nonsingular as $T \to \infty$.*

*Proof.* We can study the invertibility of the limiting $T \to \infty$ matrix by applying standard results on the asymptotic behavior of Hermitian Toeplitz matrices (Böttcher et al., 2009). Let the autocorrelation be $K_{t,t+\tau} = a_\tau$. The three cases of interest

are exponential correlations

$$a_\tau = e^{-|\tau|/\xi}$$

with correlation length $\xi > 0$, nearest-neighbor correlations

$$a_\tau = \begin{cases} 1 & \tau = 0 \\ b/2 & \tau = \pm 1 \\ 0 & \text{otherwise} \end{cases}$$

with $b \in [0, 1)$ (see Appendix I), and power-law correlations

$$a_\tau = \frac{1}{(1 + |\tau|)^\chi}$$

with exponent $\chi > 0$. The key object is then the *symbol* of the Toeplitz operator corresponding to the finite matrices, which is the Fourier transform of the autocorrelation:

$$w(\theta) = \sum_{k=-\infty}^{\infty} a_k e^{ik\theta},$$

where $\theta \in [0, 2\pi)$. For the three cases of interest, we can easily work out that

$$w(\theta) = \frac{\sinh(1/\xi)}{\cosh(1/\xi) - \cos(\theta)} \qquad \text{(exponential)}$$

$$w(\theta) = 1 + b\cos(\theta) \qquad \text{(nearest-neighbor)}$$

$$w(\theta) = e^{-i\theta}\,\mathrm{Li}_\chi(e^{i\theta}) + e^{i\theta}\,\mathrm{Li}_\chi(e^{-i\theta}) - 1 \qquad \text{(power-law)}$$

where $\mathrm{Li}_\chi(z)$ is the polyogarithm of order $\chi$. It is easy to verify that all three of these are real, positive continuously-differentiable functions that are symmetric about $\theta = \pi$ and minimized at that point, where they take the values

$$w(\pi) = \tanh\frac{1}{2\xi} \qquad \text{(exponential)}$$

$$w(\pi) = 1 - b \qquad \text{(nearest-neighbor)}$$

$$w(\pi) = 2(1 - 2^{1-\chi})\zeta(\chi) - 1 \qquad \text{(power-law)}, \tag{31}$$

where for power-law correlations the result for $\chi = 1$ is understood in a limiting sense.[4] By standard results on the minimum eigenvalue of Hermitian Toeplitz matrices, as these symbols are well-behaved $w(\pi)$ both gives the $T \to \infty$ limit of the minimum eigenvalue and a lower bound on the minimum eigenvalue at any finite $T$. A stronger bound is given in (Böttcher et al., 2009), but we require only the weakest statement. Thus, as for each case of interest $w(\pi) > 0$ within the valid range of parameters, all of the matrices of interest remain invertible. For illustrative purposes, we compare (31) to numerical computation of the minimum eigenvalue for power-law correlations in Figure 7. □

Under the assumption that $K$ remains invertible in the large $T$ limit, we have that $\lim_{\lambda \to 0} \mathrm{df}_K(\lambda) = 1$ at any value of $T$. More generally $\mathrm{df}_K$ is a continuous monotonically decreasing function of $\lambda$ that goes from 1 when $\lambda = 0$ to 0 as $\lambda \to \infty$. As $\lambda \to \infty$ we can expand and get at linear order that $\mathrm{df}_K = \frac{1}{\lambda}\frac{1}{T}\mathrm{Tr}K = \frac{1}{\lambda}$. Writing:

$$\mathrm{df}_K(\lambda) = \frac{S_K(\mathrm{df}_K(\lambda))^{-1}}{\lambda + S_K(\mathrm{df}_K(\lambda))^{-1}}$$

implies that at small $\lambda$, $S_K$ is bounded from above by a $T$-independent constant at any $T$. $S_K$ is also bounded from below by 1 (see F.1). Then, writing

$$\kappa = \frac{\lambda}{1 - q\mathrm{df}_1}S_K(q\mathrm{df}_1),$$

we see that $S_K$ will not contribute any pole or zero that would effect the scaling properties of $\kappa$ as $\lambda \to 0$. Consequently, correlated data does not affect the scaling law.

## F. Double descent analysis

In this appendix, we study how double descent is effected by correlations. We also more generally study how correlations affect the key quantities of interest relative to the uncorrelated setting.

---

[4] In particular, we have $\lim_{\chi \to 1} w(\pi) = \log 4 - 1 \simeq 0.386\ldots$.

### F.1. Bounds on renormalized ridges

**Lemma F.1.** *For positive-definite $\boldsymbol{K}$ such that $\frac{1}{T}\operatorname{Tr}(\boldsymbol{K}) = 1$ we have that $S_{\boldsymbol{K}}(\tilde{\mathrm{df}}) \geq 1$ for all values of $\tilde{\mathrm{df}} \in (0, 1]$.*

*Proof.* We adapt an argument from Zavatone-Veth & Pehlevan (2023). We recognize that for fixed $\tilde{\kappa} > 0$ the function:

$$\rho \mapsto \frac{\rho}{\tilde{\kappa} + \rho}$$

is concave. Consquently, Jensen's inequality yields:

$$\mathrm{df}_{\boldsymbol{K}}(\tilde{\kappa}) = \mathbb{E}_\rho \frac{\rho}{\tilde{\kappa} + \rho} \leq \frac{\mathbb{E}_\rho \rho}{\tilde{\kappa} + \mathbb{E}_\rho \rho} = \frac{1}{\tilde{\kappa} + 1} = \mathrm{df}_{\mathbf{I}}(\tilde{\kappa})$$

pointwise in $\tilde{\kappa}$. Here $\mathbb{E}_\rho$ is the expectation over the spectrum of $\boldsymbol{K}$. We have also used that $\mathbb{E}_\rho[\rho] = \frac{1}{T}\operatorname{Tr}(\boldsymbol{K}) = 1$. The above inequality is strict unless $\boldsymbol{K} = \mathbf{I}_T$ or $\tilde{\kappa} = 0$.

We next observe that both $\mathrm{df}_{\boldsymbol{K}}(\lambda)$ and $\mathrm{df}_{\mathbf{I}}(\lambda)$ are monotonically decreasing function in $\lambda$ that are equal to 1 only when $\lambda = 0$. This means that the solutions to the equations $\tilde{\mathrm{df}} = \mathrm{df}_{\boldsymbol{K}}^1(\lambda)$ and $\tilde{\mathrm{df}} = \mathrm{df}_{\mathbf{I}}^1(\lambda)$ are unique for all $\mathrm{df} \in (0, 1]$ and is also monotonically decreasing. Consequently we have:

$$\mathrm{df}_{\boldsymbol{K}}^{-1}(\tilde{\mathrm{df}}) \leq \mathrm{df}_{\mathbf{I}}^{-1}(\tilde{\mathrm{df}}) = \frac{1 - \tilde{\mathrm{df}}}{\tilde{\mathrm{df}}}$$

Upon dividing both sides by $\mathrm{df}_{\boldsymbol{K}}^{-1}$ we get the desired equality $\Rightarrow S_{\boldsymbol{K}}(\tilde{\mathrm{df}}) \geq 1$. This is an equality when $\boldsymbol{K} = \mathbf{I}_T$. When $\tilde{\kappa} = 0$ and thus $\tilde{\mathrm{df}} = 1$, $S_{\boldsymbol{K}}(\tilde{\mathrm{df}})$ may still be greater than one.

$\square$

**Proposition F.2.** *Let $\kappa_c$ be the renormalized ridge $\lambda S$ when the data has correlation structure $\boldsymbol{K}$ and let $\kappa_u$ be the corresponding value of $\kappa$ when there is no correlation between data points. We have $\kappa_c \geq \kappa_u$. Moreover, in the ridgeless limit $\kappa_c = \kappa_u$.*

*Proof.* In the correlated setting, we recall that the renormalized ridge $\kappa_c$ solves

$$\kappa_c = \frac{\lambda S_{\boldsymbol{K}}(q\mathrm{df}_{\boldsymbol{\Sigma}}^1(\kappa_c))}{1 - q\mathrm{df}_{\boldsymbol{\Sigma}}^1(\kappa_c)}.$$

Call the numerator $\tilde{\lambda}$. From the preceding lemma we have that $\tilde{\lambda} \geq \lambda$. Thus, we have that $\kappa_c$ is equivalent to the self-consistent solution for the equation

$$\kappa_c = \frac{\tilde{\lambda}}{1 - q\mathrm{df}_{\boldsymbol{\Sigma}}^1(\kappa_c)}.$$

But this is the same as the self-consistent equation for a linear regression problem with an explicit ridge $\tilde{\lambda} \geq \lambda$. The proposition then follows from the fact that $\kappa$ is monotonic in $\lambda$ in the uncorrelated ridge regression setting.

As $\lambda \to 0$, $\kappa$ is entirely determined by the pole structure of $q\mathrm{df}_{\boldsymbol{\Sigma}}^1 = 1$. This is independent of any structure on $\boldsymbol{K}$ and so in the ridgeless limit, $\kappa_c = \kappa_u$. $\square$

**Corollary F.3.** *Fix $q$. The presence of correlations either decreases or keeps constant $\mathrm{df}_1, \tilde{\mathrm{df}}_1$. In the ridgeless limit, they are unchanged.*

*Proof.* Evaluating $\mathrm{df}_1$ as $\mathrm{df}_1 = \mathrm{df}_{\boldsymbol{\Sigma}}^1(\kappa)$ we have since $\kappa_c \geq \kappa_u$ and $\mathrm{df}_1$ is monotone decreasing in $\kappa$ that $\mathrm{df}_1$ decreases. Since $\tilde{\mathrm{df}}_1 = q\mathrm{df}_1$, this also decreases. Because $\kappa$ is unchanged in the ridgeless limit, the last part of the corollary follows. $\square$

**Corollary F.4.** *Fix $q$. The presence of correlations either decreases or keeps constant $\mathrm{df}_2$. In the ridgeless limit, it is unchanged.*

*Proof.* The proof is as in the prior corollary, noting that $\mathrm{df}_2 = \mathrm{df}_{\boldsymbol{\Sigma}}^2(\kappa)$ is monotone decreasing in $\kappa$ and $\kappa_c \geq \kappa_u$. $\square$

**Proposition F.5.** *Let $\tilde{\kappa}_c$ be the renormalized ridge $\lambda\tilde{S}$ when the data has correlation structure $\boldsymbol{K}$ and let $\tilde{\kappa}_u$ be the corresponding value of $\tilde{\kappa}$ when there is no correlation between data points. We have $\tilde{\kappa}_c \leq \tilde{\kappa}_u$. Moreover, in the ridgeless limit $\tilde{\kappa}_c \leq \tilde{\kappa}_u$ still.*

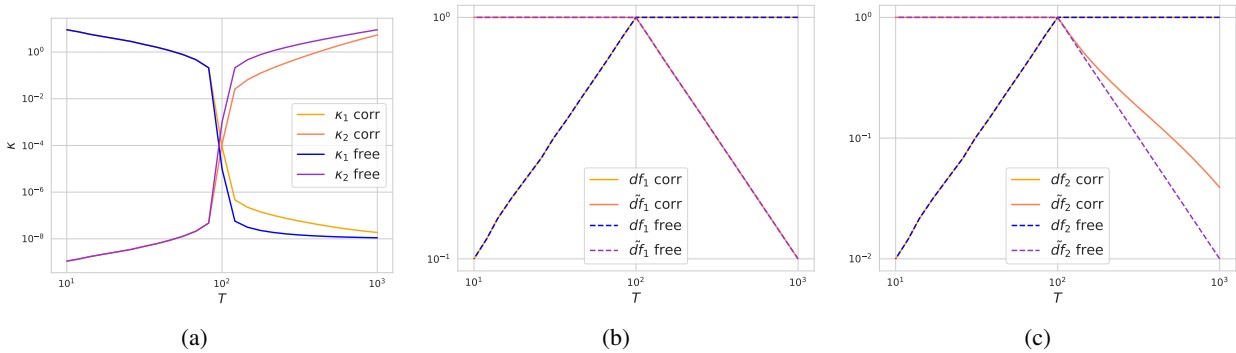

(a)                 (b)                 (c)

*Figure 8.* Theory curves comparing $\kappa, \tilde{\kappa}, \mathrm{df}_1, \tilde{\mathrm{df}}_1, \mathrm{df}_2, \tilde{\mathrm{df}}_2$ for correlated data $\boldsymbol{K} \neq \boldsymbol{I}$ vs uncorrelated data $\boldsymbol{K} = \boldsymbol{I}$ at $\lambda = 10^{-8}$ for $N = 100$ on isotropic data. For the correlations, we choose exponential correlations with length $\xi = 10^2$. a) We see that when $T < N$, $\kappa, \tilde{\kappa}$ strongly agree, whereas when $T > N$, $\kappa$ becomes of order the ridge while $\tilde{\kappa}$ becomes order 1. The effects of correlations on $\tilde{\kappa}$ are therefore noticeable in this limit. b) We see that the $\mathrm{df}_1, \tilde{\mathrm{df}}_1$ are unchanged between correlated (solid) and uncorrelated (dashed) data when the ridge is small. c) We see a similar behavior for $\mathrm{df}_2$ but not for $\tilde{\mathrm{df}}_2$ when $T > N$. Still, at first order near $T = N$, we see agreement of $\tilde{\mathrm{df}}_2$ between the correlated and uncorrelated settings, as predicted by our theory.

*Proof.* When $\lambda \neq 0$ we can rewrite the duality relation (20) as

$$\frac{q}{\lambda} \kappa \mathrm{df}^1_{\boldsymbol{\Sigma}}(\kappa) = \frac{1}{\tilde{\kappa}}.$$

This separately holds true for the pairs $\kappa_u, \tilde{\kappa}_u$ and $\kappa_c, \tilde{\kappa}_c$. We note that on the left hand side $\kappa \mathrm{df}^1_{\boldsymbol{\Sigma}}(\kappa)$ is a sum of $\frac{\kappa}{\lambda_i + \kappa}$ over the eigenspectrum $\lambda_i$ of $\boldsymbol{\Sigma}$. Each term is monotone increasing in $\kappa$, and thus increases as we go from the uncorrelated $\kappa_u$ to the correlated $\kappa_c$. Consequently, $\tilde{\kappa}_c \leq \tilde{\kappa}_u$.

In the ridgeless limit, by the prior corollary, $\tilde{\mathrm{df}}$ is unchanged between correlated and uncorrelated data. Consequently, we have:

$$\frac{1}{1 + \tilde{\kappa}_u} = \mathrm{df}^1_{\boldsymbol{I}_T}(\kappa_u) = \mathrm{df}^1_{\boldsymbol{K}}(\kappa_c) \leq \frac{1}{1 + \tilde{\kappa}_c}$$

Thus, we again have $\tilde{\kappa}_c \leq \tilde{\kappa}_u$ with equality only if $\boldsymbol{K} = \boldsymbol{I}_T$ or $\kappa_c = 0$. $\qquad\square$

### F.2. Ridgeless limit

The limiting behavior of the two renormalized ridges $\kappa$ and $\tilde{\kappa}$ depends on whether one is in the underparameterized regime $T > N$ ($q < 1$) or the overparameterized regime $T < N$ ($q > 1$). In the underparameterized regime, we have that $\kappa \downarrow 0$ as $\lambda \downarrow 0$, while generically $\tilde{\kappa}$ remains non-zero and solves the self-consistent equation $q = \tilde{\mathrm{df}}_1(\tilde{\kappa})$. Conversely, in the overparameterized regime $\tilde{\kappa} \downarrow 0$ as $\lambda \downarrow 0$ while $\kappa$ remains non-zero and solves $1/q = \mathrm{df}_1(\kappa)$. We numerically illustrate these complementary limiting behaviors in Appendix J.

Our task is now to determine the corresponding limits of the out-of-sample risk. First, we consider the case of matched correlations. There, in the underparameterized regime we have $\gamma \to \tilde{\mathrm{df}}_2/q$, while in the overparameterized regime we have $\gamma \to q\mathrm{df}_2$. As a result, we find that

$$\lim_{\lambda \downarrow 0} R_g \simeq \begin{cases} \dfrac{\tilde{\mathrm{df}}_2}{q - \tilde{\mathrm{df}}_2} \sigma_\epsilon^2 & q < 1, \\[2ex] \dfrac{\kappa^2}{1 - q\mathrm{df}_2} \boldsymbol{w}^\top \boldsymbol{\Sigma}(\boldsymbol{\Sigma} + \kappa)^{-2} \boldsymbol{w} + \dfrac{q\mathrm{df}_2}{1 - q\mathrm{df}_2} \sigma_\epsilon^2 & q > 1, \end{cases}$$

In the overparameterized ridgeless setting, since neither $\kappa$, $\mathrm{df}_1$ or $\mathrm{df}_2$ is modified, the generalization error is exactly identical as that for linear regression. In the underparameterized limit, we study how $\tilde{\mathrm{df}}_2$ behaves. We can write

$$\mathrm{df}^2_{\boldsymbol{K}}(\tilde{\kappa}_c) = \mathbb{E}_\rho \left( \frac{\rho}{\rho + \tilde{\kappa}_c} \right)^2 \geq \left[ \mathbb{E}_\rho \left( \frac{\rho}{\rho + \tilde{\kappa}_c} \right) \right]^2 = \mathrm{df}^1_{\boldsymbol{K}}(\tilde{\kappa}_c)^2 = \mathrm{df}^1_{\boldsymbol{I}}(\kappa_u)^2 = \mathrm{df}^2_{\boldsymbol{I}}(\kappa_u),$$

where the expectation is taken over the eigenspectrum of $\boldsymbol{K}$. Moreover, near $q = 1$ when $\tilde{\kappa}$ is small we have that at linear

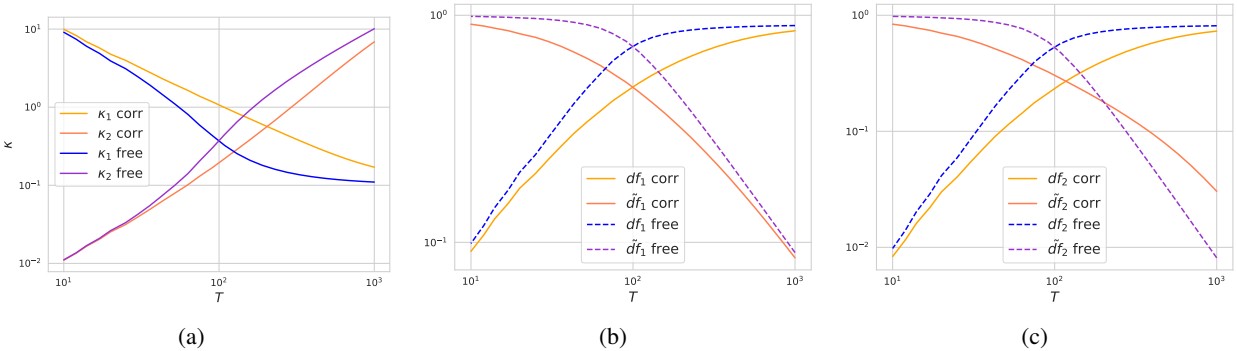

| (a) | (b) | (c) |

*Figure 9.* Theory curves comparing $\kappa, \tilde{\kappa}, \mathrm{df}_1, \tilde{\mathrm{df}}_1, \mathrm{df}_2, \tilde{\mathrm{df}}_2$ for correlated data $\boldsymbol{K} \neq \boldsymbol{I}$ vs uncorrelated data $\boldsymbol{K} = \boldsymbol{I}$ at $\lambda = 10^{-1}$ for $N = 100$ on isotropic data. For the correlations, we choose exponential correlations with length $\xi = 10^2$. a) We see across the board that in the presence of explicit ridge, $\kappa$ grows while $\tilde{\kappa}$ shrinks under correlations b) We see that the $\mathrm{df}_1, \tilde{\mathrm{df}}_1$ are always decreased by the presence of correlations when the ridge is present. c) We see a similar behavior for $\mathrm{df}_2$. For $\tilde{\mathrm{df}}_2$, we also see a decrease in the neighborhood of the double descent peak $T = N$. Still, at first order near $T = N$, we see agreement of $\tilde{\mathrm{df}}_2$ between the correlated and uncorrelated settings, as predicted by our theory.

order:

$$\mathrm{df}_{\boldsymbol{K}}^2(\tilde{\kappa}_c) = 1 - 2\tilde{\kappa}_c \frac{1}{T} \mathrm{Tr}[\boldsymbol{K}^{-1}] + O(\tilde{\kappa}_c^2) = \mathrm{df}_{\boldsymbol{K}}^1(\tilde{\kappa}_c)^2 + O(\tilde{\kappa}_c^2) = \mathrm{df}_{\boldsymbol{I}}^2(\tilde{\kappa}_u) + O(\tilde{\kappa}_u^2).$$

Thus, at first order the double descent peak is unaffected by correlations in the underparameterized regime as well. We illustrate these observations in Figure 8.

### F.3. Effective ridge enhancement

If $\lambda$ is finite, we have by Proposition F.2 that by F.5 that $\kappa_c > \kappa_u$ and $\tilde{\kappa}_c < \tilde{\kappa}_u$. Specifically, let $\hat{\boldsymbol{\Sigma}}_{\boldsymbol{K}}$ be the empirical covariance of the data when the data points have correlations $\boldsymbol{K}$. We can write the following equivalence:

$$\hat{\boldsymbol{\Sigma}}_{\boldsymbol{K}}(\hat{\boldsymbol{\Sigma}}_{\boldsymbol{K}} + \lambda)^{-1} \simeq \hat{\boldsymbol{\Sigma}}_{\boldsymbol{I}}(\hat{\boldsymbol{\Sigma}}_{\boldsymbol{I}} + \lambda S_{\boldsymbol{K}}(q\mathrm{df}_1))^{-1}$$

Thus adding correlations amounts to increasing the ridge. Because $\boldsymbol{\Sigma}_{\boldsymbol{K}}$ is bounded from above from Proposition E.1, in the ridgeless limit the $\lambda S_{\boldsymbol{K}}$ will remain zero.

When there is explicit ridge, we have that $\mathrm{df}_1, \tilde{\mathrm{df}}_1, \mathrm{df}_2$ will shrink from their uncorrelated values. Because to linear order in $\kappa, \tilde{\kappa}$, we have $\tilde{\mathrm{df}}_2 = (\tilde{\mathrm{df}}_1)^2$ at $q = 1$ and because correlations cause $\tilde{\mathrm{df}}_1^2$ to decrease at first order in $\kappa$ at this point, we get that correlations cause $\tilde{\mathrm{df}}_2$ to decrease in a neighborhood of $q = 1$.

As a result of this, at first order in $\kappa, \tilde{\kappa}$, we can write $\gamma = q\mathrm{df}_2(\kappa)$. Near $q = 1$, this will also shrink in the correlated case relative to the uncorrelated case. Thus, the double descent effect will be further reduced. We illustrate these observations in Figure 9.

### F.4. Mismatched correlations

Now we consider the case of mismatched correlations. We consider the fully general setting in which $\boldsymbol{\Sigma}' \neq \boldsymbol{\Sigma}$ and $\boldsymbol{K}' \neq \boldsymbol{K}$. In the underparameterized regime, we have $\mathrm{df}_{\boldsymbol{\Sigma}\boldsymbol{\Sigma}'}^2 \to \frac{1}{N} \mathrm{Tr}(\boldsymbol{\Sigma}^{-1}\boldsymbol{\Sigma}')$, so $\gamma_{\boldsymbol{\Sigma}\boldsymbol{\Sigma}'} \to q^{-1}\tilde{\mathrm{df}}_2 \frac{1}{N} \mathrm{Tr}(\boldsymbol{\Sigma}^{-1}\boldsymbol{\Sigma}')$. Similarly, $\gamma_{\boldsymbol{\Sigma}\boldsymbol{\Sigma}'\boldsymbol{K}\boldsymbol{K}'} \to q^{-1}\mathrm{df}_{\boldsymbol{K}\boldsymbol{K}'}^2 \frac{1}{N} \mathrm{Tr}(\boldsymbol{\Sigma}^{-1}\boldsymbol{\Sigma}')$. In the overparameterized regime, we similarly have $\mathrm{df}_{\boldsymbol{K}\boldsymbol{K}'}^2 \to \frac{1}{T} \mathrm{Tr}(\boldsymbol{K}^{-1}\boldsymbol{K}')$, which leads to $\gamma_{\boldsymbol{\Sigma}\boldsymbol{\Sigma}'} \to q\mathrm{df}_{\boldsymbol{\Sigma}\boldsymbol{\Sigma}'}^2$ and $\gamma_{\boldsymbol{\Sigma}\boldsymbol{\Sigma}'\boldsymbol{K}\boldsymbol{K}'} \to q\mathrm{df}_{\boldsymbol{\Sigma}\boldsymbol{\Sigma}'}^2 \frac{1}{T} \mathrm{Tr}(\boldsymbol{K}^{-1}\boldsymbol{K}')$. Combining these results, we find that

$$\lim_{\lambda \downarrow 0} R_g \simeq \begin{cases} \dfrac{q^{-1}\mathrm{df}_{\boldsymbol{K}\boldsymbol{K}'}^2 \frac{1}{N} \mathrm{Tr}(\boldsymbol{\Sigma}^{-1}\boldsymbol{\Sigma}')}{1 - q^{-1}\tilde{\mathrm{df}}_2} \sigma_\epsilon^2 & q < 1, \\[2em] \kappa^2 \bar{\boldsymbol{w}}(\boldsymbol{\Sigma} + \kappa)^{-1}\boldsymbol{\Sigma}'(\boldsymbol{\Sigma} + \kappa)^{-1}\bar{\boldsymbol{w}} + \kappa^2 \dfrac{q\mathrm{df}_{\boldsymbol{\Sigma}\boldsymbol{\Sigma}'}^2}{1 - q\mathrm{df}_2} \bar{\boldsymbol{w}}\boldsymbol{\Sigma}(\boldsymbol{\Sigma} + \kappa)^{-2}\bar{\boldsymbol{w}} \\[1em] \quad + \dfrac{q\mathrm{df}_{\boldsymbol{\Sigma}\boldsymbol{\Sigma}'}^2 \frac{1}{T} \mathrm{Tr}(\boldsymbol{K}^{-1}\boldsymbol{K}')}{1 - q\mathrm{df}_2} \sigma_\epsilon^2 & q > 1 \end{cases}$$

The effect of mismatch is easiest to understand in the overparameterized regime, where the factor $\frac{1}{T}\operatorname{Tr}(\boldsymbol{K}^{-1}\boldsymbol{K}')$ multiplies the same expression for the noise term that appears for uncorrelated datapoints. In the special case $\boldsymbol{K}' = \mathbf{I}_T$, we can use Jensen's inequality to bound

$$\frac{1}{T}\operatorname{Tr}(\boldsymbol{K}^{-1}) \geq \frac{1}{\operatorname{Tr}(\boldsymbol{K})/T} = 1$$

with equality iff $\boldsymbol{K} = \mathbf{I}_T$, hence in this special case mismatch generically increases the error.

### F.5. Further comments on the effect of mismatched correlations

Here, we briefly comment further on the case in which the noise is uncorrelated ($\boldsymbol{K}' = \mathbf{I}_T$). In greatest generality, we seek to bound

$$\mathrm{df}^2_{\boldsymbol{K},\boldsymbol{K}'=\mathbf{I}_T}(\tilde{\kappa}) = \frac{1}{T}\operatorname{Tr}[\boldsymbol{K}(\boldsymbol{K}+\tilde{\kappa})^{-2}]$$

in terms of

$$\tilde{\mathrm{df}}_2(\tilde{\kappa}) = \frac{1}{T}\operatorname{Tr}[\boldsymbol{K}^2(\boldsymbol{K}+\tilde{\kappa})^{-2}]$$

We prove a simple upper bound, which follows from a negative association argument:

**Proposition F.6.** *For any $\tilde{\kappa} \geq 0$ and $\boldsymbol{K}$ invertible, we have*

$$\mathrm{df}^2_{\boldsymbol{K},\boldsymbol{K}'=\mathbf{I}_T}(\tilde{\kappa}) \leq \left(\frac{1}{T}\operatorname{Tr}(\boldsymbol{K}^{-1})\right)\tilde{\mathrm{df}}_2(\tilde{\kappa}),$$

*with equality when $\tilde{\kappa} = 0$.*

*Proof.* If $\tilde{\kappa} = 0$, the claim obviously holds with equality so long as $\boldsymbol{K}$ is invertible. Then, for any fixed $\tilde{\kappa} > 0$, define

$$f(\rho) = \frac{\rho^2}{(\rho+\tilde{\kappa})^2}$$

and

$$g(\rho) = \frac{1}{\rho},$$

such that

$$\mathrm{df}^2_{\boldsymbol{K},\boldsymbol{K}'=\mathbf{I}_T}(\tilde{\kappa}) = \mathbb{E}[f(\rho)g(\rho)]$$

and

$$\tilde{\mathrm{df}}_2(\tilde{\kappa}) = \mathbb{E}[g(\rho)]$$

where expectation is taken with respect to the distribution of eigenvalues of $\boldsymbol{K}$. Observe that $f(\rho)$ is for any $\tilde{\kappa} > 0$ a monotone increasing function on $(0,\infty)$, while $g(\rho)$ is a monotone decreasing function on $(0,\infty)$. Then, for any $\rho_1, \rho_2 \in (0,\infty)$, we have

$$[f(\rho_1) - f(\rho_2)][g(\rho_1) - g(\rho_2)] \leq 0,$$

so upon taking expectations for $\rho_1, \rho_2$ independently drawn from the eigenvalue distribution of $\boldsymbol{K}$ we have

$$0 \geq \mathbb{E}\left[[f(\rho_1) - f(\rho_2)][g(\rho_1) - g(\rho_2)]\right] = 2\mathbb{E}[f(\rho)g(\rho)] - 2\mathbb{E}[f(\rho)]\mathbb{E}[g(\rho)]$$

hence

$$\mathbb{E}[f(\rho)g(\rho)] \leq \mathbb{E}[f(\rho)]\mathbb{E}[g(\rho)].$$

Using the definitions of $f$ and $g$, this concludes the proof. □

## G. Asymptotics of previously-proposed extensions of the GCV

Here, we compute the high-dimensional asymptotics of previously-proposed extensions to the GCV in the presence of correlations. We write each estimator in the form

$$\hat{E}_{\text{predicted}} = G\hat{R}_{in}$$

Moreover, we introduce the notation

$$\boldsymbol{H} = (\hat{\boldsymbol{K}} + \lambda)^{-1}\hat{\boldsymbol{K}}$$

for the smoothing matrix, in terms of which the predictions on the training set are given as

$$\hat{y} = Hy,$$

as these estimators make use of this matrix. For comparison, we recall from the main text that the asymptotically precise CorrGCV estimator is

$$G_{\text{CorrGCV}} = S(\text{df}_1) \frac{\tilde{\text{df}}_1}{\tilde{\text{df}}_1 - \tilde{\text{df}}_2}$$

### G.1. The GCV

The standard GCV can be written as

$$G_{\text{GCV}} = \frac{1}{[1 - \frac{1}{T} \text{Tr}(H)]^2},$$

hence we have immediately that

$$G_{\text{GCV}} \simeq \frac{1}{[1 - \tilde{\text{df}}_1(\tilde{\kappa})]^2}.$$

We denote this estimator by $\text{GCV}_1$ in Figure 5 and in all plots.

### G.2. The Altman estimator

Starting from the GCV, Altman (1990) and later Opsomer et al. (2001) consider regression with correlated errors

$$\text{cov}(\epsilon) = \sigma_\epsilon^2 K_\epsilon$$

and and consider the estimator

$$G_{\text{Altman}} = \frac{1}{[1 - \frac{1}{T} \text{Tr}(HK_\epsilon)]^2}$$

Asymptotically, we have immediately that

$$G_{\text{Altman}} \simeq \frac{1}{\{1 - \frac{1}{T} \text{Tr}[K_\epsilon K(K + \tilde{\kappa})^{-1}]\}^2}$$

Putting $K_\epsilon = K$, we can write

$$\frac{1}{T} \text{Tr}[K^2(K + \tilde{\kappa})^{-1}] = \frac{1}{T} \text{Tr}[K(I - \tilde{\kappa}(K + \tilde{\kappa})^{-1})] = \frac{1}{T} \text{Tr}(K) - \tilde{\kappa}\tilde{\text{df}}_1,$$

hence, as $\frac{1}{T} \text{Tr}(K) = 1$ by normalization, we have

$$G_{\text{Altman}} \simeq \frac{1}{\tilde{\kappa}^2 \tilde{\text{df}}_1^2}$$

Using the duality relation

$$\frac{\kappa\tilde{\kappa}}{\lambda} = \frac{1}{\tilde{\text{df}}_1},$$

this reduces to

$$G_{\text{Altman}} \simeq \frac{\kappa^2}{\lambda^2} = S^2.$$

We denote this estimator by $\text{GCV}_2$ in Figure 5 and in all plots.

### G.3. The GCCV estimator of Carmack *et al.*

In our notation, Carmack et al. (2012) again assume that the label noise has

$$\text{cov}(\epsilon) = \sigma_\epsilon^2 K_\epsilon$$

and consider the estimator

$$G_{\text{Carmack}} = \frac{1}{[1 - \frac{1}{T} \text{Tr}(2HK_\epsilon - HK_\epsilon H^\top)]^2}.$$

The first term is identical to Altman (1990)'s estimator, while the second is an additional correction. We have

$$\frac{1}{T} \text{Tr}(HK_\epsilon) \simeq \frac{1}{T} \text{Tr}[K_\epsilon K(K + \tilde{\kappa})^{-1}]$$

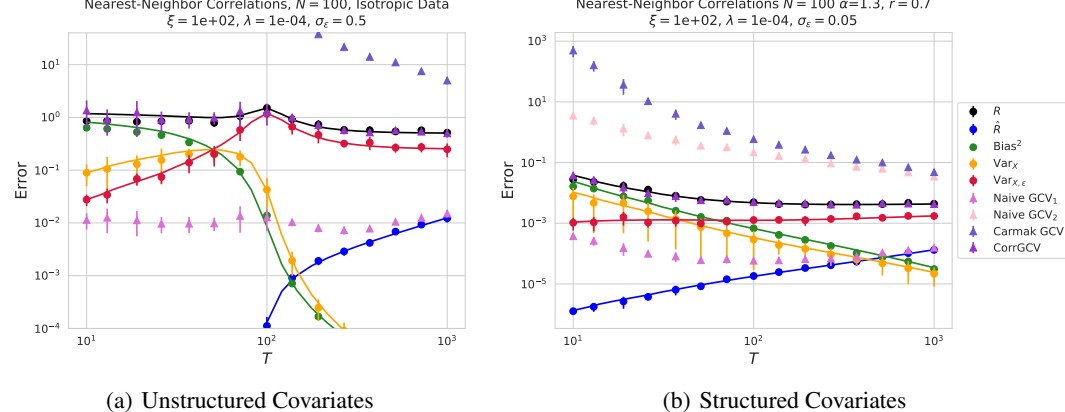

(a) Unstructured Covariates

(b) Structured Covariates

*Figure 10.* Plot of the Carmack et al. (2012) estimator relative to other GCV estimators defined earlier. We note that similar to the naive GCVs 1 and 2, the Carmack estimator fails to correctly predict $R$. a) Unstrucutred Covariates. b) Structured Covariates with source and capacity exponents as labelled.

and

$$\frac{1}{T}\operatorname{Tr}(\boldsymbol{H}\boldsymbol{K}_\epsilon\boldsymbol{H}^\top) = \frac{1}{T}\operatorname{Tr}[\boldsymbol{K}_\epsilon\hat{\boldsymbol{K}}(\hat{\boldsymbol{K}}+\lambda)^{-1}] - \lambda\frac{1}{T}\operatorname{Tr}[\boldsymbol{K}_\epsilon\hat{\boldsymbol{K}}(\hat{\boldsymbol{K}}+\lambda)^{-2}]$$
$$= \frac{1}{T}\operatorname{Tr}[\boldsymbol{K}_\epsilon\hat{\boldsymbol{K}}(\hat{\boldsymbol{K}}+\lambda)^{-1}] + \lambda\partial_\lambda\frac{1}{T}\operatorname{Tr}[\boldsymbol{K}_\epsilon\hat{\boldsymbol{K}}(\hat{\boldsymbol{K}}+\lambda)^{-1}]$$
$$\simeq \frac{1}{T}\operatorname{Tr}[\boldsymbol{K}_\epsilon\boldsymbol{K}(\boldsymbol{K}+\tilde{\kappa})^{-1}] - \lambda\frac{\partial\tilde{\kappa}}{\partial\lambda}\frac{1}{T}\operatorname{Tr}[\boldsymbol{K}_\epsilon\boldsymbol{K}(\boldsymbol{K}+\tilde{\kappa})^{-2}],$$

so

$$\frac{1}{T}\operatorname{Tr}(2\boldsymbol{H}\boldsymbol{K}_\epsilon - \boldsymbol{H}\boldsymbol{K}_\epsilon\boldsymbol{H}^\top) \simeq \frac{1}{T}\operatorname{Tr}[\boldsymbol{K}_\epsilon\boldsymbol{K}(\boldsymbol{K}+\tilde{\kappa})^{-1}] + \lambda\frac{\partial\tilde{\kappa}}{\partial\lambda}\frac{1}{T}\operatorname{Tr}[\boldsymbol{K}_\epsilon\boldsymbol{K}(\boldsymbol{K}+\tilde{\kappa})^{-2}].$$

Setting $\boldsymbol{K}_\epsilon = \boldsymbol{K}$, we have

$$1 - \frac{1}{T}\operatorname{Tr}(2\boldsymbol{H}\boldsymbol{K} - \boldsymbol{H}\boldsymbol{K}\boldsymbol{H}^\top) \simeq \tilde{\kappa}\tilde{\mathrm{df}}_1 - \lambda\frac{\partial\tilde{\kappa}}{\partial\lambda}\tilde{\mathrm{df}}_2$$

as $\frac{1}{T}\operatorname{Tr}(\boldsymbol{K}) = 1$. Now, using (24) and the defining equation $\lambda\tilde{S} = \tilde{\kappa}$, we have

$$1 - \frac{1}{T}\operatorname{Tr}(2\boldsymbol{H}\boldsymbol{K} - \boldsymbol{H}\boldsymbol{K}\boldsymbol{H}^\top) \simeq \tilde{\kappa}\tilde{\mathrm{df}}_1 - \tilde{\kappa}\frac{1 - \frac{\mathrm{df}_2}{\mathrm{df}_1}}{1 - \gamma}\tilde{\mathrm{df}}_2$$

with

$$\gamma = \frac{\mathrm{df}_2}{\mathrm{df}_1}\frac{\tilde{\mathrm{df}}_2}{\tilde{\mathrm{df}}_1}.$$

Then, we have

$$G_{\mathrm{Carmack}} \simeq \left[\tilde{\kappa}\tilde{\mathrm{df}}_1 - \tilde{\kappa}\frac{1 - \frac{\mathrm{df}_2}{\mathrm{df}_1}}{1 - \gamma}\tilde{\mathrm{df}}_2\right]^{-2}.$$

We now write

$$\frac{\mathrm{df}_2}{\mathrm{df}_1} = \gamma\frac{\tilde{\mathrm{df}}_1}{\tilde{\mathrm{df}}_2}$$

and use the duality relation to expand $\tilde{\kappa}\tilde{\mathrm{df}}_1 = \lambda/\kappa = 1/S$, which upon combining terms gives

$$G_{\mathrm{Carmack}} \simeq \left[\frac{1}{S}\frac{1}{1 - \gamma}\frac{\tilde{\mathrm{df}}_1 - \tilde{\mathrm{df}}_2}{\tilde{\mathrm{df}}_1}\right]^{-2}.$$

We recognize the term in brackets as the noise term when $\boldsymbol{K} = \boldsymbol{K}'$ in (26). In general this implies that

$$G_{\text{Carmack}} = (1 - \gamma)^2 \, S^2 \underbrace{\left( \frac{\tilde{\text{df}}_1}{\tilde{\text{df}}_1 - \tilde{\text{df}}_2} \right)^2}_{\text{CorrGCV}^2} = S^2 \left( \frac{1 - \frac{\text{df}_2 \tilde{\text{df}}_2}{\text{df}_1 \tilde{\text{df}}_1}}{1 - \frac{\tilde{\text{df}}_1}{\tilde{\text{df}}_2}} \right)^2 .$$

When $\boldsymbol{K} = \mathbf{I}_T$, $\tilde{\text{df}}_1 / (\tilde{\text{df}}_1 - \tilde{\text{df}}_2) = 1/(1 - 1/(\tilde{\kappa} + 1))$

We plot an example of this estimator for correlated data in Figure 10; note the substantial deviation. This is to be expected, as there was no claim in Carmack et al. (2012) that this estimator should work for $\boldsymbol{X}$ correlations.

## H. Weighted risks and the MMSE estimator

In this Appendix, we briefly comment on related risks and the MMSE estimator. As discussed in the main text, one might consider a weighted loss

$$L_{\boldsymbol{M}}(\boldsymbol{w}) = \frac{1}{T}(\boldsymbol{X}\boldsymbol{w} - \boldsymbol{y})^\top \boldsymbol{M}(\boldsymbol{X}\boldsymbol{w} - \boldsymbol{y}) + \lambda \|\boldsymbol{w}\|^2;$$

for $\boldsymbol{M}$ a general positive-definite symmetric weighting matrix. Under our Gaussian statistical assumptions, this is equivalent to using an unweighted loss with $\boldsymbol{M} \leftarrow \mathbf{I}_T$, $\boldsymbol{K} \leftarrow \boldsymbol{M}^{1/2}\boldsymbol{K}\boldsymbol{M}^{1/2}$, and $\boldsymbol{K}' \leftarrow \boldsymbol{M}^{1/2}\boldsymbol{K}'\boldsymbol{M}^{1/2}$.

We now observe that the Bayesian MMSE estimator is equivalent to minimizing a weighted loss with the particular choice $\boldsymbol{M} = (\boldsymbol{K}')^{-1}$. Under our statistical assumptions, the MMSE estimator is simply given by the posterior mean:

$$\hat{\boldsymbol{w}}_{\text{MMSE}} = \int d\boldsymbol{w} \, \boldsymbol{w} \, p(\boldsymbol{w} \,|\, \boldsymbol{X}, \boldsymbol{y}).$$

Using our Gaussian assumptions on the data and a Gaussian prior corresponding to the ridge penalty, the posterior is

$$\begin{aligned}
p(\boldsymbol{w} \,|\, \boldsymbol{X}, \boldsymbol{y}) &\propto p(\boldsymbol{X}, \boldsymbol{y} \,|\, \boldsymbol{w})p(\boldsymbol{w}) \\
&= p(\boldsymbol{y} \,|\, \boldsymbol{X}, \boldsymbol{w})p(\boldsymbol{X})p(\boldsymbol{w}) \\
&\propto \exp\left( -\frac{1}{2}(\boldsymbol{y} - \boldsymbol{X}\boldsymbol{w})^\top (\boldsymbol{K}')^{-1}(\boldsymbol{y} - \boldsymbol{X}\boldsymbol{w}) - \frac{1}{2}\operatorname{Tr}(\boldsymbol{K}^{-1}\boldsymbol{X}\boldsymbol{\Sigma}^{-1}\boldsymbol{X}^\top) - \frac{1}{2}\lambda\|\boldsymbol{w}\|^2 \right).
\end{aligned}$$

Discarding $\boldsymbol{w}$-independent terms and completing the square, this means that the posterior over $\boldsymbol{w}$ is Gaussian with mean

$$(\boldsymbol{X}^\top (\boldsymbol{K}')^{-1}\boldsymbol{X} + \lambda)^{-1}\boldsymbol{X}^\top (\boldsymbol{K}')^{-1}\boldsymbol{y}$$

and covariance

$$(\boldsymbol{X}^\top (\boldsymbol{K}')^{-1}\boldsymbol{X} + \lambda)^{-1}.$$

Therefore, the MMSE estimator corresponds to minimizing a weighted loss with

$$\boldsymbol{M} = (\boldsymbol{K}')^{-1}.$$

If $\boldsymbol{K}' = \boldsymbol{K}$, this then corresponds to minimizing an unweighted loss for uncorrelated datapoints.

However, we now observe that this procedure is only possible if one has omniscient knowledge of $\boldsymbol{K}'$, as reliably estimating $(\boldsymbol{K}')^{-1}$ from samples is challenging.

## I. $S$-transforms for certain Toeplitz covariance matrices

In this Appendix, we record formulas for the spectral statistics of a few tractable and practically-relevant classes of Toeplitz covariance matrices. We direct the interested reader to work by Kühn & Sollich (2012) or Basak et al. (2014) for studies of the limiting spectral properties of *empirical* autocovariance matrices.

### I.1. Nearest-neighbor correlations

We begin with the simplest non-trivial example: nearest-neighbor correlations. In this case we give a self-contained analysis, which hints at some of the approaches that can be used for more general classes of Toeplitz matrices. Suppose that

$$K_{ts} = \begin{cases} 1 & t = s \\ b/2 & t = s \pm 1 \\ 0 & \text{otherwise} \end{cases}$$

is a symmetric tridiagonal Toeplitz matrix with diagonal elements 1 and off-diagonal elements $b/2$. We have set the diagonal elements equal to 1 without loss of generality as this is equivalent to choosing an overall scale. Tridiagonal Toeplitz matrices are exactly diagonalized by Fourier modes (Noschese et al., 2013), and their eigenvalues are known to be

$$\lambda_t = 1 + b\cos\frac{\pi t}{T+1}$$

where $t = 1, \ldots, T$. Clearly, for the matrix to be positive-definite we must have $|b| < 1$; we will assume $b > 0$ without loss of generality as the cosine term is symmetric. For a test function $\phi$, we therefore have

$$\frac{1}{T}\sum_{t=1}^{T}\phi(\lambda_t) = \frac{1}{T}\sum_{t=1}^{T}\phi\left(1 + b\cos\frac{\pi t}{T+1}\right).$$

Recognizing this as a Riemann sum for an integral with respect to $x = t/(T+1)$, it is easy to see that

$$\lim_{T\to\infty}\frac{1}{T}\sum_{t=1}^{T}\phi(\lambda_t) = \int_0^1 dx\,\phi(1 + b\cos\pi x).$$

Putting $\lambda = 1 + b\cos(\pi x)$, we have $\lambda = 1 + b$ when $x = 0$ and $\lambda = 1 - b$ when $x = \pi$. In this range, we can invert the relationship to find that $x = \arccos[(\lambda - 1)/b]/\pi$. Differentiating, we have $dx = -\frac{1}{\pi\sqrt{(\lambda-1+b)(1+b-\lambda)}}d\lambda$. Thus, the integral becomes

$$\int_{1-b}^{1+b} d\lambda\,\frac{1}{\pi\sqrt{(\lambda - 1 + b)(1 + b - \lambda)}}\phi(\lambda).$$

This shows that for any $b > 0$ the limiting density of eigenvalues is

$$\frac{1}{\pi\sqrt{(\lambda - 1 + b)(1 + b - \lambda)}}\mathbf{1}_{\lambda\in[1-b,1+b]},$$

which is an arcsine distribution centered at 1 of width $b$. In other words, the limiting distribution is simply the pushforward of the uniform measure on $[0, 1]$ by $x \mapsto 1 + b\cos(\pi x)$, much as the distribution at finite size is the pushforward of the uniform measure on $\{1/(T+1), \ldots, T/(T+1)\}$ by the same function.

Applying the result of this digression, the limit of the traced resolvent is

$$g_K(z) \to \int_{1-b}^{1+b} d\lambda\,\frac{1}{\pi\sqrt{(\lambda - 1 + b)(1 + b - \lambda)}}\frac{1}{z - \lambda}$$

$$= \frac{1}{\sqrt{(z-1)^2 - b^2}}$$

From this, we can obtain with a bit of algebra the corresponding $S$-transform

$$S_K(t) = \frac{(1+t) - \sqrt{1 + b^2 t(1+t)}}{(1 - b^2)t}.$$

We observe that the limit as $b \downarrow 0$ of these results gives

$$\lim_{b\downarrow 0} g_K(z) = \frac{1}{z - 1}$$

$$\lim_{b\downarrow 0} S_K(t) = 1,$$

which recover the expected results for the identity matrix.

### I.2. Exponential correlations

Now we consider the case of exponential correlations

$$K_{ts} = e^{-|t-s|/\xi}$$

for some correlation length $\xi$, which is treated in the textbook of Potters & Bouchaud (2020). Though the eigenvectors of this matrix are not precisely Fourier modes at finite size, one can argue that the approximation error in the resolvent resulting from treating $K$ as circulant becomes negligible, and from that determine the limit. In the end, one finds that

$$S_K(t) = \frac{t + 1}{bt + \sqrt{1 + (b^2 - 1)t^2}}, \quad b = \coth 1/\xi.$$

# J. Further experiments

## J.1. Exponential correlations

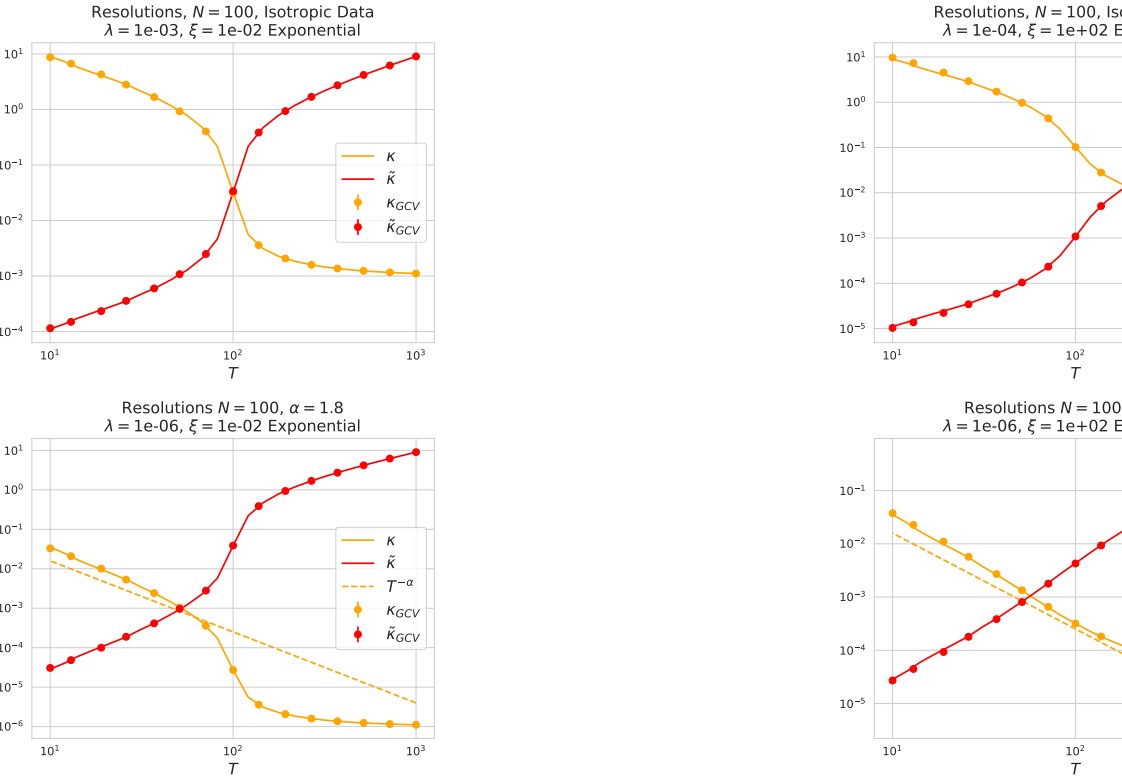

*Figure 11.* Plots of the resolutions $\kappa, \tilde{\kappa}$ under exponential correlations. Top: Isotropic Data. Bottom: Anisotropic power law data with the eigenvalues of $\boldsymbol{\Sigma}$ having power law decay $k^{-\alpha}$, $\alpha = 1.8$. Overlaid in dashed is the expected scaling in the overparameterized regime. Left: $\xi = 10^{-2}$, essentially uncorrelated. Right: $\xi = 10^2$, strongly correlated. We fix $N = 100$ and vary $T$. Overlaid in dashed is the scaling expected in the overparameterized regime when $\alpha > 1$.

In this section we study the predictions of the theory for various relevant quantities in the case of both strong and weak exponential correlations. Plots of the error curves in this setting are numerous in the main text. We test this across both isotropic and structured data.

In Figure 11, we plot several different curves for $\kappa, \tilde{\kappa}$ as $T$ varies from small to large across differently structured datasets. We then plot the corresponding degrees of freedom in Figure 12. Finally, we verify the duality relation holds empirically in Figure 13.

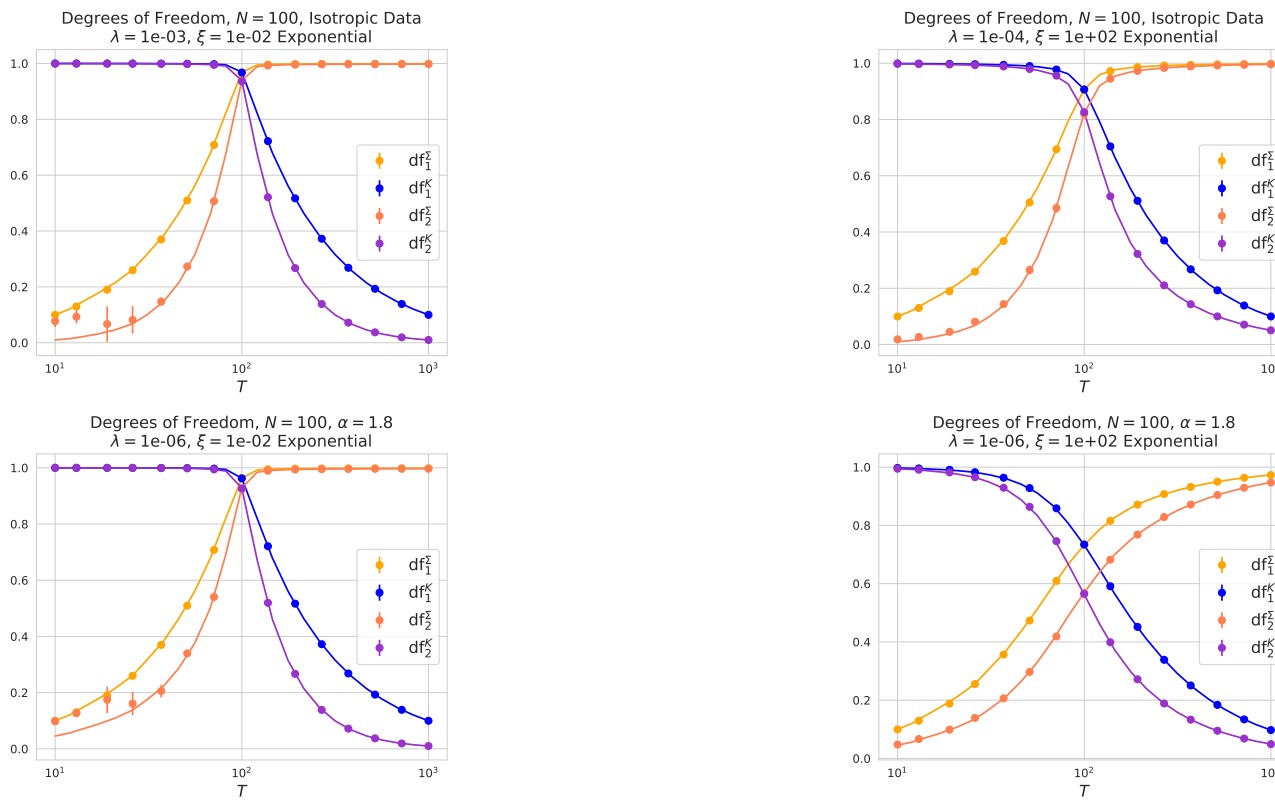

Figure 12. Plots of the the degrees of freedom $\mathrm{df}_1, \tilde{\mathrm{df}}_1, \mathrm{df}_2, \tilde{\mathrm{df}}_2$ under exponential correlations as in Figure 11. Solid lines are theory, dots with error bars are empirics. The lack of substantial error bars stems from the fact that all quantities concentrate. We find that although $\mathrm{df}_2$ can be numerically sensitive, $\tilde{\mathrm{df}}_2$ remains numerically precise even in the presence of strong correlations.

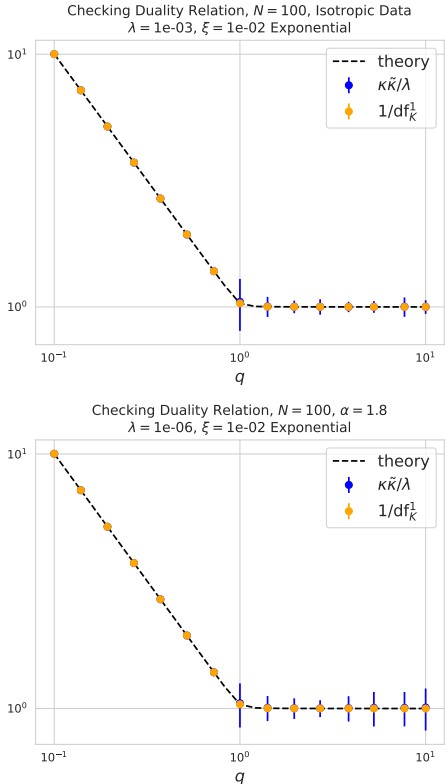
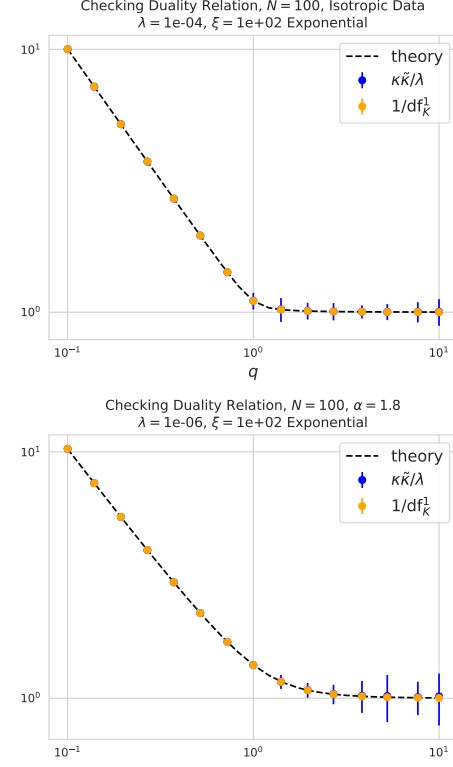

*Figure 13.* Plots verifying the duality relation $\frac{\kappa\tilde{\kappa}}{\lambda} = \frac{1}{\widetilde{df}_1}$ hold for the settings in Figures 11 and 12 under exponential correlations. Dashed black lines lines are theory, dots with error bars are empirics. The lack of substantial error bars stems from the fact that all quantities concentrate.

## J.2. Nearest-neighbor correlations

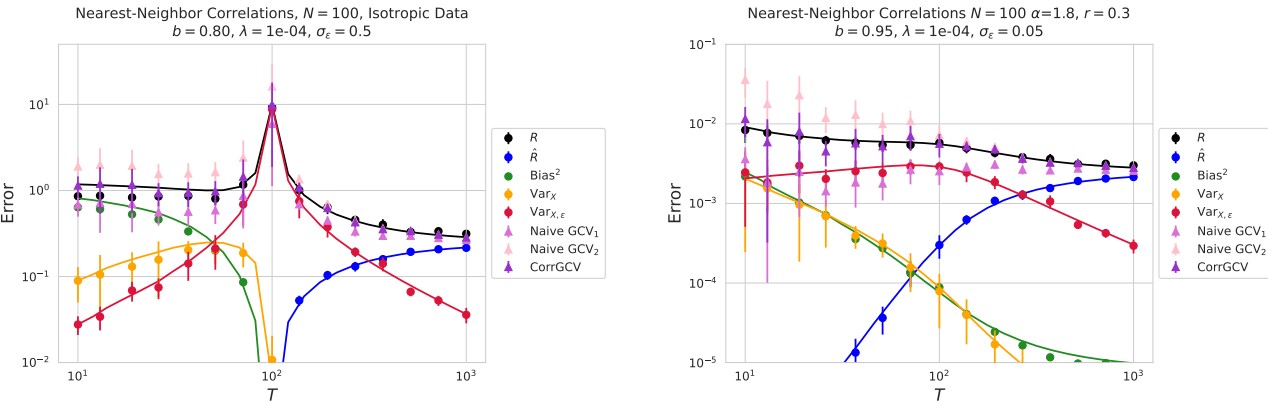

*Figure 14.* Risks, Sources of Variance, and Risk Estimators, similar to Figure 1, but with nearest neighbor correlations. a) Isotropic covariates. b) Structured covariates under source and capacity conditions.

The $S$-transform for a correlation matrix $K$ where only nearest neighbor data points are correlated can be straightforwardly obtained. We do so in Section I.1. In general, the effect of correlations is quite weak unless the off-diagonal correlation element $b/2$ has $b$ close to $1$. In Figure 14, we plot the risk curves and sources of variance in both the setting of isotropic and anisotropic data. In general, all estimators tend to agree better for the choices of $b$ listed compared to the strongly exponentially correlated data. However, only the CorrGCV consistently correctly estimates the out-of-sample risk across $T$, $b$, $\lambda$, and $\sigma_\epsilon$. Especially at small value of $T$, we see that only the CorrGCV estimator is the most reliable.

We further plot the resolutions in Figure 15 and degrees of freedom in Figure 16 across two different values of $b = 0.5, 0.95$ in both the isotropic and anisotropic case.

## J.3. Power-law correlations

For power law correlations, we do not have an explicit analytic formula for $S_K$. We instead estimate it by interpolating $\mathrm{df}_K$ as a function of $\lambda$ and esimating the functional inverse $\mathrm{df}_K^{-1}$, as discussed in Section A.2 . We ensure our interpolator is compatible with autograd so that we can run Algorithm 1 to get estimators for $\mathrm{df}_2, \tilde{\mathrm{df}}_2$ and thus for the CorrGCV.

We show two examples with power-law correlations in Figure 17. We further plot the resolutions in Figure 18 and degrees of freedom in Figure 19 across two different values of $\chi = 1.5, 0.1$ in both the isotropic and anisotropic case.

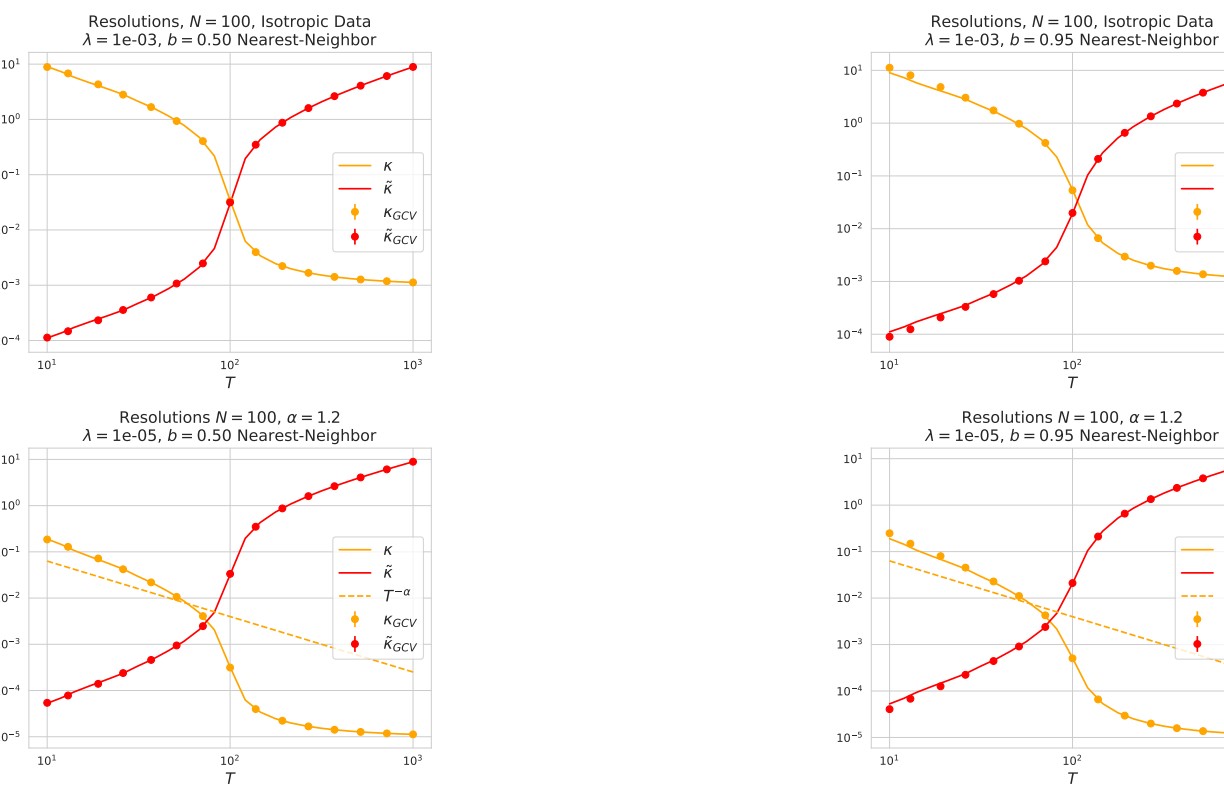

*Figure 15.* Plots of the resolution $\kappa, \tilde{\kappa}$ under nearest neighbor correlations. Top: Isotropic Data. Bottom: Anisotropic power law data with the eigenvalues of $\boldsymbol{\Sigma}$ having power law decay $k^{-\alpha}$, $\alpha = 1.2$. Left: $b = 0.5$, weakly correlated. Right: $b = 0.95$, strongly correlated. We fix $N = 100$ and vary $T$. Solid lines are theory dots with error bars are empirics. Overlaid in dashed is the expected scaling in the overparameterized regime when $\alpha > 1$.

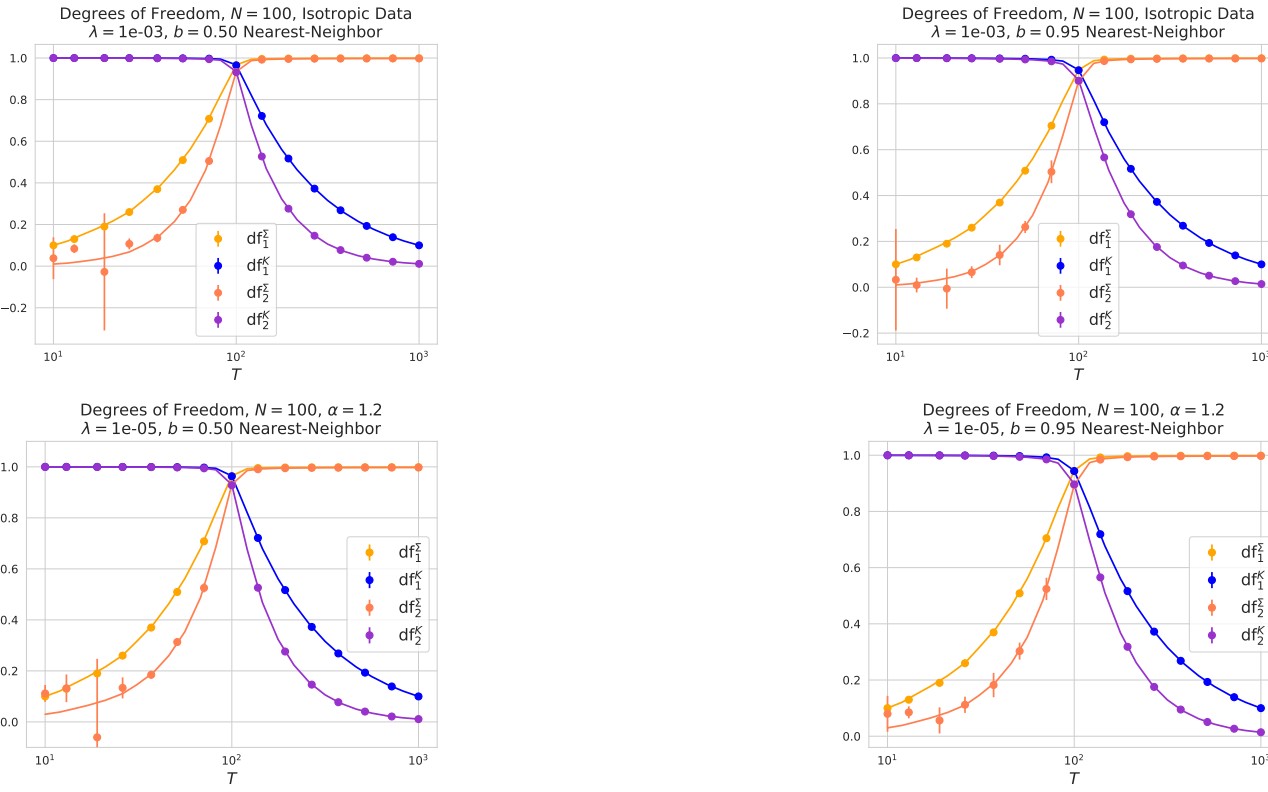

*Figure 16.* Plots of the the degrees of freedom $\mathrm{df}_1, \tilde{\mathrm{df}}_1, \mathrm{df}_2, \tilde{\mathrm{df}}_2$ under nearest-neighbor correlations as in Figure 15 Solid lines are theory, dots with error bars are empirics. The lack of substantial error bars stems from the fact that all quantities concentrate. We find that although $\mathrm{df}_2$ can be numerically sensitive, $\tilde{\mathrm{df}}_2$ remains numerically precise even in the presence of strong correlations.

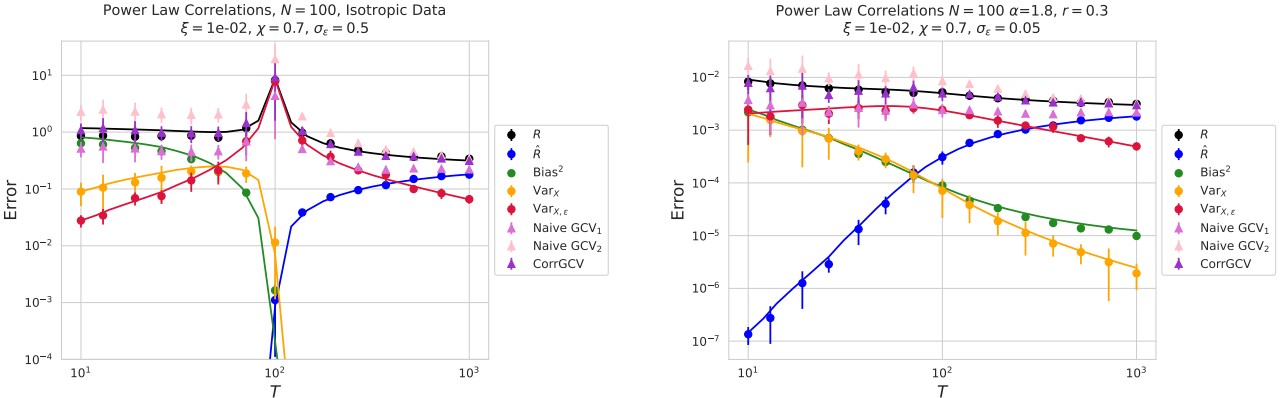

*Figure 17.* Risks, Sources of Variance, and Risk Estimators, similar to Figure 1, but with power law correlations $\mathbb{E}[\boldsymbol{x}_t \cdot \boldsymbol{x}_s] \propto (1+|t-s|)^{-\chi}$ with exponent $\chi = 0.7$. a) Isotropic covariates. b) Structured covariates under source and capacity conditions.

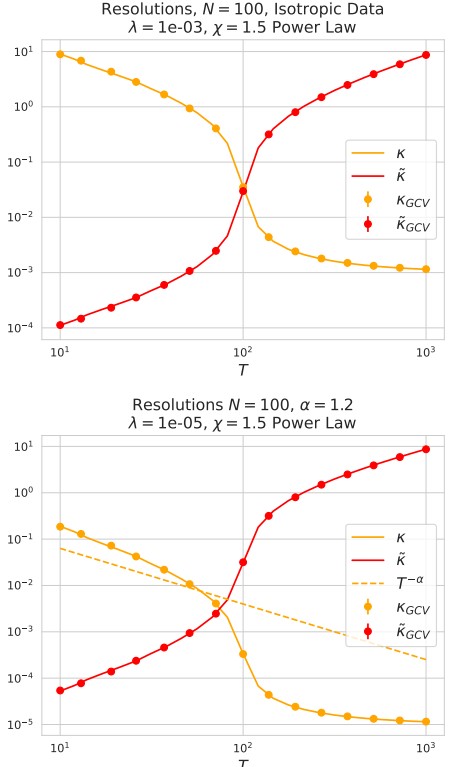
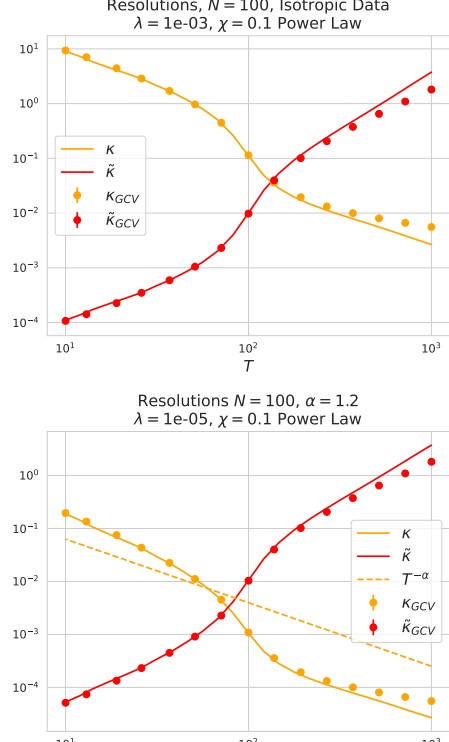

*Figure 18.* Plots of the resolution $\kappa, \tilde{\kappa}$ under power law correlations. Top: Isotropic Data. Bottom: Anisotropic power law data with the eigenvalues of $\Sigma$ having power law decay $k^{-\alpha}$, $\alpha = 1.2$. Overlaid in dashed is the expected scaling in the overparameterized regime. Left: $\chi = 1.5$, weakly correlated. Right: $\chi = 0.1$, strongly correlated. We fix $N = 100$ and vary $T$. For strongly correlated data, we see a slight deviation between $\kappa$ in theory and empirics. This is due to the low-degree polynomial interpolation estimator of $S_{\boldsymbol{K}}(q\mathrm{df}_1)$ performing more poorly near $q\mathrm{df}_1 = 1$. This can be resolved with a more flexible interpolant.

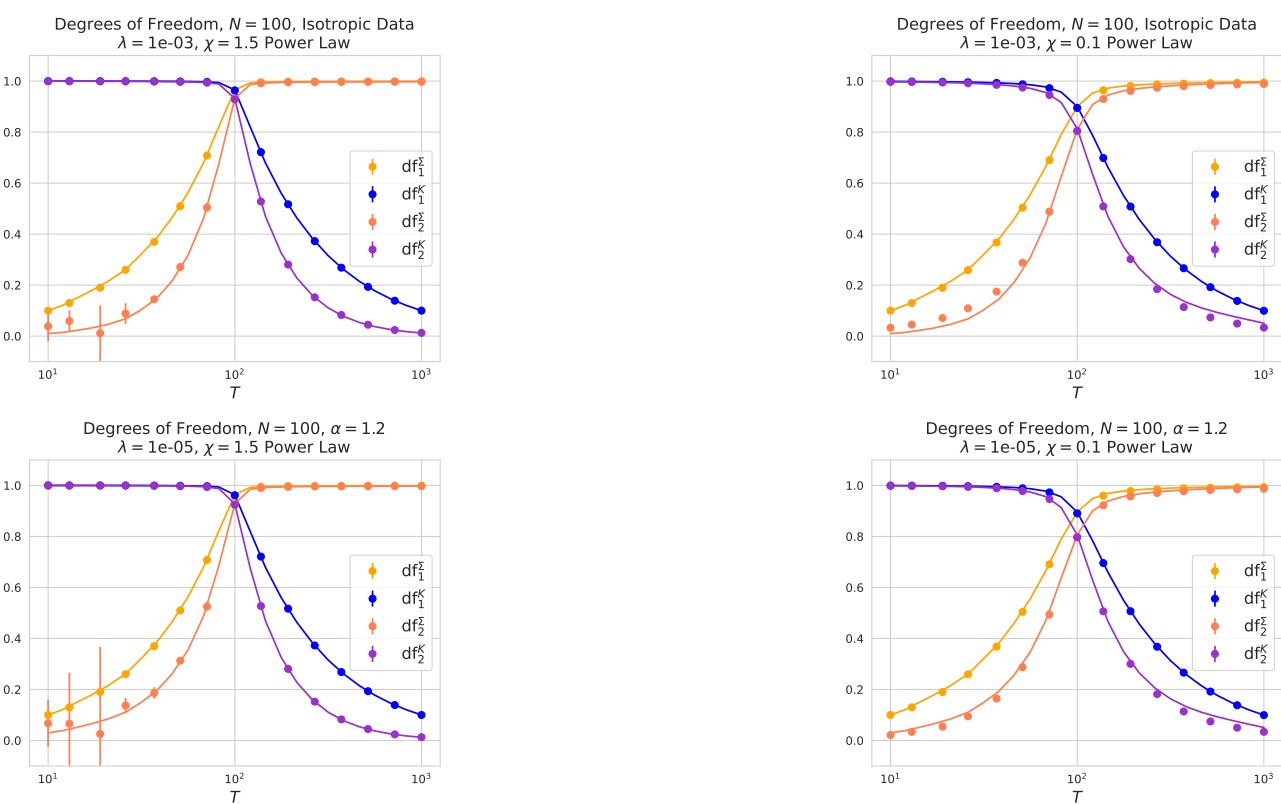

*Figure 19.* Plots of the the degrees of freedom $\mathrm{df}_1, \tilde{\mathrm{df}}_1, \mathrm{df}_2, \tilde{\mathrm{df}}_2$ under power law correlations as in Figure 18. Solid lines are theory, dots with error bars are empirics. The lack of substantial error bars stems from the fact that all quantities concentrate. We find that although $\mathrm{df}_2$ can be numerically sensitive, $\tilde{\mathrm{df}}_2$ remains numerically precise even in the presence of strong correlations.

## K. Experimental details

Error bars are always reported over ten runs of the same regression over different datasets. We ensembled over these datasets to calculate the $\mathrm{Var}_{\boldsymbol{X}}$ term empirically. We also held out a target free of label noise to calculate the $\mathrm{Var}_{\boldsymbol{X}\epsilon}$ component of the variance empirically.

We used JAX (Bradbury et al., 2018) to perform all the linear algebraic manipulations. All experiments were done primarily on a CPU with very little compute required. We also tested a speedup of running our code on a GPU, amounting to less than 1 GPU-day of compute usage. As mentioned in the main text, all code is available at https://github.com/Pehlevan-Group/S_transform.

