# OpenReview forum: "Risk and cross validation in ridge regression with correlated samples"
_ICML.cc/2025/Conference — ICML 2025 poster_

### Official Review · Reviewer_GGLG · 2025-03-07

**Overall Recommendation:** 3

**Summary:**

This is a theoretical paper that studies cross-validation in ridge regularized linear regression. The authors study an understudied regime in the literature on cross-validation: when the samples are *not* i.i.d. In this case, they show that the traditional generalized cross-validation (GCV) estimator does not correctly estimate the out-of-sample error. By theoretically analyzing the exact out of sample error, the authors derive a cross-validation estimator that *does* correctly estimate the out-of-sample error. The authors extend the results to the case of distribution shift (the test point at which one wishes to assess the out-of-sample error comes from a different distribution than the training data), and when the test point is correlated with the training data.

**Claims And Evidence:**

I want to be up front that I really struggled to understand the detailed results of the paper, as I don't have a background in random matrix theory / free probability.

My one suggestion along these lines is that the paper could do a much better job of highlighting what the important results are, breaking them out into theorem/lemma/proposition environments, and then discussing the importance of these results. As-is, the paper reads a bit like a stream of formulae, without much connection back to results that matter to the broader machine learning community. I say this because I think there's some good potential impact to the ML community in this paper, but it's a little hard to dig out as-is. The only significant discussion is the final paragraph of the conclusion which comments on the applications of the theory. I think significantly adding to that discussion and moving more of the technical results to the appendix would strengthen the paper and better connect it to the ML community.

**Essential References Not Discussed:**

I don't believe there is anything major missing.

**Experimental Designs Or Analyses:**

The paper is light on experiments because it's a theoretical paper. The use of synthetic data to back up the theoretical results is a nice addition.

**Methods And Evaluation Criteria:**

Yes, this is a theoretical paper, and the authors use synthetic data to demonstrate their results. This seems appropriate to me.

**Other Comments Or Suggestions:**

Nothing else

**Other Strengths And Weaknesses:**

1. I think the theoretical development could be better organized to help readers. First, as mentioned above, I think the major results should be broken out into Proposition / Lemma / Theorem environments. This would help not only highlight the major results but also collect all of the necessary assumptions. E.g., which results require the independence of the noise $\epsilon$ from the covariates $x$? This is stated in one sentence in Appendix C.5 (~line 1202), but I'm not sure which results it applies to.
2. It wasn't always clear where to find the proof of various results in the paper. E.g., the first main result is derived in Section 3.2, which just states that some of the results are derived "in Appendix C." But Appendix C contains many of the proofs.

**Questions For Authors:**

No questions.

**Relation To Broader Scientific Literature:**

Cross-validation (CV) has a long history in statistics and machine learning. Despite its long history, its theoretical properties are complicated. In the last ~10 years, a growing body of work has started to understand the theory of CV. However, this body of work has primarily focused on models fit to i.i.d. samples, which is not realistic in practice. So, while the model studied by the authors is relatively simple (ridge regression), it starts to fill in an important gap in our understanding of a widely used procedure (CV).

... all of this is assuming I've correctly understood the results of the paper!

**Theoretical Claims:**

I attempted to check the proofs of some of the claims, but I don't have the required mathematical background to do so.

I am a little worried whether much of the broader machine learning community has the needed background, and so whether ICML is an appropriate venue for this paper. An example of this is in Appendix C.1, which is a "warm-up" proof of preliminary results.The proof uses terms like "insertions", "Wick contractions", "crossing diagrams" that I've never heard of and struggled to find definitions of online. And the proof proceeds by drawing pictures, rather than using algebra, in a way that I'm also not familiar with.

---

> ### Author Rebuttal · Authors · 2025-03-30
>
> We thank the reviewer for their detailed and forthright assessment of our paper. We appreciate your concerns regarding the clarity of our manuscript, and will revise it to make its impact clearer.
>
> - Following your suggestion, we will re-organize the main text and Appendices into Theorem-Proof style. This will make the paper easier to navigate for the reader, and as you noted will break the lengthy Appendix C into more manageable chunks.
>
> - Your concerns regarding the accessibility of the proof methods to the ICML audience are well-taken. Our terminology, and the particular diagrammatic approach to the proof, is taken from physics. The core idea is to compute $\mathbb{E} \hat{\Sigma} (\hat{\Sigma} + \lambda I)^{-1}$ by expanding order-by-order in $1/\lambda$ and then re-summing the resulting series of moments. Here, "Wick contractions'' are pairings of indices corresponding to non-vanishing joint moments of elements of the Gaussian random matrix. This terminology comes from the fact that Isserlis' theorem for Gaussian moments is known in physics as "Wick's theorem'' (Wick's theorem in physics is in fact far more general, but the terminology is applied also in the case of a Gaussian random vector). This approach is closely related to the diagrammatic formulation of the moment method in random matrix theory, where one builds a diagrammatic notation to represent the combinatorics of which elements in the expansion of an expression like $\mathbb{E}_{M} \mathrm{Tr}(M^n)$ give non-negligible contributions. "Crossing diagrams'' refer to a subset of these pairings of indices that give rise to non-zero moments but do not contribute at leading order to the final result. "Inserting'' a matrix corresponds to adding it to an ordered product of matrices at a particular location. We apologize for this rather heavy use of jargon; we will clarify each term in the updated version of our manuscript.

---

> > ### Comment · Reviewer_GGLG · 2025-04-05
> >
> > Thanks for the response! The quick description here starts to help me understand some of the results. E.g. I am familiar with Isserlis' theorem, and I can kind of see how a diagramatic proof could be more helpful here over an algebraic one. I think adding more details about this proof technique / how Isserlis' theorem is being used will make the paper more self-contained for a general machine learning audience.
> >
> > Best, GGLG

---

> > > ### Author Response · Authors · 2025-04-07
> > >
> > > Thank you again for your valuable feedback!
> > >
> > > To briefly respond to the question about the application of Isserlis' theorem implicit in your comment: In the proof, one must (roughly speaking) study expectations of the form $\mathbb{E}[ (X^{\top} X)^{n} ]$ for some integer $n$, where $X$ is a $T \times N$ Gaussian matrix. This can be expanded as a sum over joint moment of elements of the Gaussian matrix $X$. For example, for $n=2$ we have $$\mathbb{E}[ (X^{\top} X)^{2} ]\_{ij} = \sum\_{k,l,m} \mathbb{E}[X\_{k i} X\_{k l} X\_{m l}  X\_{m j}].$$ Then, Isserlis' theorem can be applied to expand this joint moment as $$\mathbb{E}[X\_{k i} X\_{k l} X\_{m l}  X\_{m j}] = K\_{kk} \Sigma\_{il} K\_{mm} \Sigma\_{lj} + K\_{km} \Sigma\_{il} K\_{km} \Sigma\_{l j} + K\_{k m} \Sigma\_{ij} K\_{k m} \Sigma\_{ll},$$ which leads to $$\mathbb{E}[ (X^{\top} X)^{2} ]\_{ij} = \text{Tr}(K)^2 (\Sigma^2)\_{ij} + \text{Tr}(K^2) (\Sigma^2)\_{ij}  + \text{Tr}(K^2) \text{Tr}(\Sigma) \Sigma\_{ij}.$$ In the high-dimensional limit $N,T\to\infty$ with $N/T = \Theta(1)$, we assume that all traces are $\Theta(N)$: $\text{Tr}(\Sigma) \sim \text{Tr}(K) \sim \text{Tr}(\Sigma^2) \sim \text{Tr}(K^2) \sim \Theta(N)$. Therefore, the first and third terms are of the same order, while the $\text{Tr}(K^2) (\Sigma^2)\_{ij}$ term is sub-leading. This is a simple example, as one must figure out how to study moments with arbitrary $n$, but it illustrates the key components of the proof. The diagrammatic notation succinctly represents these types of sums.

---

### Official Review · Reviewer_XMHo · 2025-03-12

**Overall Recommendation:** 2

**Summary:**

The paper proposes CorrGCV, a modified version of the more well known generalized cross validation estimator (GCV) to estimate out-of-sample risk from in-sample data.

**Claims And Evidence:**

The theoretical derivations are sound.

**Essential References Not Discussed:**

Essential references have been mentioned.

**Experimental Designs Or Analyses:**

Simulations are sound, however the paper lacks real world application analyses.

**Methods And Evaluation Criteria:**

The theoretical derivations are sound, the experiments are convincing.

**Other Comments Or Suggestions:**

n/a

**Other Strengths And Weaknesses:**

* Modified CorrGCV is very interesting, and an improvement
* However, it is unclear how sensitive the results will breakdown if assumptions are violated
* Simulations are sound, however the paper lacks real world application analyses.

**Questions For Authors:**

* However, it is unclear how sensitive the results will breakdown if assumptions are violated: assumptions on correlation structure are too strcit
* Simulations are sound, however the paper lacks real world application analyses.

**Relation To Broader Scientific Literature:**

The paper proposes CorrGCV, a modified version of the more well known generalized cross validation estimator (GCV) to estimate out-of-sample risk from in-sample data.

**Theoretical Claims:**

The theoretical derivations are sound.

---

> ### Author Rebuttal · Authors · 2025-03-30
>
> We are glad that the referee found our theoretical results "sound'', and our experiments "convincing.'' We appreciate the referee's concerns regarding demonstration of real-world applications, but given that our paper is theoretical in nature we are surprised by the strongly negative assessment. In particular, it is the first theoretical analysis of high-dimensional ridge regression with correlated samples, and thus contributes to the extensive theoretical literature on this simple and fundamental learning algorithm.
>
> We would like to elaborate on why we believe the assumption of Gaussian covariates (under which our theoretical results were derived) is not highly restrictive. This assumption is motivated by the extensive literature on Gaussian universality in ridge regression: in the high-dimensional regime of interest, the risk obtained for some non-Gaussian dataset is asymptotically equal to that for Gaussian data of matched covariance (see, for instance, Hu and Lu https://arxiv.org/abs/2009.07669, Dubova et al https://arxiv.org/abs/2009.07669, or Misiakiewicz and Saeed https://arxiv.org/abs/2403.08938, or the many references cited within these works). Indeed, contemporaneous work from Luo et al (https://arxiv.org/abs/2406.11666) shows universality for data where features are uncorrelated but samples are correlated. And, since the submission of our manuscript, work by Moniri and Hassani (https://arxiv.org/abs/2412.03702) has shown universality for covariates of the form $X \overset{d}{=} K^{1/2} Z \Sigma^{1/2}$ where $Z$ has general sub-Gaussian i.i.d. entries; this generalizes our model in which we assumed the entries of $Z$ to be Gaussian. Therefore, we do not believe the assumptions under which are theoretical results are derived to be overly restrictive. We mentioned this motivation in our submitted manuscript, and will endeavor to make it more explicit in the revised version of our work.
>
> In light of this, we respectfully request the referee to reconsider their assessment. While a comprehensive experimental validation of the CorrGCV estimator with real data will be important to establish its broad applicability, to do this justice would require a manuscript-length study in its own right. Therefore, we believe that the theoretical contributions of this work can stand alone.

---

> > ### Comment · Reviewer_XMHo · 2025-04-06
> >
> > By assumptions on correlation structures are too unrealistic, and bounds are only asymptotic.

---

> > > ### Author Response · Authors · 2025-04-07
> > >
> > > We thank the referee for their continued engagement. We would like to clarify two points raised in their brief rebuttal comment:
> > >
> > > First, we would like to emphasize that our results are not bounds. Rather, they are sharp asymptotics in the sense that they precisely give the limiting behavior of the risk. As our experiments show, they are predictive even for relatively modest dimension and number of training examples. Moreover, though doing so would be outside the scope of the present paper, the finite-size corrections could in principle be entirely quantitatively controlled. One way to do so is to compute a series of corrections in $1/N$ and $1/T$, as discussed in the Appendices. Another way would be to quantitatively bound the error terms. In particular, we conjecture that it should be possible to prove explicit high-probability multiplicative error bounds roughly of the form $R = [1+O(T^{-1/2})] \mathcal{R}$ where $R$ is the out-of-sample risk and $\mathcal{R}$ is the asymptotic deterministic equivalent as derived in our work. For the case of independent training points, dimension-free bounds of this form have been obtained by Cheng and Montanari in https://arxiv.org/abs/2210.08571, and by Misiakiewicz and Saeed in https://arxiv.org/abs/2403.08938. These proofs are lengthy and rather technical, and were developed following detailed study of the sharp asymptotics previously obtained for independent data. In sum, we do not view the asymptotic nature of our results to be a strong limitation.
> > >
> > > Second, we would like to emphasize again that though our data model $X \overset{d}{=} K^{1/2} Z \Sigma^{1/2}$ does impose restrictions (in particular that the correlations across samples and across features factorize), previous works have obtained sharp asymptotics only for diagonal $K$, i.e., for independent training points. Therefore, we make a much weaker assumption on the correlation structure of the data than has been standard in the study of ridge regression. We respectfully request that you clarify why you do not view this generalization to be a sufficient advance. For instance, do you have a particular data generating model in mind?

---

### Official Review · Reviewer_BdGx · 2025-03-13

**Overall Recommendation:** 5

**Summary:**

This paper investigates the problem of high-dimensional ridge regression with correlated data, which is a common feature in time series. Using methods from RMT and free probability, the authors derive sharp asymptotics for the in- and out-of-sample risk, showing that the standard cross-validation estimator fails for non-iid data, as it is asymptotically biased. They introduce a corrected GCV estimator that is unbiased and accurately predicts the out-of-sample risk, extending their results to the setting where test points are correlated with training points. Finally, they show that correlations can smooth the double-descent peak.

**Claims And Evidence:**

All claims are supported by convincing evidence and limitations are clearly stated.

**Essential References Not Discussed:**

I am not aware of any essential references that have been omitted.

**Experimental Designs Or Analyses:**

I have no issues to discuss.

**Methods And Evaluation Criteria:**

The methods used are well suited to the problem considered, as they are frequently employed in the study of ridge regression problems.

**Other Comments Or Suggestions:**

-In several plots, the colors for CorrGCV and NaiveGCV_1 are very similar. I suggest changing one of them to improve clarity.

-Even if clearly distinguishable from the context, I believe that the notations ${\rm df}^2_{A}$ and ${\rm df}^2_{\Sigma\Sigma'}$ (etc.) may be misleading, since ${\rm df}^2_{\Sigma\Sigma'} \neq {\rm df}^2_{A = \Sigma\Sigma'}$.

-Typo on line 436: the reference authors are repeated twice.

**Other Strengths And Weaknesses:**

I have not identified any major weaknesses. The presentation is clear, and the authors provide detailed explanations on algorithmic implementations, experiments, and the background knowledge necessary to understand their derivations. Even with its limitations, which are appropriately discussed, this paper provides original and interesting theoretical results closely related to time series applications.

**Questions For Authors:**

I do not have additional important questions.

**Relation To Broader Scientific Literature:**

This work is related to several studies in the literature on ridge regression, extending for the first time previous findings to anisotropic data that exhibit correlations between samples (and additionally to correlations between test and training data). Most previous works have considered iid data or correlations in the label noise. Besides providing a corrected GCV estimator, the authors analyze the standard estimator as well as other estimators proposed in the literature, deriving sharp asymptotics and demonstrating their failure in high dimensions.

**Theoretical Claims:**

I have no issues to discuss.

---

> ### Author Rebuttal · Authors · 2025-03-30
>
> We thank the reviewer for their careful reading of our manuscript, and are gratified by their strongly positive assessment. We will update the paper to address all three of their comments:
>
> - We will change the color used for the CorrGCV to to distinguish it from the NaiveGCV\_1.
>
> - We will adopt the notation $\mathrm{df}^{2}\_{\Sigma, \Sigma'}$ to distinguish it from $\mathrm{df}^{2}\_{A=\Sigma\Sigma'}$. We would appreciate the reviewer's feedback on whether this notation is sufficiently distinct, or if further changes would be helpful.
>
> - We have fixed the typo on Line 436, thank you for catching this.

---

> > ### Comment · Reviewer_BdGx · 2025-04-08
> >
> > Thank you for your reply. The proposed notation seems sufficiently clear to me.

---

> > > ### Author Response · Authors · 2025-04-08
> > >
> > > We are glad that these changes address your concerns. Thanks again for your careful assessment of our submission!

---

### Official Review · Reviewer_E8tJ · 2025-03-16

**Overall Recommendation:** 3

**Summary:**

This paper employs novel techniques from random matrix theory and free probability to analyze the asymptotic properties of the generalized cross-validation (GCV) as an empirical risk estimator for high-dimensional ridge regression, particularly in settings with cross-sectional and temporal correlations in both the covariates and the label errors. By leveraging insights from the in-sample and out-of-sample risks, the authors develop a new consistent estimator, CorrGCV, for the out-of-sample risk when the label noise exhibits the same correlation structure as the covariates. The performance of CorrGCV is further examined through a series of numerical experiments. Overall, the paper is well-written and was a pleasure to read.

**Claims And Evidence:**

Yes.

**Essential References Not Discussed:**

1. Line 63-66, the authors state that "However, most of these works focus on correlations only in the label noise, and none of them show
that their estimators are asymptotically exact in high dimensions. " This is not entire accurate. There has been some work in the smoothing spline literature (which is essentially a high-dimensional Kernel ridge regression) on how to approximate the out-of-sample risk consistently when there are correlations. For example, [1]-[3].

2. The asymptotic properties of the GCV for independent data have also been investigated in depth in \cite{ref5} and \cite{ref6} for kernel ridge regression, which I believe are also relevant.

References:

[1]. Wang, Y. (1998). Smoothing spline models with correlated random errors. Journal of the American Statistical Association, 93(441), 341-348.

[2]. Xu, G., & Huang, J. Z. (2012). Asymptotic optimality and efficient computation of the leave-subject-out cross-validation.

[3]. Gu, C., & Ma, P. (2005). Optimal smoothing in nonparametric mixed-effect models.

[5]. Xu, G., Shang, Z., & Cheng, G. (2018). Optimal tuning for divide-and-conquer kernel ridge regression with massive data. In International Conference on Machine Learning (pp. 5483-5491). PMLR.

[6]. Xu, G., Shang, Z., & Cheng, G. (2019). Distributed generalized cross-validation for divide-and-conquer kernel ridge regression and its asymptotic optimality. Journal of computational and graphical statistics, 28(4), 891-908.

**Experimental Designs Or Analyses:**

For the most part, the experimental designs make sense; however, I would like to see a bit more. For example, the purpose of cross-validation is essentially to choose the optimal tuning parameter $\lambda$ that minimizes the out-of-sample risk. For this approach to be effective, CorrGCV must accurately approximate the risk over a wide range of $\lambda$ values. Therefore, it would be beneficial to verify this behavior, rather than relying solely on the current setting with a fixed $\lambda = 0.001$.

**Methods And Evaluation Criteria:**

Yes.

**Other Comments Or Suggestions:**

I believe that the motivation for the model considered in (1) needs to be significantly strengthened. A practical approach would be to provide concrete examples that demonstrate the usefulness of model (1) in real-world applications. In particular, since the proposed CorrGCV is specifically designed for the case where $K=K'$, it would be beneficial to include examples that illustrate why this assumption is both common and useful in practice.

**Other Strengths And Weaknesses:**

Strength: The analysis is thorough and informative, and the paper is quite well written.

Weakness: I am not sure how widely applicable the proposed CorrGCV criterion is in practice, since it requires that the label noise has the same correlation structure as the covariates ($K=K'$). Such a requirement does not seem to be very common in practical applications.

**Questions For Authors:**

1. In the last paragraph of Section 1, the authors discuss the weighted least squares loss with a weight matrix $M$. It is stated that "this is equivalent to considering an isotropic loss under the mapping ... and as a result, our asymptotics apply immediately to general choices of $M$." I am afraid that this is not so simple. Under such a mapping, one would have to change the definition of the response vector $\mathbf{y}$, and consequently, the definition of the risk would become dependent on $M$. Therefore, the theory would be applied to a different in-sample and out-of-sample risk than that originally defined for $\mathbf{y}$. Could you please clarify this point?

2. Can you provide more details on the derivation of the equations in (5)? As someone who is not familiar with free probability theory, I find it difficult to understand why these equations hold. Is this due to some special properties of Wishart matrices?

3. What are the conditions on $K$ and $\Sigma$ for equation (6) to hold? There is no requirement on the eigenvalues of $K$ and $\Sigma$？

4. In my opinion, the most interesting case is discussed in Section 3.3, as I believe it is very common in practice. However, the conclusion from this section appears to be that "there is nothing we can do to approximate the out-of-sample risk." Is my understanding correct? If so, this outcome is quite disappointing and significantly limits the practical value of this work, as assuming $K=K'$ is rather restrictive. Could you explore the possibility of developing an alternative cross-validation method to approximate the out-of-sample risk based on the relationship between $R_g$ and $\hat{R}$?

**Relation To Broader Scientific Literature:**

I believe that the results of the paper enhance our understanding of the behavior of the out-of-sample risk when correlations are present in the data.

**Theoretical Claims:**

Yes.

---

> ### Author Rebuttal · Authors · 2025-03-30
>
> Thank you for your careful assessment of our manuscript. First, with regards to your questions and concerns:
>
> 1. You are correct in stating that with the introduction of a weighting $M$ the definition of the effective vector of measured responses changes. Our observation here is simply that under our Gaussian statistical assumption on the matrix of observed covariates $X \overset{d}{=} K^{1/2} Z \Sigma^{1/2}$ and noise $\epsilon \overset{d}{=} (K')^{1/2} \eta$ (for $Z$ and $\eta$ having i.i.d. Gaussian elements), the minimizer of the weighted risk is equal in distribution to the minimizer of an unweighted risk with $K$ and $K'$ replaced by $M^{1/2} K M^{1/2}$ and $M^{1/2} K' M^{1/2}$, respectively. Therefore, one can apply the same results to obtain the asymptotics of the out-of-sample risk as defined in Line 147-148, which does not depend on the choice of weighting. We will revise the discussion starting on Line 157, and Appendix H, to clarify this issue.
>
> 2. For a discussion of the derivation of (5), please see our response to Reviewer GGLG, who voiced similar concerns about the accessibility of the proof. In brief, the multiplicative property of the $S$-transform is one of the fundamental results of free probability. In this case it follows from studying the asymptotics of the resolvent of a Wishart matrix, but this multiplicative property extends to more general products of ``free'' random matrices (see Appendix B or the textbook of Potters and Bouchaud for a definition).
>
> 3. The basic assumption we need to make on $K$ and $\Sigma$ is that their empirical eigenvalue distributions tend to well-defined limiting measures as $T$ and $N$ tend to infinity, respectively. The support of these measures should be bounded strictly away from zero, and spectral moments of all orders (e.g., $\frac{1}{T} \mathrm{Tr}(K^{n})$) should be bounded. In our revised manuscript, we will make this clearer by stating these conditions as an explicit Assumption (as part of re-organization into Theorem-Proof style, as requested by Reviewer GGLG).
>
> 4. Our focus on the case $K=K'$ is fundamentally motivated by our theoretical result that in that case the asymptotic risks are proportional. This is the same proportionality property that enables the definition of the (easily-computable) classic GCV as an asymptotically unbiased estimator of the out-of-sample risk for independent training samples. Our analysis shows that if $K \neq K'$ then the training and test risks are not in general directly proportional, which means that an asymptotically un-biased estimator should not have the simple form of the GCV. This does not, however, mean that no estimator of the out-of-sample risk exists; it just must be something more than a multiplicative correction to the training risk.
> One practically-relevant setting in which one would expect to have $K=K'$ is if the noise arises through components of the target function that cannot be learned through linear regression. Namely, nonlinear components will act as effective noise, as is discussed in the literature on Gaussian universality cited in our response to Reviewer XMHo. As mentioned there, we believe that a comprehensive examination of practical applications of the CorrGCV estimator is beyond the scope of this paper, as it is primarily theoretical in its aims. We will expand our discussion of Gaussian universality in our revised manuscript.
>
>
> In addition to the changes reflected above, we will make the following additions to our manuscript:
>
> - We will add a figure showing a sweep over values of $\lambda$ to show that the CorrGCV is broadly effective.
>
> - We will add citations to the five references you suggested; we regret that we missed these relevant works in preparing our manuscript.

---

> > ### Comment · Reviewer_E8tJ · 2025-04-05
> >
> > I would like thank the authors for addressing my questions.
> >
> > For 1,  what I meant is that the weighted risk is different from the unweighted risk function, and depends on the chosen weight matrix. The purpose of weighted least square is to reduce the unweighted risk by applying an appropriate weight matrix. If you switch the goal to minimize the weighted risk, the approach is less meaningful.
> >
> > For 4, I am still looking for a concrete example where $K=K'$ is a necessary assumption.

---

> > > ### Author Response · Authors · 2025-04-07
> > >
> > > Thank you for following up; we would like to clarify our responses to the two points mentioned.
> > >
> > > We regret that our description of how the weighted risk is used was not clear. To be concrete, we consider a case in which the estimator $\hat{w}\_{M} = \text{argmin}\_{w} \hat{R}\_{M}(w)$ is defined by minimizing a weighted in-sample risk $\hat{R}\_{M}(w) = (Xw-y)^{\top} M (Xw-y)$ for some weighting matrix $M$. Then, we consider the out-of-sample risk $R(\hat{w}\_{M}) = \mathbb{E}\_{(x,y)}[ (\hat{w}\_{M}^{\top} x - y)^{2} ]$ for that estimator. Here, the distribution of the test sample is given by $x \sim \mathcal{N}(0,\Sigma)$ and $y\,|\, x \sim \mathcal{N}(\bar{w}^{\top} x, \sigma_{\epsilon}^2)$. The definition of the out-of-sample risk---which is, as the reviewer notes, what we fundamentally want to minimize---is not affected by the choice of $M$. What we compute is the asymptotics of $R(\hat{w}\_{M})$, i.e., the out-of-sample risk for the weighted estimator. We hope that this clarification addresses your concern; we will revise our manuscript to make this point clear.
> > >
> > > We emphasize again that the condition $K = K'$ is a *result* of our asymptotic analysis, not an assumption, and our theoretical results include the case $K \neq K'$. We do not quite follow what the reviewer seeks in terms of a case where $K = K'$ is a *necessary* assumption. Our point is that $K = K'$ is sufficient for there to exist an asymptotically unbiased non-omniscient estimator of the out-of-sample risk that can be written as a multiplicative correction to the in-sample risk. With regards to particular examples where $K = K'$, we can of course identify generative models where this is naturally the case (see below), but cannot provide a complete taxonomy.
> > >
> > > In regards to specific generative models in which one has $K = K'$:
> > >
> > > - Suppose that one observes only a subset of the relevant features, and fits a mis-specified model. Then, as discussed at length in Section 5 of Hastie et al, https://arxiv.org/abs/1903.08560, one will incur additional out-of-sample risk due to mis-specification bias, as well as mis-specification variance that acts identically to additive noise on the targets. This case is of clear practical relevance, as in many settings one cannot observe all relevant covariates.
> > >
> > > - Suppose that the target function has un-learnable modes. That is, assume that there are components of the target function that cannot be learned with ridge regression even with infinite training data, which would correspond to vanishingly small eigenvalues of $\Sigma$ (see, for instance, Canatar et al., https://www.nature.com/articles/s41467-021-23103-1, Xiao et al. https://iopscience.iop.org/article/10.1088/1742-5468/ad01b7, or Atanasov et al., https://arxiv.org/abs/2405.00592 for related discussions). This is closely related to the presence of un-observed features. Then, these components will act as an effective noise with covariance $K$.
> > >
> > > We will mention these and other examples in the updated version of our manuscript. We would like to re-iterate that a comprehensive empirical study of how well the proposed CorrGCV estimator performs when applied to real data requires a manuscript in its own right. We acknowledge that this is a limitation of the present work, but emphasize that our goal here is to lay theoretical foundations, as past works have not derived precise asymptotics for high-dimensional ridge regression with correlated samples.

---

### Decision · Program_Chairs · 2025-05-01

**Decision:**

Accept (poster)

**Comment:**

The submission has as main goal the study of a suitable cross validation estimator in the case of correlated samples. It has collected an overall positive feedback because of the novelty of the approach, based on random matrix theory and free probability tools, and of the results. The text clarity has also been remarked. The work has a theoretical nature and by consequence some concerns on the actual real-world application of the proposed estimator and analysis have been raised (due to a number of assumptions required for a full theoretical investigation). Nevertheless, the paper presents an interesting analysis on its own that might serve as starting point for the study of cross validation in presence of correlation. I thererefore suggests *Weak Accept*.